# A mechanosensitive caveolae–invadosome interplay drives matrix remodelling for cancer cell invasion

Pedro Monteiro ®[1,2] ✉, David Remy[1], Eline Lemerle[3], Fiona Routet[1], Anne-Sophie Macé[4], Chloé Guedj ®[4], Benoit Ladoux ®[5], Stéphane Vassilopoulos ®[3], Christophe Lamaze ®[2,6] ✉ & Philippe Chavrier ®[1,6] ✉

Invadosomes and caveolae are mechanosensitive structures that are implicated in metastasis. Here, we describe a unique juxtaposition of caveola clusters and matrix degradative invadosomes at contact sites between the plasma membrane of cancer cells and constricting fibrils both in 2D and 3D type I collagen matrix environments. Preferential association between caveolae and straight segments of the fibrils, and between invadosomes and bent segments of the fibrils, was observed along with matrix remodelling. Caveola recruitment precedes and is required for invadosome formation and activity. Reciprocally, invadosome disruption results in the accumulation of fibril-associated caveolae. Moreover, caveolae and the collagen receptor β1 integrin co-localize at contact sites with the fibrils, and integrins control caveola recruitment to fibrils. In turn, caveolae mediate the clearance of β1 integrin and collagen uptake in an invadosome-dependent and collagen-cleavage-dependent mechanism. Our data reveal a reciprocal interplay between caveolae and invadosomes that coordinates adhesion to and proteolytic remodelling of confining fibrils to support tumour cell dissemination.

Extracellular matrix (ECM) remodelling is fundamental to physiological processes such as tissue morphogenesis and wound healing and contributes to pathological conditions such as inflammatory diseases, tissue fibrosis and cancer progression[1–4]. During breast cancer progression, carcinoma cells escape from the primary tumour by breaching the basement membrane that surrounds ductal epithelial cells and infiltrate the collagenous connective tissue of the stroma[3,5]. To invade through the fibrous type-I-collagen-rich network, tumour cells cleave the fibrils by forming specialized actin-based non-protruding cell–ECM contact sites, generically termed invadosomes (invadopodia being viewed as protrusive counterparts)[6], endowed with pericellular collagenolytic activity. The concerted action of invadosomal actin filament polymerization that produces forces pushing against the plasma membrane (PM) that are transmitted to the contacted collagen fibrils, and membrane type I matrix metalloproteinase (MT1-MMP)-mediated collagen cleavage—which increases fibril compliance—is key to the formation of invasive tracks by tumour cells[7–10]. MT1-MMP is essential for invadosome formation and pericellular ECM breakdown, and its upregulation

[1]Actin and Membrane Dynamics Laboratory, Institut Curie—Research Center, CNRS UMR144, PSL Research University, Paris, France. [2]Membrane Mechanics and Dynamics of Intracellular Signalling Laboratory, Institut Curie—Research Center, CNRS UMR3666, INSERM U1143, PSL Research University, Paris, France. [3]Institute of Myology, Sorbonne Université, INSERM UMRS 974, Paris, France. [4]Cell and Tissue Imaging Facility (PICT-IBiSA), Institut Curie, PSL Research University, Paris, France. [5]Institut Jacques Monod, Université de Paris, CNRS UMR 7592, Paris, France. [6]These authors contributed equally: Christophe Lamaze and Philippe Chavrier. ✉e-mail: pedro.monteiro@curie.fr; christophe.lamaze@curie.fr; philippe.chavrier@curie.fr

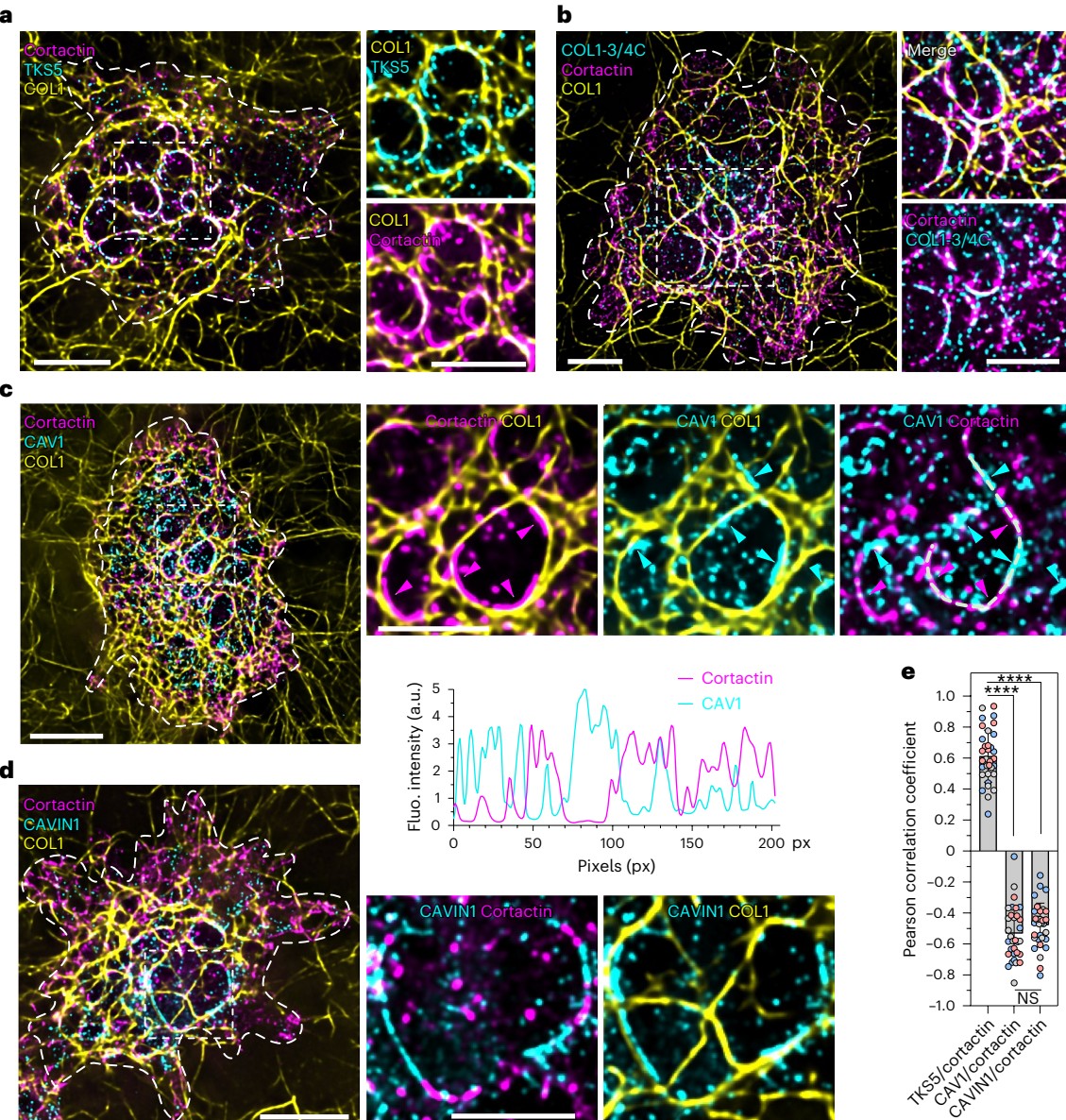

**Fig. 1 | Alternating distribution of caveolae and invadosomes along confining type I collagen fibrils. a–b,** MDA-MB-231 cells cultured on a layer of fibrous type I collagen for 60 min were fixed and stained for the indicated markers. The dotted line indicates the cell contour. Insets: magnified views of the boxed regions. The distribution of key invadosome components, TKS5 and cortactin (**a**), and invadosome collagenolytic activity (cleaved collagen neo-epitope detected using COL1-3/4C antibodies) (**b**) are shown. **c,** The alternating caveolae and invadosome distribution revealed by profiling of the cortactin and CAV1 signal intensity in association with the collagen fibril (line scan). The arrowheads indicate accumulations of CAV1 (cyan) and cortactin (magenta).

**d,** Alternating CAVIN1 and cortactin distribution. **e,** Pearson correlation coefficient calculated based on TKS5/cortactin, CAV1/cortactin and CAVIN1/cortactin signal intensity profiles. Data are mean ± s.d. $n = 35$ (TKS5/cortactin), $n = 32$ (CAV1/cortactin) and $n = 33$ (CAVIN1/cortactin) line scans from three biologically independent experiments. Statistical analysis was performed using one-way analysis of variance (ANOVA) with Kruskal–Wallis test; ****$P < 0.0001$; NS, not significant. The distribution of individual pixel intensity values is shown in Extended Data Fig. 1a,d,e. Source numerical data are provided as Source Data. Scale bars, 10 μm (**a,c** and **d** (left) and **b**) and 5 μm (**a,c** and **d** (right)). a.u., arbitrary units; fluo., fluorescence.

in breast cancer is correlated with poor prognosis and lower survival[11–14]. F-actin assembly at invadosomes requires the ARP2/3 complex and upstream activators, the Rho-GTPase CDC42 and effector N-WASP, as well as the F-actin branch stabilizer cortactin and the adaptor protein TKS5 (tyrosine kinase substrate with five SH3 domains)[15]. How cancer cells sense the composition, topology and mechanical property of the matrix environment to mount the adapted collagenolytic response remains unclear. Reports indicate that ECM adhesion through integrins and integrin signalling have a role in the regulation of MT1-MMP trafficking and invadosome formation in different cancer cell types[16–18].

Caveolae are bulb-shaped PM invaginations with a diameter of 60–80 nm composed of two major structural protein families— caveolins and cavins[19,20]. Caveolin-1 (CAV1), the main caveolae component, is required for caveola formation[21]. CAV1 binds to lipids including cholesterol, and can oligomerize and assemble as 8S scaffolds that combine and interact with cavins to form caveola pits[22–24]. Caveolae show high levels of plasticity, ranging from individual pits to higher-order rosette structures or even flattened membrane subdomains scaling with membrane tension[25–27]. Caveola plasticity underlies a diversity of cellular functions such as membrane lipid subdomain organization,

lipid and protein uptake and signal transduction[27–29]. Furthermore, the bulb shape confers substantial membrane buffering capacity to caveolae that can function as mechanosensors and mechanoprotectors in PM adaptation to environmental mechanical cues[25,30–33]. Furthermore, CAV1 is involved in integrin signalling and focal adhesion assembly and dynamics[34]. Depending on cancer type and stage, caveolins and cavins have been implicated both in tumour suppression and oncogenesis[31]. Particularly, CAV1 acts as a tumour suppressor at early stages of breast cancer development, whereas it is linked to tumour progression and metastasis at later stages[31,35–37].

Early reports correlated CAV1 loss-of-function with perturbation of invadosome gelatinolytic activity[38,39]. Yet, the mechanism underlying caveola function in matrix degradation is unclear, and whether and how caveolae and invadosomes could work in concert during cancer cell invasion has not been addressed. In this Article, we unveil a cross-talk between invadosomes and caveolae in invasive cancer cells. We detected the presence of caveolae at PM–collagen fibril contact sites in an alternating distribution with invadosomes. Silencing of core caveola components negatively impinged on invadosome formation and activity. Reciprocally, invadosome disruption resulted in fibril-associated caveola accumulation. By monitoring fibril-associated caveolae and invadosomes over time, we showed that the peak of CAV1 and CAVIN1 recruitment preceded invadosome formation. Moreover, we observed that CAV1/CAVIN1 and β1 integrin collagen receptors co-localized at PM–fibril contact sites, with integrins controlling CAV1 recruitment to the fibrils. In turn, CAV1 mediated surface β1 integrin clearance together with bound ECM in an invadosome- and collagen-cleavage-dependent mechanism. Our data highlight a distribution and adhesion function for caveolar structures, which promote invadosome formation and control ECM remodelling, supporting a reciprocal interplay between invadosomes and caveolae in sensing and remodelling constricting matrix fibrils during tumour cell dissemination.

## Results

### Alternating assembly of caveolae and invadosomes at PM–collagen fibril contact sites

When cultured on top of a layer of type I collagen fibrils, invasive tumour cells form collagenolytic structures named invadosomes in close association with the matrix fibrils[40]. Accordingly, 60 min after plating MDA-MB-231 breast adenocarcinoma cells on the fibrillar network, invadosomes were detected as F-actin-, cortactin- and TKS5-enriched curvilinear structures that formed at contact sites between the PM and the underlying collagen fibrils[14,41,42] (Fig. 1a and Extended Data Fig. 1a). The transmembrane matrix metalloproteinase MT1-MMP accumulated in invadosomes and mediated cleavage of contacted fibrils[14,41,42] (Fig. 1b and Extended Data Fig. 1b,c). To assess whether and how caveolae contributed to invadosome formation and activity, we examined the intracellular distribution of CAV1. In addition to a scattered distribution of small-sized, dotty structures at the PM (Fig. 1c), immunofluorescence analysis revealed the accumulation of CAV1 in larger, elongated structures aligned with the underlying fibrils (Fig. 1c (cyan arrowheads)). Notably, CAV1 accumulations did not co-localize, but were excluded from TKS5-, cortactin- and MT1-MMP-positive invadosomes as indicated by intensity profiling along the fibrils and the negative Pearsons coefficient of the two markers (Fig. 1c,e (magenta arrowheads) and Extended Data Fig. 1b,d). Staining revealed an alternating organization of caveolar and invadosome structures along the remodelled fibrils. Although CAV1 represents the major core caveola component, several reports have suggested that CAV1 also exists as oligomers or scaffolds that may have roles outside caveolae[22,43,44]. We therefore investigated whether CAV1-positive structures at PM–fibril contact sites could represent bona fide caveolae using CAVIN1—a component of the caveola coat[29,45,46]. CAVIN1 co-localized with CAV1 in PM structures, both outside and along the fibrils (Extended Data Fig. 1f), and it alternated with cortactin- and MT1-MMP-positive invadosomes along collagen fibrils (Fig. 1d,e and Extended Data Fig. 1c,e). Taken together, these results indicate that caveolar structures accumulate along collagen fibrils close to, but non-overlapping with invadosomes, and suggest the coalescence of fibril-associated caveolae into large clusters. Notably, we found no overlap between peripheral focal adhesions stained with paxillin (and weakly positive for cortactin) and more-internal, elongated, cortactin-rich paxillin-negative invadosomes, although minimal colocalization cannot be ruled out (Extended Data Fig. 1g).

### Caveolae and invadosomes segregate according to fibril curvature

Bending of the matrix fibrils depends on a dual action of actin polymerization-based pushing forces and MT1-MMP-mediated proteolytic cleavage of the fibrils at invadosomes[14]. First, we analysed the architecture of the collagen network at large scale by comparing the general fibril orientation in cell-associated and cell-free areas of the network. We observed that co-orientation of the fibrils was higher outside the cells coinciding with aligned, straighter fibrils, whereas it was reduced and fibrils were more curved underneath the cells with high invadosomal activity (Fig. 2a,b and Extended Data Fig. 2a).

**Fig. 2 | Caveola and invadosome distribution along PM–collagen fibril contact sites correlate with straight versus bent collagen fibril segments. a**, Regions of analysis of the collagen network. Cell staining with cortactin is shown in Extended Data Fig. 2a. **b**, Quantification of the collagen fibril orientation index using the OrientationJ Dominant Direction module in Fiji (Methods) in cell-free or cell-associated regions of the collagen network measured 30 min or 60 min after cell seeding. For the box plots, the centre line shows the mean, the box limits show the first (Q1, 25th percentiles) to third (Q3, 75th percentiles) quartiles of the distribution, and the whiskers show the minimum to maximum values. $n = 581$ (cell free), $n = 72$ (cell associated, 30 min) and $n = 73$ (cell associated, 60 min) fields from three biologically independent experiments. Statistical analysis was performed using one-way ANOVA with Kruskal–Wallis test; *$P = 0.0395$; ***$P = 0.0005$; ****$P < 0.0001$. **c**, MDA-MB-231 cells cultured on a layer of fibrous type I collagen for 60 min were stained. Left, the distribution of invadosome (cortactin) and caveolae (CAV1) markers along the confining fibrils. The dotted lines and arrows indicate the position and direction of the profiled line scan, respectively. Middle, representation of the tangential curvature along collagen fibrils using the indicated lookup table (LUT). Right, the cortactin and CAV1 signal intensity and tangential curvature were plotted for each pixel. **d**, The frequency distribution of tangential curvature in caveolae or invadosomes based on the cortactin (cort)/ CAV1 intensity ratio (caveolae, blue bars: cortactin/CAV1 ratio < 1; invadosomes, red bars: cortactin/CAV1 ratio > 1) classes. Data are mean ± s.d. $n = 3$ biologically independent experiments. Statistical analysis was performed using two-way ANOVA with Šídák's correction; **$P = 0.003$, ***$P = 0.0004$, ****$P < 0.0001$. **e**, Analysis as in **c** comparing cortactin and CAVIN1 markers. Left, the distribution of cortactin and CAVIN1 markers along the confining fibrils. The dotted lines and arrows indicate the position and direction of the profiled line scan, respectively. Middle, representation of tangential curvature along collagen fibrils using the indicated LUT. Highly curved segments are numbered. Right, cortactin and CAVIN1 signal intensities and tangential curvature are plotted for each pixel. **f**, The frequency distribution of tangential curvature as in **d** based on the cortactin/CAVIN1 intensity ratio (caveolae, blue bars: cortactin/CAVIN1 ratio < 1; invadosomes, red bars: cortactin/CAVIN1 ratio > 1) classes. Data are mean ± s.d. $n = 3$ biologically independent experiments. Statistical analysis was performed using two-way ANOVA with Šídák's correction; *$P < 0.01$, **$P < 0.005$, ***$P = 0.0001$, ****$P < 0.0001$. **g**, The top row is a gallery of selected time frames (in minutes) from a time-lapse sequence of MDA-MB-231 cells expressing CAV1–GFP (cyan) and DsRed–cortactin (magenta) plated on a layer of Cy5-labelled collagen I fibrils (yellow). The bottom row is a representation of tangential curvature along a selected collagen fibril using the indicated LUT. Source numerical data are provided as Source Data. Scale bars, 10 μm (**a**), 5 μm (**c** and **e**) and 2 μm.

Furthermore, we noticed that the fibril orientation metrics decreased over time by comparing cell-associated networks 30 min or 60 min after seeding (Fig. 2b, Extended Data Fig. 2b and Supplementary Video 1). Collectively, these data confirm the ability of MDA-MB-231 cells to remodel the pericellular ECM through collagen cleavage and fibril bending[14].

Given the alternation of caveola and invadosome structures, we lowered the scale of our analysis at the fibril level by profiling tangential curvature along individual fibrils using the Kappa Curvature module in Fiji. While invadosomes associated with highly curved, outward-pointing portions of the fibrils (Fig. 2c–f and Extended Data

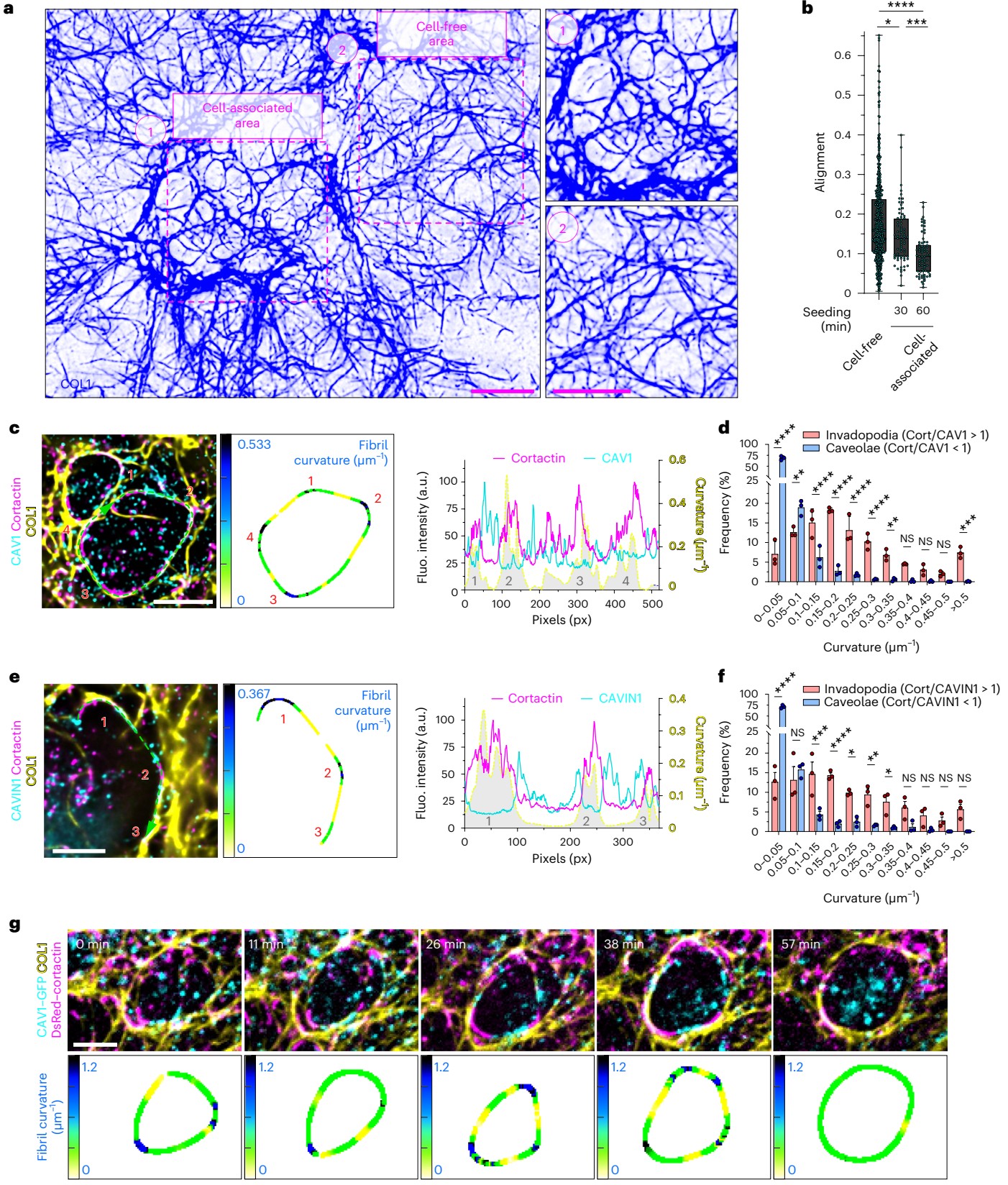

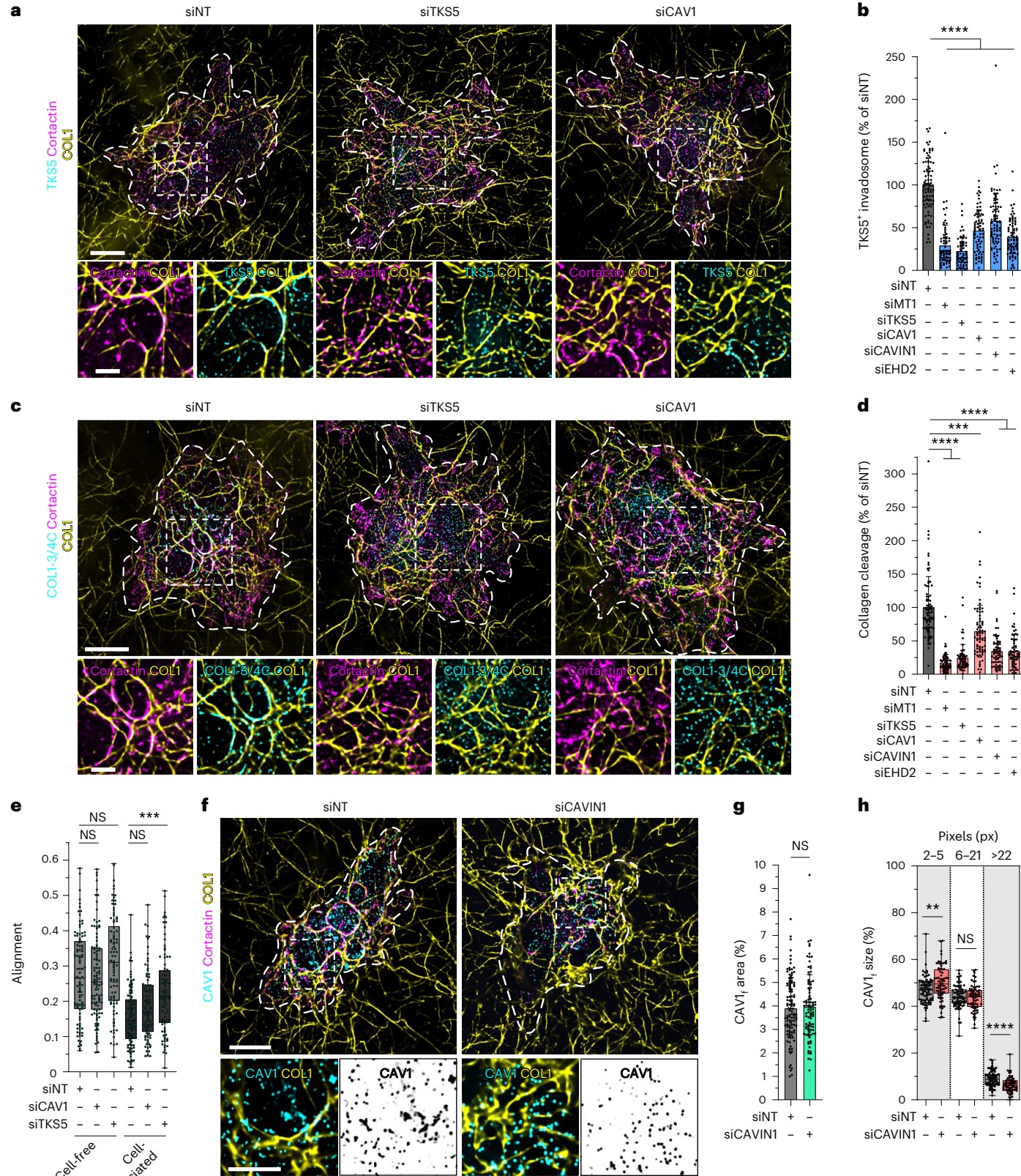

Fig. 2c), CAV1/CAVIN1-positive caveola components preferentially associated with straighter fibril segments (Fig. 2c–f). CAV1 and cortactin intensity signals, as well as curvature, were profiled for each pixel along the fibril (line-scan analysis), and pixels were stratified according to their cortactin/CAV1 intensity ratio and correlated with pixel curvature value. Whereas caveolae (cortactin/CAV1 ratio < 1) were associated with low curvature (that is, straighter segments), invadosomes (cortactin/CAV1 ratio > 1) were present on more-curved portions of the fibrils (Fig. 2d). Similar observations were made with CAVIN1 (Fig. 2e,f). Live-cell imaging of MDA-MB-231 cells expressing CAV1–GFP and DsRed–cortactin confirmed the alternating distribution of caveolae and invadosomes along collagen fibrils (Fig. 2g and Supplementary

**Fig. 3 | Caveolae are required for invadosome function. a**, MDA-MB-231 cells with the indicated KDs using siRNA treatment were cultured on a layer of type I collagen fibrils for 60 min, fixed and stained for TKS5 and cortactin invadosome markers. The dotted line shows the cell contour. Insets: magnified views of the boxed regions. **b**, Quantification of TKS5-positive invadosomes normalized to control siNT-treated cells. Data are mean ± s.d. $n = 89$ (siNT), $n = 75$ (siMT1-MMP), $n = 67$ (siTKS5), $n = 77$ (siCAV1), $n = 83$ (siCAVIN1) and $n = 77$ (siEHD2) cells from three biologically independent experiments. Statistical analysis was performed using one-way ANOVA with Kruskal–Wallis test; ****$P < 0.0001$. **c**, Cells were stained for the cleaved type I collagen neoepitope COL1-3/4C and cortactin. Insets: magnified views of the boxed regions. **d**, Type I collagen cleavage (on the basis of the COL1-3/4C signal) by the indicated cell populations. Data were normalized to control siNT-treated cells. Data are mean ± s.d. $n = 84$ (siNT), $n = 75$ (siMT1-MMP), $n = 65$ (siTKS5), $n = 78$ (siCAV1), $n = 67$ (siCAVIN1) and $n = 68$ (siEHD2) cells from three biologically independent experiments. Statistical analysis was performed using one-way ANOVA with Kruskal–Wallis test; ***$P = 0.0005$, ****$P < 0.0001$. **e**, Quantification of the collagen fibril orientation index in cell-free and cell-associated regions of the collagen network for the indicated populations of siRNA-treated cells. $n = 102$ (cell free, siNT), $n = 103$

(cell free, siCAV1), $n = 90$ (cell free, siTKS5), $n = 104$ (cell associated, siNT), $n = 89$ (cell associated, siCAV1), $n = 89$ (cell associated, siTKS5) fields from three biologically independent experiments. Statistical analysis was performed using one-way ANOVA with Kruskal–Wallis test; ***$P = 0.0004$. **f**, MDA-MB-231 cells knocked down for *CAVIN1* and cultured on type I collagen for 60 min were stained for CAV1 and cortactin. Insets: magnified views of the boxed regions. **g**, The ratio (as a percentage) of the area of cell-underlying fibrils with overlapping CAV1 signal to the total surface of cell-underlying fibrils (CAV1$_f$ area). Data are mean ± s.d. $n = 112$ (siNT) and $n = 95$ (siCAVIN1) cells from four biologically independent experiments. Statistical analysis was performed using unpaired two-tailed *t*-tests. **h**, The effect of *CAVIN1* silencing on the size distribution of fibril-associated CAV1-positive structures stratified in three different size classes (CAV1$_f$ size). $n = 82$ (siNT) and $n = 70$ (siCAVIN1) cells from three biologically independent experiments. Statistical analysis was performed using unpaired two-tailed *t*-tests; **$P = 0.0021$, ****$P < 0.0001$. For the box plots in **e** and **h**, the centre line shows the mean, the box limits show the first (Q1, 25th percentiles) to third (Q3, 75th percentiles) quartiles of the distribution, and the whiskers show the minimum to maximum values. Source numerical data are provided as Source Data. Scale bars, 10 μm (**a**, **c** and **f** (top)) and 5 μm (**f** (bottom)).

Video 2). Taken together, these data uncover the alternation of softened, bent-fibril regions due to invadosome activity, and straighter, possibly more rigid, caveola-associated segments of the matrix fibrils.

## Ultrastructural and super-resolution characterization of collagen fibril-associated caveolae

To better understand the organization of caveolar structures in association with collagen fibrils, MDA-MB-231 cells plated on top of the collagen network were unroofed and processed for platinum replica electron microscopy (PREM), and immunogold labelling of GFP-tagged CAV1 or CAVIN1. We observed accumulations of CAV1–GFP or CAVIN1–GFP in 60–80-nm-sized buds characteristic of caveolae in fibril-free regions of the PM (Extended Data Fig. 3a–c (arrows)). Moreover, CAV1–GFP and CAVIN1–GFP labelling was detected in micrometre-sized clusters of caveolar structures associated with the ridge formed by the PM wrapped around the electron-dense collagen fibrils lying underneath the ventral cell surface[14] (Extended Data Fig. 3a–c (arrowheads)).

Moreover, using stochastic optical reconstruction microscopy (STORM)[47], we confirmed the presence of small, low-intensity CAV1-positive profiles that probably correspond to caveola pits in

fibril-free regions of the PM and some in association with the fibrils, as well as brighter fibril-associated caveolar clusters juxtaposed with cortactin-positive invadosomes (Extended Data Fig. 3d). Taken together, high-resolution light and electron microscopy data demonstrate that individual and clustered caveolar structures are found at PM–collagen fibril contact sites in breast cancer cells, and these caveolar structures are excluded from and alternating with invadosomes. These results suggest potential caveola plasticity in adapting to the fibril topology to adopt a clustered organization that is possibly involved in adhesion to the fibrils.

## Caveolae are required for invadosome formation and collagenolytic activity

We investigated the contribution of caveolae to invadosome formation and activity. We individually knocked down (KD) *CAV1*, *CAVIN1* and *EHD2* by treatment with short interfering RNA (siRNA). We found a significant reduction in invadosome formation and collagen cleavage capacity after treatment; the effects were almost as strong as after depleting the key invadosome components *SH3PXD2A* (encoding TKS5) or *MMP14* (encoding MT1-MMP) (Fig. 3a–d and Extended Data Fig. 4a–c). Moreover, we found that fibrils were significantly more

**Fig. 4 | Invadosome-dependent control of caveola association with matrix fibrils. a**, MDA-MB-231 cells knocked down for *SH3PXD2A* were cultured on a layer of type I collagen fibrils and stained for CAV1 and cortactin (left). CAV1 distribution is shown with an inverted grey LUT (middle). The dotted line shows the cell contour. Insets: magnified views of the boxed regions (right). **b**, The ratio (as a percentage) of the area of cell-underlying fibrils with overlapping CAV1 signal to the total surface of cell-underlying fibrils was analysed in the depleted cell populations (CAV1$_f$ area). Data are mean ± s.d. $n = 163$ (siNT), $n = 145$ (siMT1-MMP) and $n = 193$ (siTKS5) cells from four biologically independent experiments. Statistical analysis was performed using one-way ANOVA with Kruskal–Wallis multiple-comparison test; ****$P < 0.0001$. **c**, The size distribution of fibril-associated CAV1-positive structures in depleted cells. $n = 138$ (siNT), $n = 145$ (siMT1) and $n = 153$ (siTKS5) cells from four biologically independent experiments. Statistical analysis was performed using one-way ANOVA with Kruskal–Wallis multiple-comparison test; **$P = 0.0018$, ****$P < 0.0001$. **d**, The effect of the actin polymerization inhibitors CK-666 and CytoD on TKS5-positive invadosomes (left and middle) and type I collagen cleavage (right) normalized to control (DMSO-treated) cells. Data are mean ± s.d. Middle: $n = 113$ (DMSO), $n = 92$ (CK-666) and $n = 109$ (CytoD) cells from three biologically independent experiments. Right: $n = 50$ (DMSO), $n = 57$ (CK-666) and $n = 45$ (CytoD) cells from two biologically independent experiments. Statistical analysis was performed using one-way ANOVA with Kruskal–Wallis test; ****$P < 0.0001$. **e**, The effect of CK-666 and CytoD on fibril-associated CAV1 signal as in **b**. Insets: magnified views of the boxed regions. Data are mean ± s.d. $n = 105$ (DMSO), $n = 146$ (CK-666) and

$n = 142$ (CytoD) cells from three biologically independent experiments. Statistical analysis was performed using one-way ANOVA with Kruskal–Wallis test; ****$P < 0.0001$. **f**, The effect of CK-666 and CytoD on the collagen fibril orientation index in cell-associated regions of the collagen network determined as in Fig. 2b. Data are mean ± s.d. $n = 98$ (DMSO), $n = 103$ (CK-666) and $n = 104$ (CytoD) fields from three biologically independent experiments. Statistical analysis was performed using one-way ANOVA with Kruskal–Wallis test; ****$P < 0.0001$. **g**, Comparison of TKS5-positive invadosomes in cells plated on collagen for 30 or 60 min. Data are mean ± s.d. $n = 116$ (30 min) and $n = 100$ (60 min) cells from three biologically independent experiments. Statistical analysis was performed using unpaired two-tailed *t*-tests; ****$P < 0.0001$. **h**, Comparison of CAV1 and cortactin-positive structures in cells plated on collagen for 30 min or 60 min. Insets: magnified views of the boxed regions. **i,j**, The time effect on CAV1 association (**i**) and the size of fibril-associated CAV1-positive structures (**j**). For **i**, data are mean ± s.d. $n = 117$ (30 min) and n = 95 (60 min) cells from three biologically independent experiments. For **j**, $n = 117$ (30 min) and $n = 95$ (60 min) cells from three biologically independent experiments. Statistical analysis was performed using one-way ANOVA with Kruskal–Wallis test (**i**) and unpaired two-tailed *t*-tests (**j**); ****$P < 0.0001$. For the box plots in **c** and **j**, the centre line shows the mean, the box limits show the first (Q1, 25th percentiles) to third (Q3, 75th percentiles) quartiles of the distribution, and the whiskers show the minimum to maximum values. Source numerical data are provided as Source Data. Scale bars, 10 μm (**a,d,e** (main image), **g** and **h** (main image)) and 5 μm (**e** and **h** (insets)).

co-oriented underneath *SH3PXD2A*-KD cells compared with control cells (Fig. 3e). Similarly, *CAV1* KD tendentially increased collagen fibril co-orientation in the network (Fig. 3e). Together, these results show that invadosome disruption (by *SH3PXD2A* or *CAV1* KD) consequently impacts the ability of tumour cells to remodel the environing collagen network.

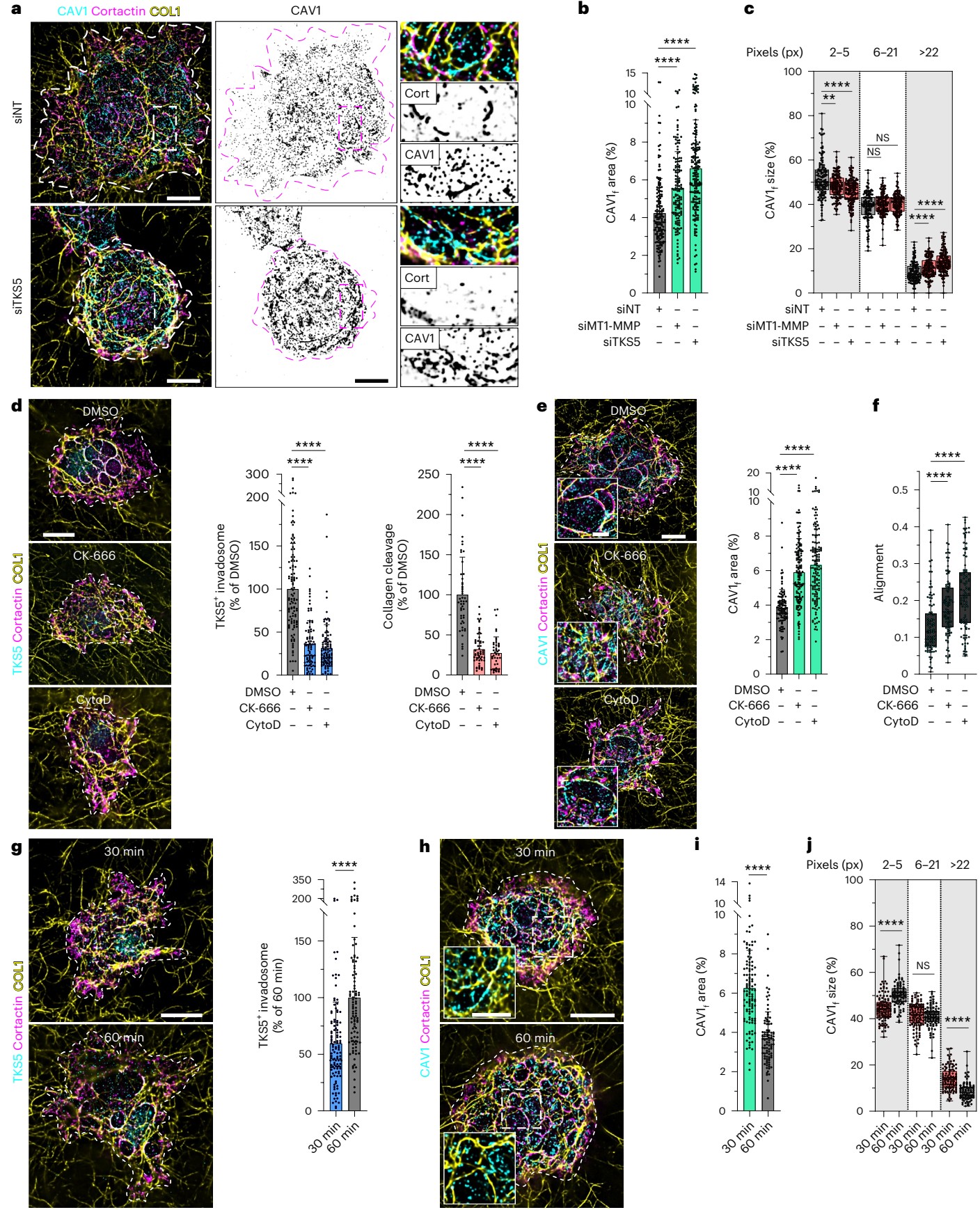

Consistent with previous reports[48], downmodulation of CAVIN1 expression also affected CAV1 levels (Extended Data Fig. 4c), raising the possibility that the effect of *CAVIN1* KD could be indirectly mediated by reduced CAV1 expression rather than by the loss of caveolae per se. To rule out this possibility, we analysed the CAV1 distribution in cells treated with siRNA against *CAVIN1* using a metric that quantifies the ratio of the fibril surface underneath the ventral PM with overlapping CAV1 signal to the total surface of cell-associated fibrils (CAV1$_f$ area; Methods and Extended Data Fig. 4d). In agreement with immunoblotting analysis (Extended Data Fig. 4c), *CAVIN1* silencing depleted the overall CAV1 signal at the PM (Fig. 3f) and did not affect the association of remaining CAV1 with the collagen fibrils (Fig. 3f,g), suggesting a preferential association of residual CAV1 with the contacted fibrils. Analysis of the size distribution of fibril-associated CAV1-positive structures (CAV1$_f$ size) in *CAVIN1*-depleted cells revealed an increase in small structures (2–5 px)–possibly CAV1 oligomers or scaffolds–and a concomitant reduction in the larger above-described caveolar clusters (>22 px) (Fig. 3f,h). Together, modulations of these two metrics by *CAVIN1* depletion indicate that clustered caveolar structures, and not CAV1 oligomers and scaffolds, are required for invadosome formation and function.

## Invadosome loss impacts caveola association with collagen fibrils

We next investigated a reciprocal control of caveola recruitment to the matrix fibrils by invadosomes. After *MMP14* or *SH3PXD2A* KD, we observed an expected strong decrease in invadosome formation and activity in MDA-MB-231 cells[14] (Fig. 3a–d and Extended Data Fig. 4a–c). Notably, we found that *MMP14* or *SH3PXD2A* silencing significantly increased CAV1 association with collagen fibrils and promoted the formation of larger matrix-associated CAV1 clusters (Fig. 4a–c). Invadosome (TKS5) disruption similarly correlated with CAVIN1 accumulation along collagen fibrils (Extended Data Fig. 5a). CAV1 and CAVIN1 total levels were not affected by *MMP14* or *SH3PXD2A* silencing (Extended Data Fig. 4c). To further validate the impact of invadosome repression on caveola association with collagen fibrils, we used pharmacological inhibitors of actin polymerization, CK-666 and cytochalasin D (CytoD), which inhibit the ARP2/3 complex and barbed-end actin filament capping, respectively[49,50], with a well-established detrimental effect on invadosome formation[14]. Inverse reduced invadosome formation and activity and enhanced CAV1 association with fibrils were observed in CK-666- and CytoD-treated cells (Fig. 4d,e and Extended Data Fig. 5b). Correlated with reduced invadosomal activity and increased CAV1 recruitment along the fibrils, collagen alignment increased after CK-666

and CytoD treatment (Fig. 4f). Moreover, the ARP2/3 complex activator N-WASP, which is necessary for invadosomal actin polymerization[51], was required for invadosome formation and enhanced CAV1 association with collagen fibrils in larger structures (Extended Data Fig. 5c–g).

We could further generalize these observations by investigating HT-1080 fibrosarcoma cells, which form TKS5- and cortactin-positive functional invadosomes when cultured on type I collagen fibrils[15]. CAV1 localized along collagen fibrils in an alternating pattern with invadosomes in HT-1080 cells (Extended Data Fig. 5h (arrowhead)). Furthermore, invadosome downmodulation after *SH3PXD2A* or *WASL* (encoding N-WASP) KD, and treatment with CK-666 or CytoD, correlated with increased CAV1 association along the collagen fibrils (Extended Data Fig. 5i–l). Taken together, these findings highlight a general mechanism whereby invadosome formation and activity are linked to caveola association with contacting collagen fibrils.

## Matrix fibril association of caveolae precedes invadosome formation during ECM adhesion

On the basis of our data establishing an alternating organization and reciprocal regulation of caveolae and invadosomes at contact sites between invasive cancer cells and the matrix, we hypothesized that the recruitment of caveolae and invadosomes along fibrils may rely on a sequential, dynamic mechanism in which fibril association of caveolae precedes invadosome formation. The dynamics of caveola and invadosome recruitment were analysed at earlier timepoints after cell adhesion to the matrix (that is, cells seeded for 30 min on the collagen network instead of 60 min). Invadosome formation increased between 30 and 60 min, while CAV1 association to the fibrils inversely decreased over time and CAV1-positive structures became smaller (Fig. 4g–j and Extended Data Fig. 5m), along with conversion of straight fibrils into curved ones (Figs. 2b and 4g,h). The fibril-associated CAVIN1 signal similarly decreased over time (Extended Data Fig. 5n). Together, these findings support a dynamic association of caveolae along the collagen fibrils that precedes and is required for invadosome formation and activity during ECM attachment of cancer cells.

## Topological and compositional cues drive CAV1 association with collagen fibrils

Our data suggested a correlation between the association of caveolar and invadosomal structures and tangential curvature of the collagen fibrils. Furthermore, PREM analysis revealed the association of elongated caveolar clusters recruited along the indented PM in contact with the underlying fibrils. Furthermore, membrane bending and membrane tension have been proposed to control caveola

**Fig. 5 | Differential contribution of topographic and composition cues to CAV1 and TKS5 membrane recruitment. a**, Left, fluorescence image of the gelatin–OG488-coated nanobar array (inverted grey LUT) with the nanobar dimensions. Right, schematics of the PM in contact with the fibril or the different nanobar arrays. **b**, MDA-MB-231 cells were cultured for 4 h on nanobar arrays coated (+ECM) or not (−ECM) with gelatin as a matrix, fixed and stained for CAV1 (top row) and TKS5 (middle row) in inverted grey LUT. Images are the projection of three optical planes at the bottom and at the top of the nanobars taken from image *z* stacks. The dotted lines show the nanobar position. Bottom, magnified views of the boxed regions. Scale bars, 10 μm (top and middle) and 5 μm (bottom). **c,d**, Top/bottom fluorescence intensity ratio of CAV1 (**c**) and TKS5 (**d**) signals as a measure of the effect of nanobar-imposed membrane curvature and gelatin coating. For **c**, *n* = 148 (−ECM) and *n* = 136 (+ECM) cells from three biologically independent experiments. For **d**, *n* = 146 (−ECM) and *n* = 154 (+ECM) cells from three biologically independent experiments. For **c** and **d**, statistical analysis was performed using unpaired two-tailed *t*-tests; ****P < 0.0001. For the box plots in **c** and **d**, the centre line shows the mean, the box limits show the first (Q1, 25th percentiles) to third (Q3, 75th percentiles) quartiles of the distribution, and the whiskers show the minimum to maximum values. **e**, MDA-MB-231 cells cultured on a fibrous type I collagen layer were stained for β1 integrin and CAV1.

The dotted line shows the cell contour. The distribution of CAV1 and β1 integrin is shown with an inverted grey LUT. Inset: magnified views of the boxed regions. Signal intensity profiles along the fibril are plotted in the right row showing CAV1 and β1 integrin co-localization along collagen fibrils. **f**, MDA-MB-231 cells silenced for β1 or β3 integrin as indicated were cultured on a layer of type I collagen fibrils for 60 min and stained for CAV1. The dotted line shows the cell contour. Inset: magnified views of the boxed regions. The effect of integrin KD on CAV1 association with cell-associated fibrils was quantified (the ratio of the area of cell-underlying fibrils with overlapping CAV1 signal to the total surface of cell-underlying fibrils). Data are mean ± s.d. *n* = 106 (siNT), *n* = 133 (siβ1 integrin), *n* = 112 (siβ3 integrin) and *n* = 96 (siβ1/β3 integrin) cells from three biologically independent experiments. Statistical analysis was performed using one-way ANOVA with Kruskal–Wallis test; ****P < 0.0001. **g**, The effect of *SH3PXD2A* (siTKS5) and integrin (siβ1/β3 integrin) KD on CAV1 association with the cell-associated fibrils. Data are mean ± s.d. *n* = 132 (siNT), *n* = 124 (siTKS5) and *n* = 107 (siTKS5 + siβ1/β3 integrin) cells from four biologically independent experiments. Statistical analysis was performed using one-way ANOVA with Kruskal–Wallis test; ****P < 0.0001. Source numerical data are provided as Source Data. Scale bars, 10 μm (**e** (left) and **f** (top)) and 5 μm (**e** (right) and **f** (bottom)).

recruitment and/or organization at the PM[26,32,33]. PREM analysis of the collagen network revealed an estimated diameter of individual fibrils of around 75–100 nm, corresponding to a minimal dimension as fibrils frequently reorganized as thicker bundles (Extended Data

Fig. 6a–c). To assess whether local membrane deformation induced by collagen fibrils might contribute to CAV1/caveola recruitment, we used nanotopography-imposed membrane curvature based on nanobars similar in size to the fibril dimension (that is, 300 nm width × 900 nm

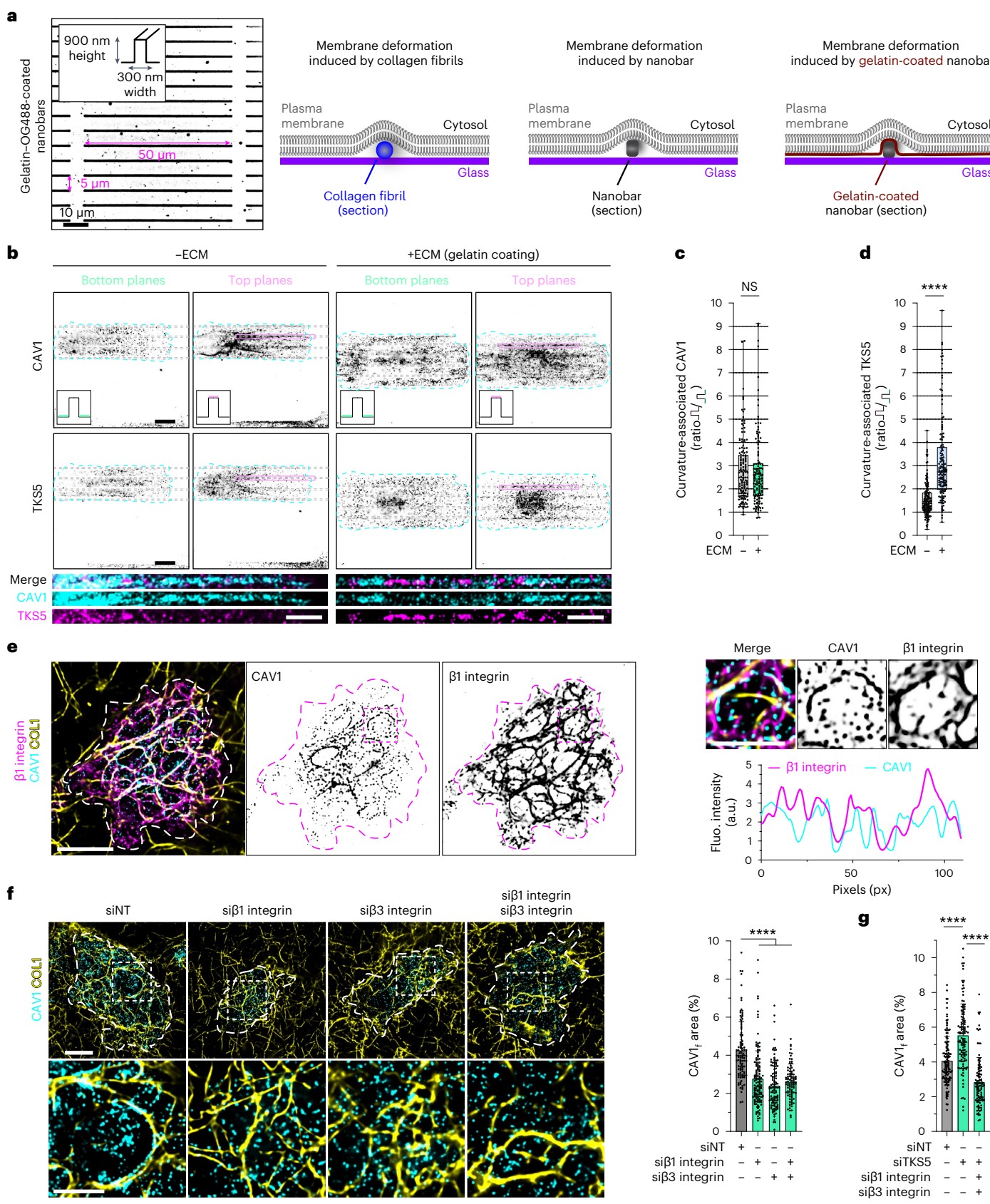

height), therefore mimicking the collagen network topology[33] (Fig. 5a). MDA-MB-231 cells were plated onto nanobar arrays coated with poly-L-lysine (−ECM) or gelatin (+ECM) and stained for CAV1 and TKS5. We observed an approximately 2.5-fold CAV1 enrichment at sites of nanobar-induced PM deformation, irrespective of matrix coating (Fig. 5b,c), indicating that CAV1 recruitment to these regions of membrane deformation was ECM independent. By contrast, TKS5 recruitment to PM deformations was maximal (around threefold) at gelatin-coated nanobars (versus around 1.5-fold enrichment under the −ECM conditions; Fig. 5b,d). Furthermore, reminiscent of the type I collagen fibril substratum (Fig. 1c), we confirmed that CAV1- and TKS5-positive accumulations were mutually exclusive on gelatin-coated nanobars (Fig. 5b (insets)). Together, these results showed that inward PM curvature is sufficient to trigger CAV1 recruitment in this simplified construct.

### Invadosome-based collagenolytic activity controls caveola-mediated β1 integrin and ECM internalization

Sensing of ECM mechanical properties by caveolae involves integrin adhesion receptors, which also interplay with MT1-MMP and with matrix remodelling[17,18,34,52]. We therefore investigated the contribution of the adhesion receptors integrins β1 and β3 to caveola recruitment at PM–collagen fibril contact sites. Notably, we observed that CAV1 co-localized with collagen-binding β1 integrin along the contacted collagen fibrils, both in MDA-MB-231 and HT-1080 cells (Fig. 5e and Extended Data Fig. 6d). CAVIN1 similarly showed a close association with β1 integrin (Extended Data Fig. 6e). To investigate the contribution of β1 and β3 integrins to CAV1 recruitment along the fibrils, these integrin receptors were downmodulated resulting in a marked decrease in CAV1 association with the fibrils (Fig. 5f and Extended Data Fig. 6f). Furthermore, increased CAV1 association with collagen fibrils observed in the absence of TKS5 (Fig. 4b) could be repressed by KD of β1 or β3 integrin (Fig. 5g). Taken together, these results indicate that CAV1/caveola recruitment along collagen fibrils is both topologically and compositionally controlled and results from the combined actions of inward PM deformation and integrin engagement.

To further characterize the consequences of CAV1/β1 integrin association along collagen fibrils, we investigated whether and how downmodulation of caveolae could impact β1 integrin localization and internalization. *CAV1* or *CAVIN1* KD increased the β1 integrin signal at the PM (Fig. 6a), suggesting a role for caveolae in β1 integrin surface clearance. Besides their association with the PM, we found that CAV1 and CAVIN1 were also present as discrete puncta associated with intracellular MT1-MMP- and cortactin-positive vesicles, previously identified as endolysosomes[41] (Extended Data Fig. 1b,c (arrowheads) and Extended Data Fig. 7a–c). Immunogold PREM analysis revealed the presence of CAV1–GFP- and CAVIN1–GFP-enriched buds on the cytosolic face of micrometre-size vesicles reminiscent of endolysosomes (Fig. 6b,c). Conspicuously, CAV1-positive endolysosomes were also positive for β1 integrin (Fig. 6d), suggesting that CAV1 may contribute to β1 integrin/ECM internalization in MDA-MB-231 cells. Consistent with this hypothesis, we observed CAV1-positive puncta detaching and moving away from PM–collagen fibril contact sites in proximity with invadosomes (Fig. 6e and Supplementary Video 3). To analyse the contribution of CAV1 to β1 integrin internalization, β1 integrin was surface-labelled, and we followed the internalization of anti-β1-integrin antibodies. After 60 min of internalization, the presence of anti-β1-integrin antibodies was detected in RAB5-, RAB7- and CAV1-positive endocytic vesicles (Fig. 6f and Extended Data Fig. 7d–f). Loss of CAV1 had a detrimental effect on the internalization of β1 integrin, as observed by the decrease in β1 integrin-positive vesicles (Fig. 6f,g). Together, these results further confirm a central role of caveolae in β1 integrin internalization[53].

We noticed that invadosome loss after *SH3PXD2A* silencing also impaired β1 integrin internalization and correlated with an increased β1 integrin signal at the PM (Fig. 6g,h), indicating that invadosome activity is required for surface β1 integrin clearance. To better characterize the involvement of invadosomes in β1 integrin internalization and to discriminate between invadosome presence and function, cells were treated with the broad-spectrum MMP inhibitor GM6001, which abolishes the collagenolytic activity without affecting the formation of invadosomes[14]. We found that GM6001-treated cells had increased CAV1 association with the collagen fibrils and increased β1 integrin signal at the PM, along with decreased β1 integrin internalization (Extended Data Fig. 7g–j). Together, these results show that invadosomes control caveola-dependent β1 integrin internalization through a mechanism relying on the proteolytic cleavage of collagen fibrils through the invadosome–MT1-MMP axis.

To validate further the involvement of invadosome function in caveola-dependent β1 integrin internalization, we used MDA-MB-231 cells overexpressing TKS5 (MDA-MB-231[TKS5]) that show enhanced invadosome formation and activity[15] (Extended Data Fig. 8a,b). In this context, we observed a decrease in CAV1 association with collagen fibrils and enhancement of intracellular CAV1 endolysosomal localization (Fig. 7a–d). CAVIN1 was found to be similarly associated with CD63-positive endolysosomes in MDA-MB-231[TKS5–GFP] cells

---

**Fig. 6 | Caveolae mediate β1 integrin internalization depending on invadosome collagenolytic activity. a**, MDA-MB-231 cells knocked down for *CAV1* or *CAVIN1* were cultured on a fibrous type I collagen layer for 60 min and stained for β1 integrin. The images show the β1 integrin signal in a projection of three optical planes in the ventral PM using an inverted grey LUT. The dotted line shows the cell contour. The β1 integrin signal in the PM plane in *CAV1*- or *CAVIN1*-depleted cells was normalized to the signal in control siNT-treated cells. Data are mean ± s.d. *n* = 90 (siNT), *n* = 83 (siCAV1) and *n* = 67 (siCAVIN1) cells from three biologically independent experiments. Statistical analysis was performed using one-way ANOVA with Kruskal–Wallis test; ****$P < 0.0001$. **b**,**c**, Anti-GFP immunogold PREM of unroofed MDA-MB-231 cells expressing CAV1–GFP (**b**) or CAVIN1–GFP (**c**). Insets: magnified views of the boxed regions to show CAV1- and CAVIN1-positive structures on the cytoplasmic surface of micrometre-size vesicles reminiscent of endolysosomes. Immunogold beads are pseudocoloured in yellow. **d**, Projection of three internal optical sections to show β1 integrin and CAV1 association with cytoplasmic vesicles. The dotted line shows the cell contour. Insets: magnified views of the boxed region. **e**, Galleries of selected time frames (in minutes) from time-lapse sequences of MDA-MB-231 cells expressing CAV1–GFP (cyan) and DsRed–cortactin (magenta) plated on a layer of Cy5-labelled collagen I fibrils (yellow). The arrowheads point to fibril-associated CAV1-positive puncta near cortactin-positive invadosomes detaching and moving away from the collagen fibril. **f**, Surface-exposed β1 integrins were labelled with antibodies added to non-permeabilized cells. After 60 min at 37 °C, the presence of internalized anti-β1-integrin antibodies was detected and the number of β1-positive vesicles per cell was plotted. The images in the bottom row show the signal of internalized anti-β1-integrin antibodies using an inverted grey LUT. The dotted line shows the cell contour. **g**, Quantification of the number of integrin-β1-positive vesicles per cell in the conditions shown in **f**. For the box plots, the centre line shows the mean, the box limits show the first (Q1, 25th percentiles) to third (Q3, 75th percentiles) quartiles of the distribution, and the whiskers show the minimum to maximum values. *n* = 148 (siNT), *n* = 154 (siCAV1) and *n* = 167 (siTKS5) cells from three biologically independent experiments. Statistical analysis was performed using one-way ANOVA with Kruskal–Wallis test; ****$P < 0.0001$. **h**, The β1 integrin signal at the ventral PM in *MMP14*-depleted (siMT1-MMP) or *SH3PXD2A*-depleted cells was plotted and normalized to the signal in control siNT-treated cells. Data are mean ± s.d. *n* = 157 (siNT), *n* = 132 (siMT1-MMP) and *n* = 190 (siTKS5) cells from five biologically independent experiments. Statistical analysis was performed using one-way ANOVA using Kruskal–Wallis test; ****$P < 0.0001$. Source numerical data are provided as Source Data. Scale bars, 10 µm (**a**,**d** and **f**), 2 µm (**e**), 200 nm (**b** and **c** (main images)) and 100 nm (**b** and **c** (insets)).

(Extended Data Fig. 8c). Increased endolysosomal CAV1 was strongly associated with intracellular β1 integrin (Fig. 7e) that was present in RAB7-positive endolysosomes (Extended Data Fig. 8d). Moreover, TKS5 overexpression increased the internalization of surface-bound anti-β1-integrin antibodies (Fig. 7f). Analysis of TKS5–GFP-positive invadosomes showed reduced β1 integrin association with these structures (Extended Data Fig. 8d,e), confirming the clearance of β1 integrin from the cell surface. Notably, *CAV1* KD strongly counteracted β1

integrin internalization in MDA-MB-231$^{TKS5–GFP}$ cells (Fig. 7f). Finally, when MDA-MB-231$^{TKS5–GFP}$ cells were cultured for 2 h on top of fluorescently labelled collagen fibrils, type I collagen was visible in CAV1-, β1-integrin- and CD63-positive endolysosomes (Extended Data Fig. 8f), suggesting that ECM internalization occurs in relation to CAV1-mediated β1 integrin internalization. To directly assess the potential contribution of caveolae to ECM uptake, MDA-MB-231$^{TKS5–GFP}$ cells were cultured on a network of fluorescently labelled fibrils and, after

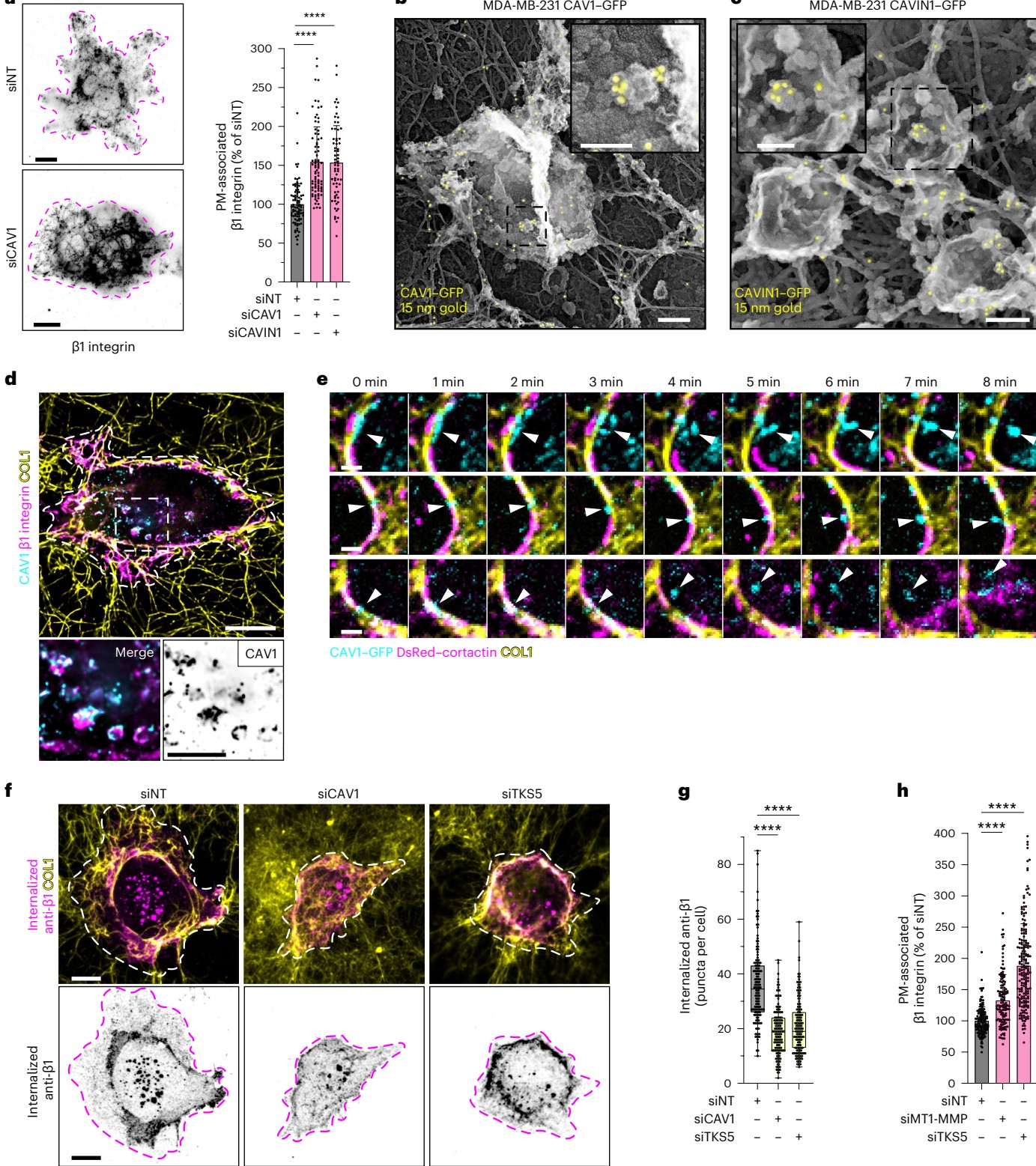

incubation, fibrils outside the cells were stained with anti-collagen-I antibodies added to non-fixed, non-permeabilized cells to exclude intracellular fibril labelling (Fig. 7g,h). Caveola disruption after *CAV1* or *CAVIN1* KD was associated with a significant decrease in collagen uptake, similar to MT1-MMP loss (Fig. 7g–i). Together, these results indicate that increased invadosome formation and the associated collagenolytic activity promote caveola-dependent β1 integrin and ECM internalization.

### Caveolae are required for 3D invasion of breast cancer cells

To investigate the consequences of caveola/invadosome interplay during invasion through a confining environment, MDA-MB-231 cells were embedded in a dense, 3D collagen matrix. Like in 2D conditions, caveolae (CAV1) and invadosomes (cortactin) localized along constricting collagen fibrils (that is, mostly ahead of the bulky nucleus[42,54]) in an alternating, mutually excluded manner (Fig. 8a–c and Extended Data Fig. 9a). We next investigated the consequence of *CAV1* silencing on the ability of breast cancer cell spheroids to invade a 3D collagen matrix environment. As compared to multicellular spheroids of MDA-MB-231 cells treated with control siRNA (siNT), invasion by *CAV1*-depleted spheroids over a 48 h period was decreased by about 40%, similar to the effect of invadosome disruption by *MMP14* or *SH3PXD2A* KD (Fig. 8d,e and Extended Data Fig. 9b). Together, we propose a mechanism whereby caveolae associate with constraining collagen fibrils and favour the formation of adjacent invadosomes. In turn, invadosome-mediated collagen cleavage by MT1-MMP weakens the resistance of the fibrils and leads to β1 integrin (and ECM) uptake through caveolae, enabling cells to dynamically release their contacts with the ECM, ultimately facilitating tumour cell invasion through the dense environment (see the discussion and model in Fig. 8f).

## Discussion

The fibrous type I collagen network consists of crisscrossed fibril bundles with empty space in between that tumour cells use as pores to facilitate their dissemination[10,42,54]. Furthermore, invasive cancer cells have a considerable potential to remodel the matrix due to the dual activity of invadosomes, which can soften the confining fibrils based on MT1-MMP collagenolytic activity as well as applying outward-pointing forces to these fibrils, powered by actin filament polymerization, as shown here and previously[14,40]. Consequently, invadosome-targeted fibrils are bent and pushed aside to clearing up a migratory path and facilitate cell (and nucleus) passage, as shown

here and previously[14,42]. The role of caveolae in tumour progression is more complex and controversial[29,31]. CAV1 expression in the stromal compartment (in carcinoma-associated fibroblasts) can promote metastasis in breast cancer, and caveola upregulation in tumour cells is associated with increased invasion and poor prognosis in several types of cancers[35,36]. Here we reveal that, at contact sites between the PM and constricting fibrils, recruitment of caveolae precedes invadosome formation and function, which is observed in the vicinity of caveolae giving rise to a marked alternation of the two structures (see the model in Fig. 8f).

On close inspection of the fibrils, we found that, although regions in direct contact with invadosomes were curved, based on MT1-MMP collagenolytic activity[14], fibril portions in association with caveolae were straighter. These observations suggest that caveola recruitment—which is positively influenced by inward membrane curvature, as shown here and previously[26,33]—preferentially occurs at sites of contact with restricting, rigid matrix fibrils that impose local PM deformations. Although our data indicate that invadosome recruitment is under the influence of matrix compositional cues, some role for PM curvature cannot be excluded. Indeed, invadosome formation requires activation of the CDC42 signalling module leading to N-WASP, ARP2/3 complex and cortactin-dependent branched actin assembly[15,51,55,56]. Recent studies indicate that components of the WASP(N-WASP) module are intrinsically sensitive to compressive load and PM deformation[57–59].

High-resolution morphological analysis revealed that caveolae are recruited as individual pits and larger clusters of individual caveolae along collagen fibrils, possibly grabbing the confining fibrils. The accumulation of integrin adhesion receptors at these sites supports an adhesive function for these caveola-rich fibril-contact sites. Possibly, the accumulation of actin filaments in nearby-forming invadosomes prevents these caveolar structures from extending further. Reciprocally, the specific lipid composition of the caveolar PM may prevent intermixing with invadosomes. Moreover, segmentation of the fibrils in compliant (invadosome-associated) regions and more rigid (caveola-associated) portions may further contribute to the segregation of invadosomes and caveolar structures along the confining fibrils based on specific invadosome and caveola mechanosensing properties[26,27,33,60]. Our data therefore indicate that, in relation to their alternated distribution, caveolae and invadosomes influence each other's organization and function supporting a mechanistic paradigm for pericellular matrix remodelling during tumour cell invasion (Fig. 8f).

---

**Fig. 7 | CAV1 is required for the invadosome-mediated invasion program of breast cancer cells in 3D collagen. a**,**b**, Immunofluorescence images of MDA-MB-231 cells overexpressing TKS5–GFP cultured on fibrous type I collagen for 60 min and stained for CAV1 (**a**) or the endolysosomal marker CD63 (**b**) as indicated. MDA-MB-231 cells were used as a control. The dotted line shows the cell contour. Insets: magnified views of the boxed regions. **c**, The effect of TKS5 overexpression on CAV1 association with cell-associated fibrils was quantified (CAV1$_f$ area). Data are mean ± s.d. $n$ = 114 (MDA-MB-231) and $n$ = 130 (MDA-MB-231$^{TKS5–GFP}$) cells from three biologically independent experiments. Statistical analysis was performed using unpaired two-tailed $t$-tests; ****$P$ < 0.0001. **d**, Quantification of the CAV1 signal on CD63-positive endolysosomes. $n$ = 94 (MDA-MB-231) and $n$ = 116 (MDA-MB-231$^{TKS5–GFP}$) cells from three biologically independent experiments. Statistical analysis was performed using unpaired two-tailed $t$-tests; ***$P$ = 0.0003. **e**, Immunofluorescence images of TKS5–GFP-overexpressing MDA-MB-231 cells on type I collagen stained for β1 integrin, CAV1 and TKS5–GFP. The images are projections of three optical planes at the level of the ventral PM (left) and inside the cell (right). The dotted line shows the cell contour. Insets: magnified views of the boxed region showing CAV1 and β1 integrin in an inverted grey LUT. **f**, After surface labelling and internalization, the number of anti-integrin-β1-antibody-positive vesicles per cell was compared in the indicated cell populations (as in Fig. 6f). $n$ = 148 (siNT/MDA-MB-231), $n$ = 152 (siNT, MDA-MB-231$^{TKS5–GFP}$) and $n$ = 149 (siCAV1,

MDA-MB-231$^{TKS5–GFP}$) cells from three biologically independent experiments. Statistical analysis was performed using one-way ANOVA using Kruskal–Wallis test; ****$P$ < 0.0001. **g**, Immunofluorescence images of unfixed, unpermeabilized MDA-MB-231 cells overexpressing TKS5–GFP cultured on fibrous Cy5-conjugated type I collagen (magenta) for 2 h and stained for collagen I (green) to discriminate between internalized and extracellular collagen (left). Middle, Cy5-conjugated (extracellular + intracellular) collagen using an inverted grey LUT. Right, extracellular collagen I fibrils stained with anti-collagen-I antibodies in unfixed, nonpermeabilized conditions and shown using an inverted grey LUT. The dotted line shows the cell contour. The arrowheads indicate internalized, Cy5-conjugated type I collagen. **h**, The analysis pipeline of internalized versus extracellular type I collagen. **i**, Quantification of the percentage of internalized collagen I (normalized to cell area) in cells that were treated with the indicated siRNAs. $n$ = 75 (siNT), $n$ = 82 (siMT1-MMP), $n$ = 100 (siCAV1) and $n$ = 85 (siCAVIN1) cells from three biologically independent experiments. Statistical analysis was performed using one-way ANOVA with Kruskal–Wallis test; **$P$ = 0.0016, ****$P$ < 0.0001. For the box plots in **d**,**f** and **i**, the centre line shows the mean, the box limits show the first (Q1, 25th percentiles) to third (Q3, 75th percentiles) quartiles of the distribution, and the whiskers show the minimum to maximum values. Source numerical data are provided as Source Data. Scale bars, 10 μm (**a**,**b**,**e** (main images) and **g** (rows 1 and 3) and 5 μm (**a**,**b**,**e** (insets) and **g** (rows 2 and 4)).

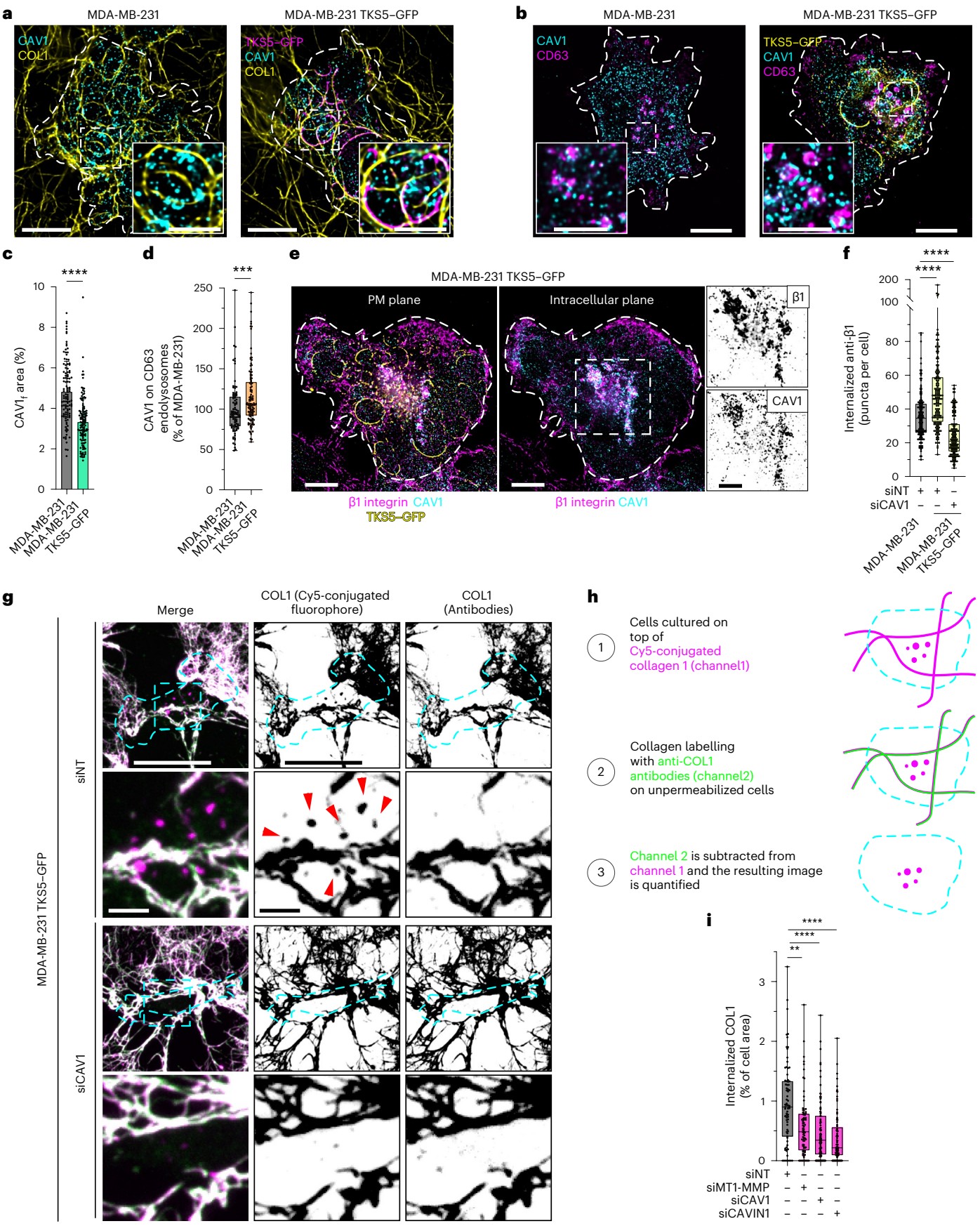

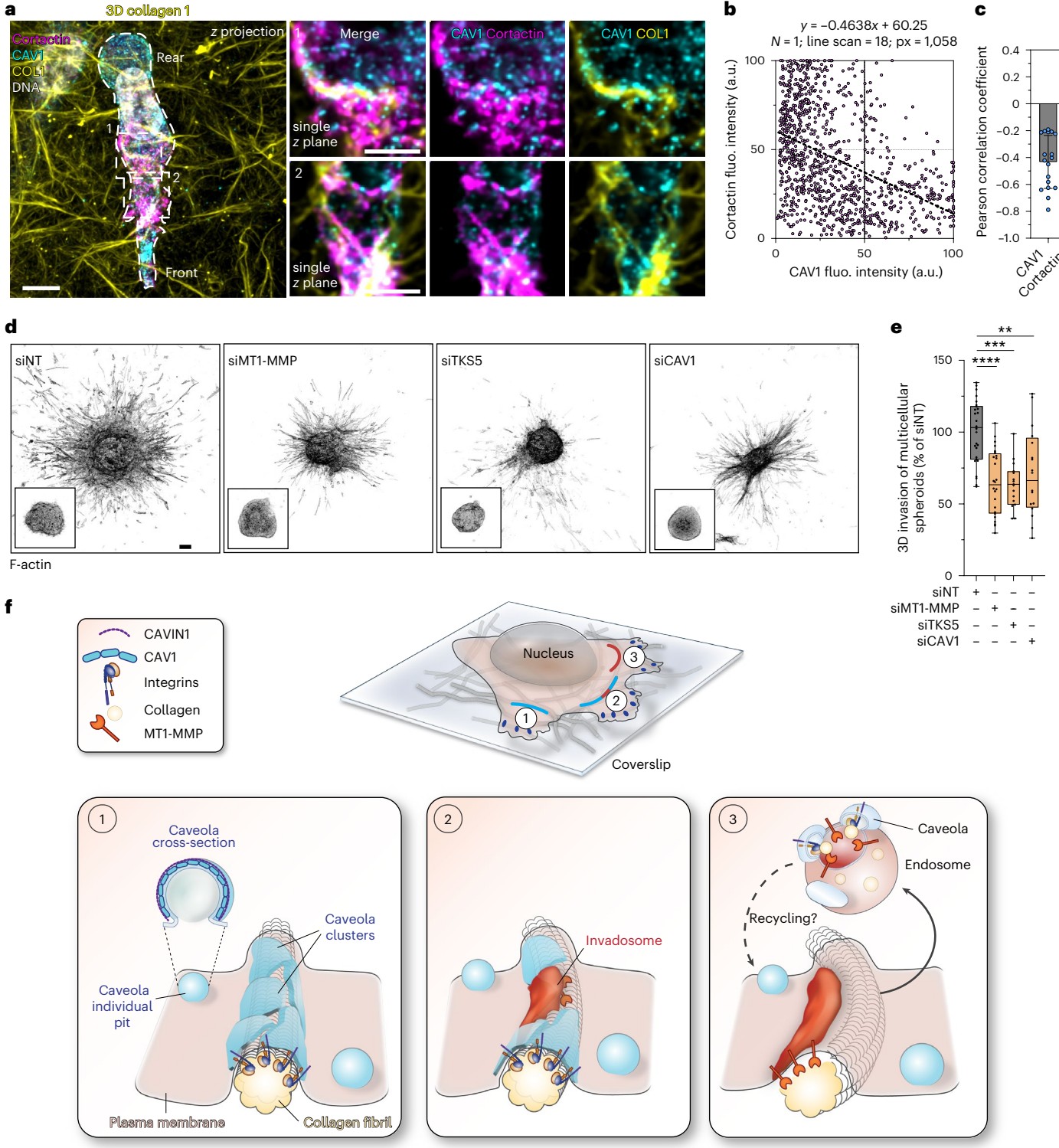

In addition to established roles of caveolae in integrin signalling and regulation of focal adhesion formation and dynamics[34,44], our data uncover a localization and function for fibril-associated caveolar structures as adhesive snap buttons that promote nearby invadosome formation and control ECM remodelling during tumour cell dissemination. Reciprocally, interfering with invadosome function leads to the accumulation and stabilization of caveolar clusters and integrins associated with straight (probably stiff) confining matrix fibrils, a situation that precludes invasion. Furthermore, our data support a mechanosensing mechanism whereby, due to proteolytic cleavage,

softening of the matrix by invadosomes and, possibly, a decrease in membrane tension, collagen integrins are disengaged from the matrix fibrils and are cleared from the cell surface together with their caveolar carriers and traffic to the endolysosomal system[26]. We also observed that ECM fragments are co-internalized with integrin receptors in a caveola-dependent manner. Moreover, phosphorylation of several Ser/Thr residues in the cytoplasmic tail of MT1-MMP and trafficking of the protease are modulated through interactions with β1 integrins[18], suggesting common routes for the clearance of these membrane proteins, their possible recycling to new sites of contact and remodelling

**Fig. 8 | CAV1 is required for the invadosome-mediated invasion program of breast cancer cells in 3D collagen. a**, MDA-MB-231 cells embedded in a 3D type I collagen gel (yellow) were stained for CAV1 (cyan) and cortactin (magenta). Inset: single *z* plane of a magnified view of the indicated boxed region. **b**, Point cloud distribution of pixel fluorescence intensity for the caveolae (CAV1) and invadosome (cortactin) markers indicating negative correlation. The dots represent 1,058 *xy* pairs (that is, pixels) of 18 line scans from one experiment. **c**, The Pearson correlation coefficient for individual line-scan profiling of CAV1/cortactin signal intensities shown in **b**. *n* = 18 line scans from one experiment. **d,e**, MDA-MB-231 cells were treated with the indicated siRNAs and multicellular spheroids were embedded in type I collagen. Spheroids were fixed and stained for F-actin after 48 h (**d**) and 3D invasion was quantified (**e**) (Methods). Insets: magnified views of multicellular spheroids at the time of embedding in the collagen network. For the box plots, the centre line shows the mean, the box limits show the first (Q1, 25th percentiles) to third (Q3, 75th percentiles) quartiles of the distribution, and the whiskers show the minimum to maximum values. *n* = 26 (siNT), *n* = 24 (siMT1-MMP), *n* = 15 (siTKS5) and *n* = 16 (siCAV1) spheroids from three biologically independent experiments. Statistical analysis was performed using one-way ANOVA using Kruskal–Wallis test; **$P$ = 0.0055, ***$P$ = 0.0001, ****$P$ < 0.0001. Source numerical data are provided as Source Data. **f**, Model of mechanosensitive caveola–invadosome interplay in matrix remodelling. Caveolae are present as invaginated pits at the PM[19,20]. When cultured on top of a meshwork of type I collagen fibrils (top schematic), cancer cells have elongated clusters of CAV1/CAVIN1-positive caveolar structures along PM–collagen fibrils contact sites, in a process that is dependent on membrane curvature, collagen-binding integrins (that is β1 integrin) and matrix stiffness. Invadosomes form in the vicinity of and alternate with caveolar clusters (1). After remodelling and weakening of the collagen fibrils through invadosomes action[14], caveolae mediate β1 integrin clearance from the cell surface, along with ECM internalization and transport to endolysosomes (2), from where some of these components could be recycled and delivered to newly formed PM–matrix contact sites as the cell moves through the matrix (3). Scale bars, 100 µm (**d**), 10 µm (**a** (left)) and 5 µm (**a** (right)).

of the matrix (Fig. 8f). Indeed, as cells move through the crowded stromal tissue and encounter new constraining fibrils, invadosomes (and caveolae) are engaged in a dynamic recycling process. Extensive work established that MT1-MMP can then be returned to the PM to form invadosomes[11,13,41]. Notably, early studies reported roles for caveolae in MT1-MMP internalization[61]. Our data show that there is little association of MT1-MMP with caveolae at PM–fibril contact sites, if any. Several reports pointed to a role for clathrin-coated pits in the clearance of surface MT1-MMP. Clathrin-coated pits are abundant in PM areas close to matrix contact sites and can contribute to cell adhesion to the fibrous collagen environment and to 3D migration[11,13,62–65] (Extended Data Fig. 3a). Here we highlight the potentiality of the invasive cancer program to repurpose major PM subdomains—that is, caveolae, invadosomes and clathrin-coated pits—to work in concert to support stromal matrix tissue remodelling and facilitate tumour invasion. Our findings also identify the caveola–invadosome axis as a potential therapeutic target to control tumour cell invasion.

## Online content

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

## Methods

### Cell culture, stable and transient transfection, and siRNA treatment

Human MDA-MB-231 breast adenocarcinoma cells obtained from ATCC (ATCC HTB-26) were grown in Leibovitz L-15 medium (Sigma-Aldrich) supplemented with 15% fetal calf serum and 2 mM glutamine (Thermo Fisher Scientific) at 37 °C under 1% $CO_2$. Human HT-1080 fibrosarcoma cells (ATCC CCL-121) were grown in Dulbecco's modified Eagle's GlutaMax medium (DMEM GlutaMAX; Thermo Fisher Scientific) supplemented with 10% fetal calf serum. Both cell lines were obtained from ATCC and were routinely tested for mycoplasma contamination. MDA-MB-231 cells stably expressing TKS5–GFP were generated by lentiviral transduction[14]. The cortactin-DsRed construct was provided by M. A. McNiven[66]. The CAV1-GFP construct was a gift from A. Helenius. The CAVIN1-GFP plasmid was a gift from A. Helenius (Addgene plasmid, 27709). For transient expression, MDA-MB-231 cells were transfected with the plasmid constructs using AMAXA nucleofection (Lonza) and analysed 48 h after transfection. For transient gene silencing, MDA-MB-231 or HT-1080 cells were treated with the indicated siRNA (50 nM) using Lullaby (OZ Biosciences) according to the manufacturer's specifications. A list of the siRNAs used for this study is provided in Supplementary Table 1. The siRNA/Lullaby mix was incubated for 30 min at room temperature and was added to the well, followed by addition of the cell suspension ($4 \times 10^4$ cells per well in a 24-well plate). Cells were analysed 72 h after transfection.

### Antibodies and drugs treatment

A list of the commercial antibodies used for this study is provided in Supplementary Table 2. A list of the drugs used in this study is provided in Supplementary Table 3. CK-666 (200 μM), CytoD (0.5 μM) or pan-MMP inhibitor GM6001 (40 μM) were added to the cells simultaneously to their seeding on collagen fibrils. Unless indicated otherwise, drug treatment was performed for 60 min. After treatment, cells were processed for immunoblotting or indirect immunofluorescence analysis as described below.

### Cell lysis and western blot analysis

After drug treatment or transient (72 h) gene silencing, cells were placed on ice, washed with ice-cold PBS and scraped using RIPA buffer (50 mM Tris HCl pH 8.0, 137 mM NaCl, 1% Triton X-100, 10 mM $MgCl_2$, 10% glycerol, protease inhibitor cocktail tablets (Roche) and phosphatase inhibitor cocktail 2 (Sigma-Aldrich)). The lysates were centrifuged at maximum speed (17,700g) for 30 min at 4 °C. The supernatant was collected in a fresh tube and the protein concentration was measured using the Pierce BCA protein assay kit. 4× Laemmli Sample Reducing buffer was added and the samples were heated for 10 min at 95 °C before analysis using SDS–PAGE on 4–12% Tris-glycine gels (Thermo Fisher Scientific). Proteins were then transferred onto a nitrocellulose membrane using the iBlot2 Dry Blotting System. The membrane was incubated for 2 h in blocking buffer (5% BSA or 5% skimmed milk in Tris-buffer saline (TBS, Interchim) pH 7.4 and 1% Tween-20; TBST), followed by incubation with primary antibodies at room temperature or incubation overnight at 4 °C. The membrane was then washed and incubated with HRP-conjugated secondary antibodies diluted in blocking buffer for 1 h at room temperature. After three washes with TBST, antibodies were detected using the Enhanced Chemiluminescence reagent (ECL, Amersham) using the ChemiDoc MP Imaging System (Bio-Rad).

### Preparation of fibrillar type I collagen

Preparation of the fluorescently labelled fibrillar type I collagen gel has been performed as described previously[67]. When required, drugs were added to the final concentration in the collagen polymerization mix (Supplementary Table 3). Polymerization of type I collagen at 2.2 mg ml$^{-1}$ was started by incubation at 37 °C in a humidified chamber ($CO_2$ cell incubator). After 150 s of polymerization, the collagen layer was washed gently in PBS, and a cell suspension ($5 \times 10^4$ cells) was added for 30–60 min at 37 °C in 1% $CO_2$.

### Indirect immunofluorescence analysis, 3D deconvolution microscopy, image deconvolution and image analysis

Cells were pre-extracted with 0.1% Triton X-100 in 4% PFA in PBS for 90 s, fixed in 4% PFA in PBS for 20 min and stained for immunofluorescence microscopy using the indicated antibodies (a list of which is provided in Supplementary Table 2). Coverslips were mounted on glass slides with Prolong-DAPI (Thermo Fisher Scientific) mounting medium. The images were acquired using a wide-field microscope (Eclipse 90i Upright; Nikon) using a ×100 Plan Apo VC 1.4 oil-immersion objective and a highly sensitive cooled interlined charge-coupled device camera (CoolSNAP HQ2; Roper Scientific). The system was controlled using Metamorph software (Molecular Devices by Gataca Systems). A z-dimension series of images was taken every 0.2 μm using a piezoelectric motor (Physik Instrumente), and the images were deconvoluted using the Nikon NIS-Elements software (3D-deconvolution module; Lucy-Richardson algorithm).

Quantification of TKS5-positive invadopodia and measurement of pericellular collagenolysis on a thin layer of type I collagen gel were detailed previously[64,67]. For quantification of CAV1 association with collagen fibrils, cells were seeded on top of a layer of Cy5-conjugated type I collagen fibrils for 60 min (unless indicated otherwise) and stained for CAV1. A z stack of three consecutives planes at the PM was made using Maximum Intensity Projection command in Fiji. A mask of the collagen fibrils was generated by applying the Substract background, Gaussian Blur and Unsharp mask commands in Fiji (depicted in Extended Data Fig. 4d). Only the collagen fibrils present underneath the cell were considered for quantification by manual drawing of the cell contour (Extended Data Fig. 4d). The CAV1 signal was thresholded and CAV1 association with collagen fibrils was determined as the percentage of cell-underlying collagen fibrils positive for CAV1 (CAV1$_f$ area). A homemade automated program for this quantification was generated in Fiji and is available on demand. The size distribution of the identified CAV1 structures was ranked and determined as the CAV1$_f$ size.

For measurement of global collagen fibril orientation, we used the OrientationJ Dominant Direction Plugin in Fiji[68]. The collagen channel was deconvoluted using Nikon NIS-Elements software (3D-deconvolution module; Lucy–Richardson algorithm) and a projection of around 20 consecutives planes (thickness, ~2 μm) was made from a z stack using the Maximum Intensity Projection command in Fiji. Regions of analysis were manually drawn underneath the cell (cell-associated area) and outside the cell (cell-free area; Fig. 2a and Extended Data Fig. 2a). Values were obtained by running the OrientationJ Dominant Direction plugin on these defined regions.

For co-localization or exclusion analysis, line scans were drawn over collagen-fibril-associated caveola and invadopodia structures; each pixel with a signal intensity value for both markers (that is CAV1/cortactin, CAVIN1/cortactin or TKS5/cortactin). Pixels were then plotted on a graph in which the x axis represents the fluorescence intensity for one channel and the y axis represents the fluorescence intensity for the second channel. Linear regression of the dot distribution was analysed for each line scan and the Pearson correlation coefficient was determined and plotted on a separated graph.

For collagen fibril (tangential) curvature measurement, Substract background was applied before using the Kappa−Curvature Analysis Plugin in Fiji[69] to measure the fibril point tangential curvature. CAV1 or cortactin signal intensity values were also obtained using the Kappa plugin. A pseudocoloured image representing the collagen fibril point curvature was generated in Fiji.

## β1 integrin internalization assay

For the β1 integrin internalization assay, cells were seeded on top of a type I collagen layer and incubated for 30 min at 37 °C under 1% $CO_2$. The medium was then removed, and the cells were incubated with anti-β1-integrin antibodies (2 µg ml$^{-1}$ diluted in precooled medium) for 10 min at 4 °C. After incubation with antibodies, cells were washed once with cooled medium and normal medium was added. Cells were incubated for 60 min at 37 °C, 1% $CO_2$ to allow antibodies internalization. After incubation, cells were washed twice (30 s + 60 s) with an acidic wash buffer (0.5 M NaCl, 0.5% acetic acid, pH 3). Cells were then pre-extracted and fixed, as previously described, and stained for immunofluorescence microscopy using fluorophore-conjugated goat anti-mouse antibodies to reveal internalized anti-β1-integrin antibodies. The images were acquired using the inverted Eclipse Ti-2 (Nikon) spinning-disk confocal microscope (Yokogawa CSU-X1 confocal scanner unit interfaced steered with NIS-Elements software, Nikon) using a ×60 Plan Apo Lambda water-immersion objective. A z stack of images was taken every 0.2 µm and 10–15 internal consecutive planes were stacked using the Maximum Intensity Projection command. Quantification of internalized β1 integrin was performed using Fiji by detecting the number of β1-integrin-positive vesicles per cell.

## Collagen internalization assay

For collagen uptake, MDA-MB-231 cells expressing TKS5–GFP were seeded on top of a Cy5-labelled type I collagen fibrous layer for 2 h at 37 °C under 1% $CO_2$. Collagen I fibrils were stained for 30 min at 4 °C with anti-collagen-I antibodies before fixation and permeabilization to exclude staining of intracellular collagen. After washing in cold PBS, cells were fixed in 4% PFA in PBS for 20 min and permeabilized in 0.1% Triton X-100 in 4% PFA in PBS for 10 min. GFP was detected using anti-GFP immunostaining (to determine the cell boundaries) and the samples were incubated with secondary anti-mouse antibodies to identify anti-collagen-I antibodies. The images were acquired using the inverted Eclipse Ti-2 (Nikon) spinning-disk confocal microscope (Yokogawa CSU-X1 confocal scanner unit interfaced steered with NIS-Elements software, Nikon) using a ×60 Plan Apo Lambda water-immersion objective. A z stack of images was taken every 0.2 µm and 15–25 internal consecutive planes were stacked using the Maximum Intensity Projection command. For quantification of intracellular collagen, a threshold was applied to images of Cy5-conjugated collagen I (channel 1) and anti-collagen I (channel 2), and the channel 2 image was subtracted from the channel 1 image using the Image Calculator plugin in Fiji. The signal corresponding to intracellular collagen was measured using the Measure plugin in Fiji and expressed as the percentage of the entire cell area.

## Microfabrication of nanobar arrays and experimental assay

Nanobar arrays used in this work were fabricated on square silicon chips using electron-beam lithography (ThunderNIL). Cells were seeded on the silicon chips coated with 0.5 µg ml$^{-1}$ poly-L-lysine (−ECM) or gelatin (+ECM) and cultured for 4 h at 37 °C under a 1% $CO_2$ atmosphere and then pre-extracted, fixed and stained for immunofluorescence microscopy as described above. Images were taken using an inverted Eclipse Ti-E (Nikon) spinning-disk confocal microscope (Yokogawa CSU-X1 confocal scanner unit interfaced with Metamorph software) using a ×60 CFI Plan Apo VC 1.4 oil-immersion objective. A z stack of images was taken every 0.2 µm by mean of a piezoelectric motor (Physik Instrumente). For quantification of CAV1 or TKS5 recruitment on nanobar-induced membrane deformations, z stacks of two consecutives planes on top of the nanobar (top) and on the bottom of the nanobar (bottom) were made using the Maximum Intensity Projection command in Fiji. The CAV1 or TKS5 signal was measured as the ratio of the signal on top of the nanobar divided by the signal aside the nanobar (defined as the sum of signal on both sides of the nanobar).

## Platinum replica transmission EM and immunogold labelling of unroofed cells

MDA-MB-231 cells were grown on glass coverslips coated with a thin layer of type I collagen as described above. Unroofing was performed by sonication. Coverslips were quickly rinsed three times in Ringer +Ca (155 mm NaCl, 3 mm KCl, 3 mm NaH$_2$PO$_4$, 5 mm HEPES, 10 mm glucose, 2 mm CaCl$_2$, 1 mm MgCl$_2$, pH 7.2), then immersed for 10 s in Ringer −Ca (155 mm NaCl, 3 mm KCl, 3 mm NaH$_2$PO$_4$, 5 mm HEPES, 10 mm glucose, 3 mm EGTA, 5 mm MgCl$_2$, pH 7.2) containing 0.5 mg ml$^{-1}$ poly-L-lysine, then quickly rinsed in Ringer −Ca then unroofed by scanning the coverslip with rapid (2–5 s) sonicator pulses at the lowest deliverable power in KHMgE buffer (70 mm KCl, 30 mm HEPES, 5 mm MgCl$_2$, 3 mm EGTA, pH 7.2). Unroofed cells were immediately fixed in KHMgE: 4% PFA for 10 min for light microscopy analysis, 4% PFA for 45 min for PREM analysis of immunogold-labelled samples and 2% PFA−2% glutaraldehyde for 10–20 min for morphological analysis by PREM. Glutaraldehyde-fixed samples were subsequently quenched using 0.1% NaBH$_4$ in KHMgE for 10 min. Immunogold labelling was performed in detergent-free buffer (KHMgE, 1% BSA), the samples were blocked for 30 min, incubated for 1 h and 30 min with the primary antibody (using a 1:20 dilution) and rinsed and incubated twice for 20 min with the gold-coupled secondary antibodies. The samples were rinsed again and post-fixed with 2% glutaraldehyde.

Sample processing for platinum-replica EM of unroofed cells was performed as follows: 2% glutaraldehyde/2% paraformaldehyde-fixed cells were further sequentially treated with 0.5% OsO$_4$, 1% tannic acid and 1% uranyl acetate before graded ethanol dehydration and hexamethyldisilazane (HMDS) substitution (LFG Distribution). The dried samples were then rotary shadowed with 2 nm of platinum (sputtering) and 4–6 nm of carbon (carbon thread evaporation) using the ACE600 metal coater (Leica Microsystems). The resultant platinum replica was floated off the glass with hydrofluoric acid (5%), washed several times with distilled water and picked up on 200-mesh formvar/carbon-coated EM grids. The grids were mounted in a eucentric side-entry goniometer stage of a transmission electron microscope operated at 120 kV (JEOL), and the images were recorded using the Xarosa digital camera (EM-SIS). The images were processed in Adobe Photoshop to adjust the brightness and contrast and were presented in inverted contrast.

## STORM acquisition and analysis

For STORM imaging, cells were seeded on coverslips (Menzel glaser 18-mm diameter #1.5) coated with a thin layer of Cy5-conjugated type I collagen fibrils for 60 min, fixed in 4% PFA in PBS and stained for CAV1 and cortactin as described above. IgG-mouse Alexa Fluor 647 (Molecular Probes) and IgG-rabbit CF568 (Merck) were used as secondary antibodies. A few minutes before the acquisition, the samples were mounted onto a glass slide with a 15-mm-diameter hole. The hole was filled with imaging buffer (Tris 50 mM, NaCl 10 mM, 10% glucose, 100 mM MEA, 0.6 mg ml$^{-1}$ glucose oxidase (Sigma-Aldrich) and 40 µg ml$^{-1}$ catalase) and sealed with Picodent twinsil (Picodent). Then, 2D images were acquired using the SAFe360 module (Abbelight) coupled to an inverted Eclipse Ti-2 (Nikon) optical microscope equipped with a ×100 oil-immersion objective (1.49 NA) and a Perfect Focus System for z focus stabilization. The samples were illuminated sequentially with 633 and 561 nm lasers (500 mW for both, Oxxius laser). A 405 nm laser (100 mW, Oxxius laser) was used for the reactivation of the fluorophores. Laser illumination was at an angle just below the critical angle to obtain a highly inclined laminated optical sheet (HiLo) illumination mode. Images were acquired using a sCMOS Hamamatsu Fusion camera. The NEO software (Abbelight) was used to localize particles, correct the drift and, finally, to reconstruct the final images. The type I collagen channel image was aligned manually to the reconstituted STORM image.

### Invasion of multicellular spheroids in 3D type I collagen

For siRNA treatment, cells were transfected by nucleofection (Kit V; Lonza) with 50 nM siRNAs as indicated. After nucleofection, cells were plated in a Petri dish. The next day, a second round of transfection was performed with 50 nM of each siRNA using Lullaby reagent according to the manufacturer's recommendations for 6 h. Multicellular spheroids were made immediately after the second round of siRNA treatment using $3 \times 10^3$ cells in a 20 µl droplet in complete L-15 medium and the hanging-drop method[70]. After 3 days, spheroids were embedded in type I collagen as described above. Spheroids were fixed in 4% PFA in PBS immediately after polymerization of the matrix (T0) or after 48 h of invasion (T2). After fixation, cells in spheroids were permeabilized for 15 min in 0.1% Triton X-100/PBS and labelled with Alexa Fluor 488–phalloidin and DAPI. For quantification of invasion in 3D type I collagen matrix, phalloidin-labelled spheroids were imaged using the inverted Eclipse Ti-2 (Nikon) spinning-disk confocal microscope (Yokogawa CSU-X1 confocal scanner unit interfaced with NIS-Elements software) using a ×20 Plan Apo Lambda water-immersion objective and collecting a stack of images along the $z$ axis with a 2 µm interval between two optical sections. The spheroid mean diameter was measured from azimuthal averaging of the intensity profile along a line centred on the spheroid using a custom plugin in Fiji software[69]. Averaging consists of measuring intensity profiles along a rotating line by 5° steps and calculating the mean value over all angles of each pixel of the line. The mean diameter was then taken as the width at 1/10 of the maximal value of these mean intensity profiles. Mean area ($\pi r^2$) was calculated from the mean diameter. Each value represents a single spheroid.

### Statistics and reproducibility

Statistical analyses and data presentation were performed using GraphPad Prism (v.8.0, v.9.0 and v.10.0). Data were tested for normal distribution using the D'Agostino–Pearson normality test and nonparametric tests were applied otherwise. Sample sizes ($n$) and significance values are indicated in the figures and figure legends; $*P < 0.05$, $**P < 0.005$, $***P < 0.0005$; NS, not significant. All of the findings were reproducible over multiple independent experiments, within a reasonable degree of variability between replicates. The number of biological replicate experiments for each assay is provided in the respective figure legends. No statistical method was used to predetermine sample size, which was determined in accordance with standard practices in the field. No data were excluded from the analyses. The investigators were not blinded to allocation during experiments and outcome assessment. All source numerical data are provided as Source Data.

### Reporting summary

Further information on research design is available in the Nature Portfolio Reporting Summary linked to this article.

## Data availability

The main data supporting the findings of this study are available within the Article and its Supplementary Information. All other data supporting the findings of this study are available from the corresponding author on reasonable request. Source data are provided with this paper.

## Code availability

All codes used for this study are available from the corresponding author on request.

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

## Acknowledgements

We thank M. Sixt and G. Montagnac for comments on the manuscript; D. Mazaud for help with STORM experiments; C. Gamblin for setting up the collagen internalization assay; A. Kawska (http://www.IlluScientia.com) for artwork; the members of our laboratories (C.L. and P.C.) for comments during the preparation of this Article; the members of the Cell and Tissue Imaging platform (PICT-IBiSA) of CNRS UMR144 and of Institut Curie and the Nikon Imaging Centre at Institut Curie-CNRS, the members of the French National Research Infrastructure France-BioImaging (ANR10-INBS-04) and the staff at the IBPS cryo-electron microscopy platform (Sorbonne University). This study was supported by the following grants: INCa 2018-1-PL BIO-08-ICR-1 (decision no. 2018-154) to C.L. and P.C.; a generous donation from Mr. T. Paulsen (InvaCell Project) and by institutional funding from Institut Curie and Centre National de la Recherche Scientifique to P.C. ANR-21-CE13-0018-01 and core funding from institute of Myology to S.V. The funders had no role in study design, data collection and analysis, decision to publish or preparation of the manuscript.

## Author contributions

Conceptualization: P.M., C.L. and P.C. Experimental design: S.V., B.L., P.M., C.L. and P.C. Investigation and data analysis: P.M., F.R. and C.G. PREM analysis: E.L. and S.V. Further information regarding PREM analysis can be obtained through S.V. (s.vassilopoulos@institut-myologie.org). Tools and resources: A.-S.M. and D.R. Funding acquisition: C.L. and P.C. Writing the original draft: P.M. and P.C. Text editing: S.V. and C.L. Supervision: P.C. All of the authors approved the final version of the manuscript and agree on the content and conclusions.

## Competing interests

The authors declare no competing interests.

## Additional information

**Extended data** is available for this paper at https://doi.org/10.1038/s41556-023-01272-z.

**Correspondence and requests for materials** should be addressed to Pedro Monteiro, Christophe Lamaze or Philippe Chavrier.

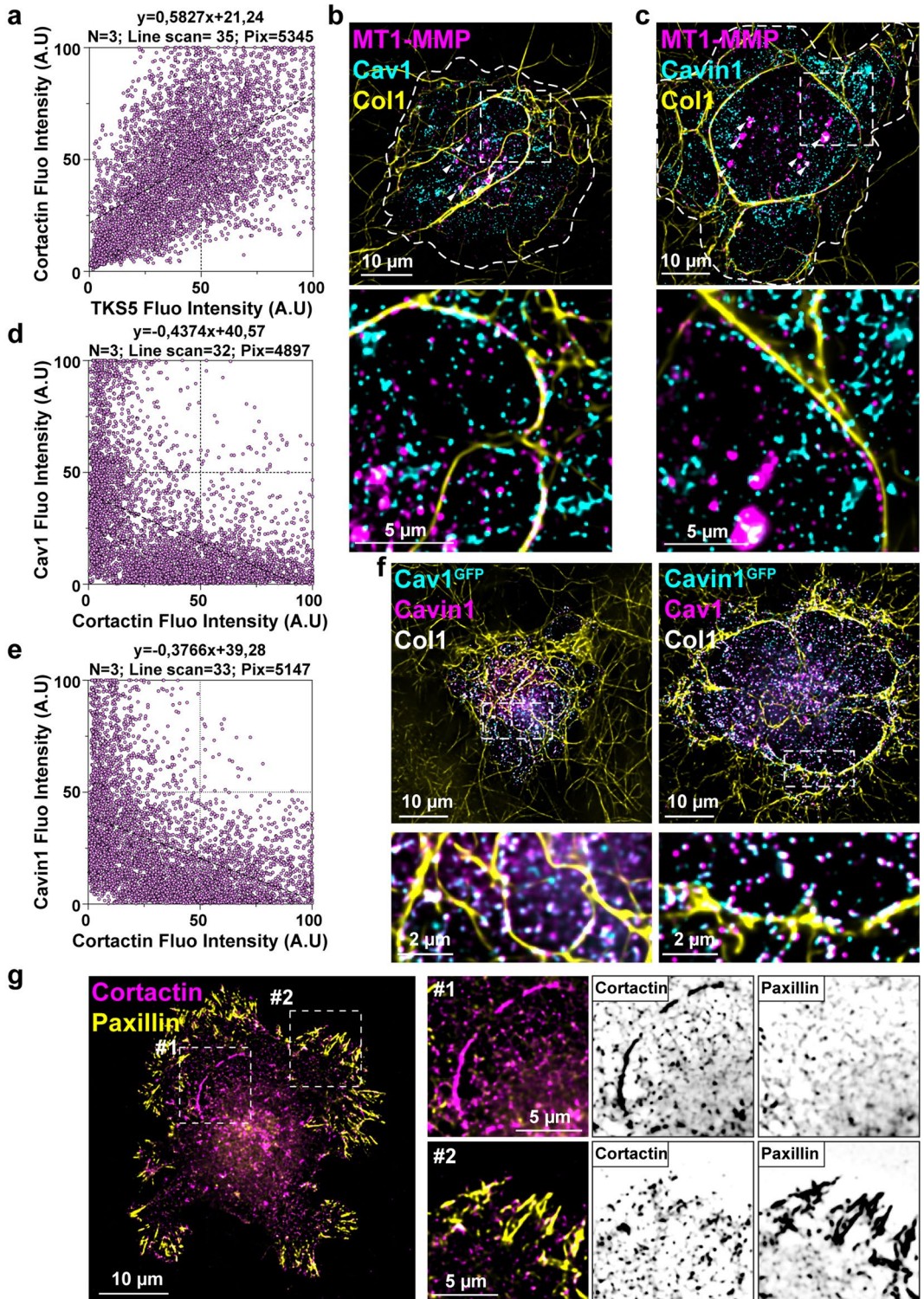

**Extended Data Fig. 1 | Localization of MT1-MMP and caveolae components in relation with type I collagen fibrils (supplements Fig. 1). (a)** Point cloud distribution of pixel fluorescence intensity for TKS5 and Cortactin showing co-localization of the markers (see linear regression). Dots represent 5345 xy pairs (that is pixels) from 35 linescans from 3 biologically independent experiments. **(b,c)** MDA-MB-231 cells cultured on a layer of fibrous type I collagen for 60 min were fixed and stained for MT1-MMP and Cav1 (b) or Cavin1 (c). Dotted line, cell contour. Insets are zoom-in view of the boxed regions. Arrowheads point to perinuclear MT1-MMP-positive endolysosomes. **(d,e)** Point cloud distribution of pixel fluorescence intensity for the indicated markers showing exclusion (inverted linear regression). (plot in d) Dots represent 4897 xy pairs (that is pixels) from 32 linescans from 3 biologically independent experiments. (plot in panel e) Dots represent 5147 xy pairs (that is pixels) from 33 linescans from 3 biologically independent experiments. Source numerical data are provided in Source Data file. **(f)** Cav1 and Cavin1 co-localization in MDA-MB-231 cells. Insets are zoom-in view of the boxed regions. **(g)** MDA-MB-231 cells plated for 60 min on type I collagen (unstained), labelled for focal adhesion protein, paxillin (yellow LUT) and invadosome marker, cortactin (magenta LUT). Insets are zoom-in view of the boxed regions showing dissociation of peripheral focal adhesion and more internal invadosome structures.

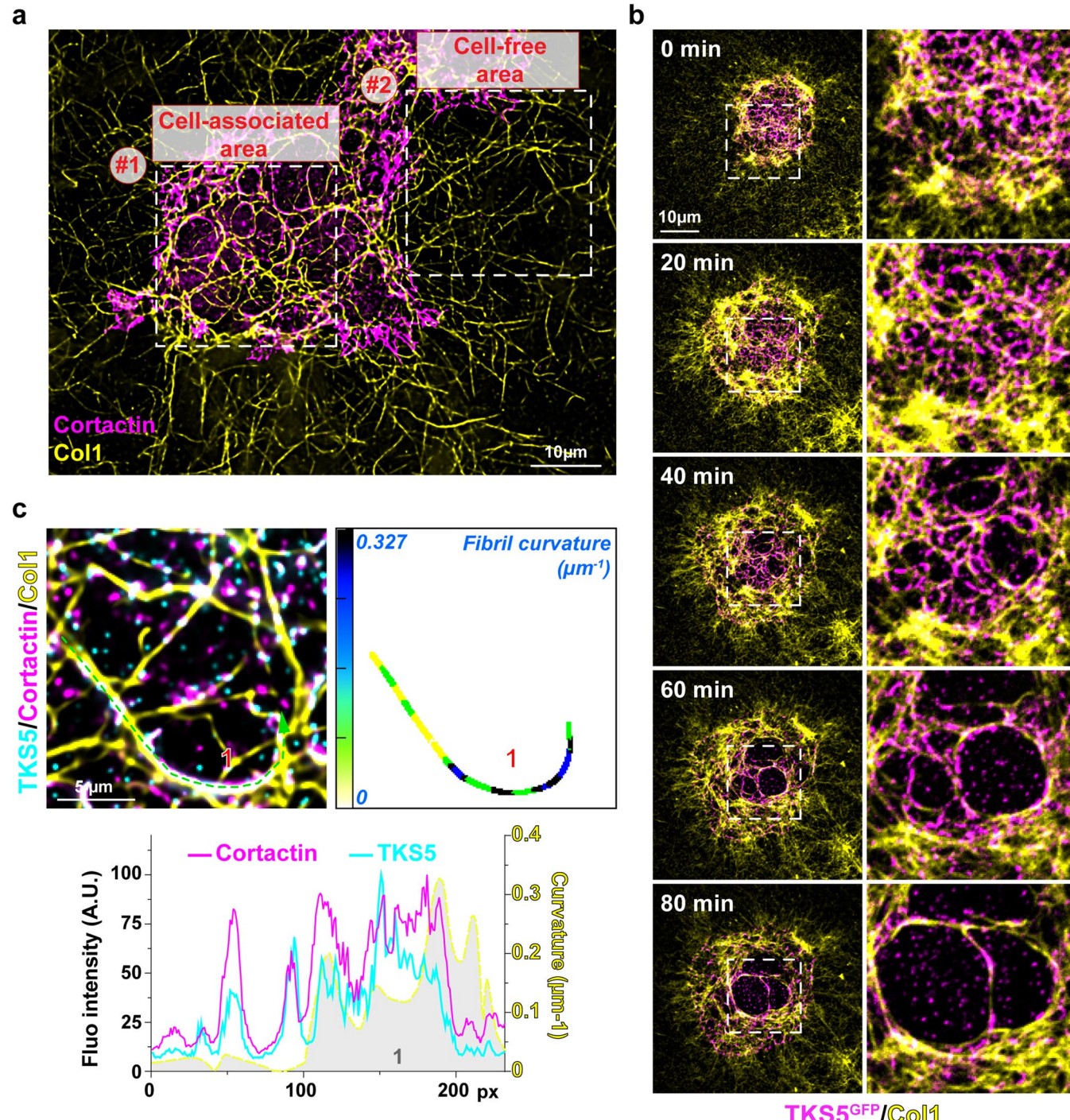

**Extended Data Fig. 2 | Correlation of fibril curvature with caveolae and invadosome association (supplements Fig. 2). (a)** MDA-MB-231 cells cultured on a layer of fibrous type I collagen (yellow) for 60 min were fixed and stained for cortactin (magenta). **(b)** Gallery of selected time frames (min) from a time-lapse sequence of MDA-MB-231 cells expressing TKS5^GFP (magenta) plated on a layer of Cy5-labelled collagen I fibrils (yellow). The right row of images is a zoom-in view of the boxed region. **(c)** MDA-MB-231 cells cultured on a layer of fibrous type I collagen for 60 min were fixed and stained for the indicated markers. Representation of tangential curvature along the collagen fibrils using the indicated LUT (top right row), in comparison with the distribution of invadosome markers along the confining fibrils (top left row). Highly curved segments are numbered. Signal intensities and tangential curvature are profiled in the bottom row. The dotted line and arrow indicate the region and direction of the linescan.

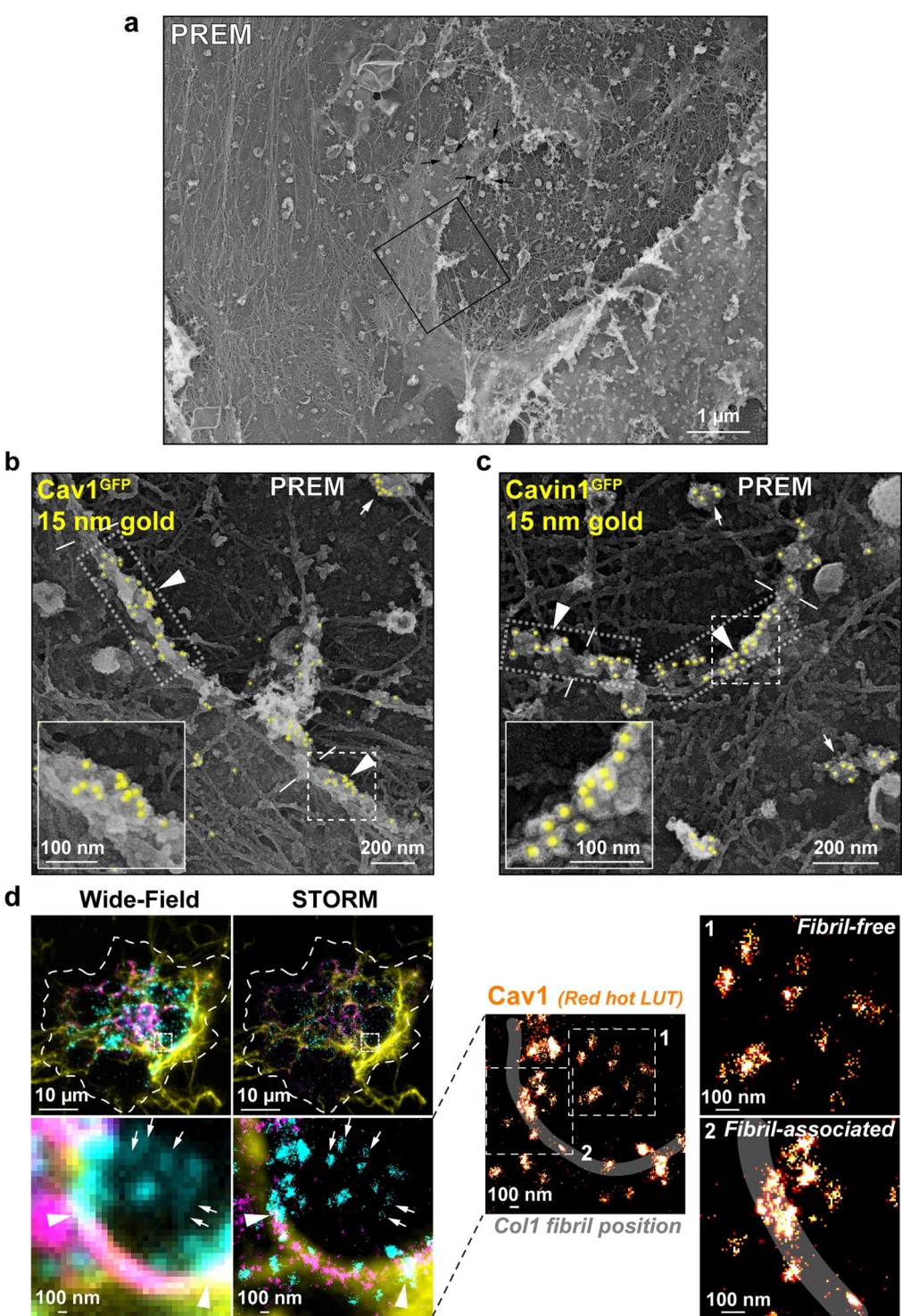

**Extended Data Fig. 3 | Ultrastructural analysis of fibril-associated caveolae and invadosome organization. (a)** Low-magnification platinum replica electron microscopy (PREM) view of an unroofed MDA-MB-231 cell plated on a layer of type I collagen fibrils in inverted grayscale. The boxed regions correspond to the high magnification PREM view shown in **b**. Arrows, clathrin-coated pits (CCPs). **(b, c)** Anti-GFP immunogold PREM of unroofed MDA-MB-231 cells expressing Cav1^GFP (b) or Cavin1^GFP (c) plated for 60 min on a thin layer of collagen I fibrils (grayscale is inverted, that is electron-denser, lighter). Immunogold beads are pseudo-coloured in yellow. Insets are zoom-in view of the boxed regions to show fibril-associated caveolar structures. Dotted-line boxes, fibril-associated Cav1^GFP and Cavin1^GFP-positive µm-size caveolar clusters. Facing dashes, electron-dense

collagen fibrils underneath the ventral plasma membrane. Arrows, individual caveolae pits. Arrowheads, fibril-associated caveolar clusters. **(d)** Corresponding wide-field (left panel) and Stochastic Optical Reconstruction Microscopy (STORM, right panel) images of an MDA-MB-231 cell plated for 60 min on fluorescently labelled fibrous type I collagen layer. After fixation, cells were stained for Cav1 and cortactin. Insets are zoom-in views of the boxed regions to show the alternation of caveolae and invadosome structures and different caveolar structure sizes in fibril-free (box #1) and fibril-associated (box #2) PM regions at high resolution. Cav1 STORM signal is pseudo-coloured with the indicated LUT. Dotted line, cell contour. Arrows and arrowheads as in panel b, c. Grey line indicates collagen fibril position.

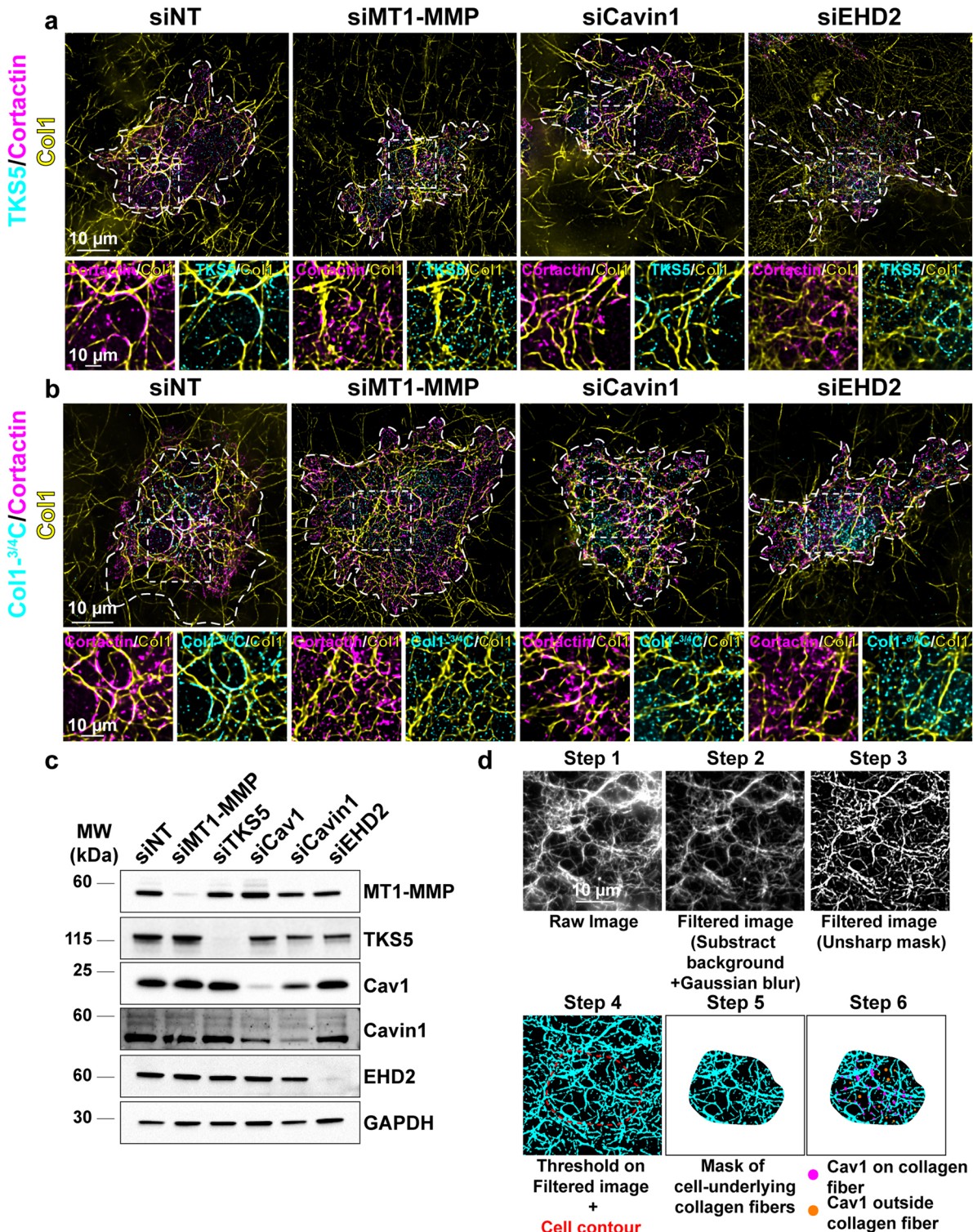

**Extended Data Fig. 4 | Characterization of MDA-MB-231 cells silenced for core caveolae and invadosome components (supplements Fig. 3). (a,b)** MDA-MB-231 cells knocked down for the indicated proteins by siRNA treatment were cultured on a layer of type I collagen fibrils for 60 min, fixed and stained for the indicated markers. Dotted line, cell contour. Insets are zoom-in view of the boxed regions showing collagen fibrils and TKS5 (a) or Col1-³/⁴C (b) distribution with an inverted grey LUT. **(c)** MDA-MB-231 cells knocked down for the indicated proteins by siRNA treatment for 72 hrs were lysed and proteins were analysed by polyacrylamide gel electrophoresis (SDS-PAGE) and immunoblotting with

the indicated antibodies (representative immunoblots are shown). Loading was estimated by detection of GAPDH. Molecular weights are in kDa. **(d)** Overview of ImageJ-based analysis pipeline of collagen fibril-associated Cav1 signal. Starting from a fluorescence image of labelled fibrous type I collagen layer, the image is filtered with the indicated filters. The cell contour is drawn manually, and a threshold is applied to generate a mask of the cell-underlying collagen fibrils. Cav1 signal (intensity, area) is quantified inside (fibril-associated Cav1 signal) and outside (Cav1 signal in the fibril-free PM region) the mask within the cell contour limit. Total cell-associated Cav1 is calculated as the sum of these two values.

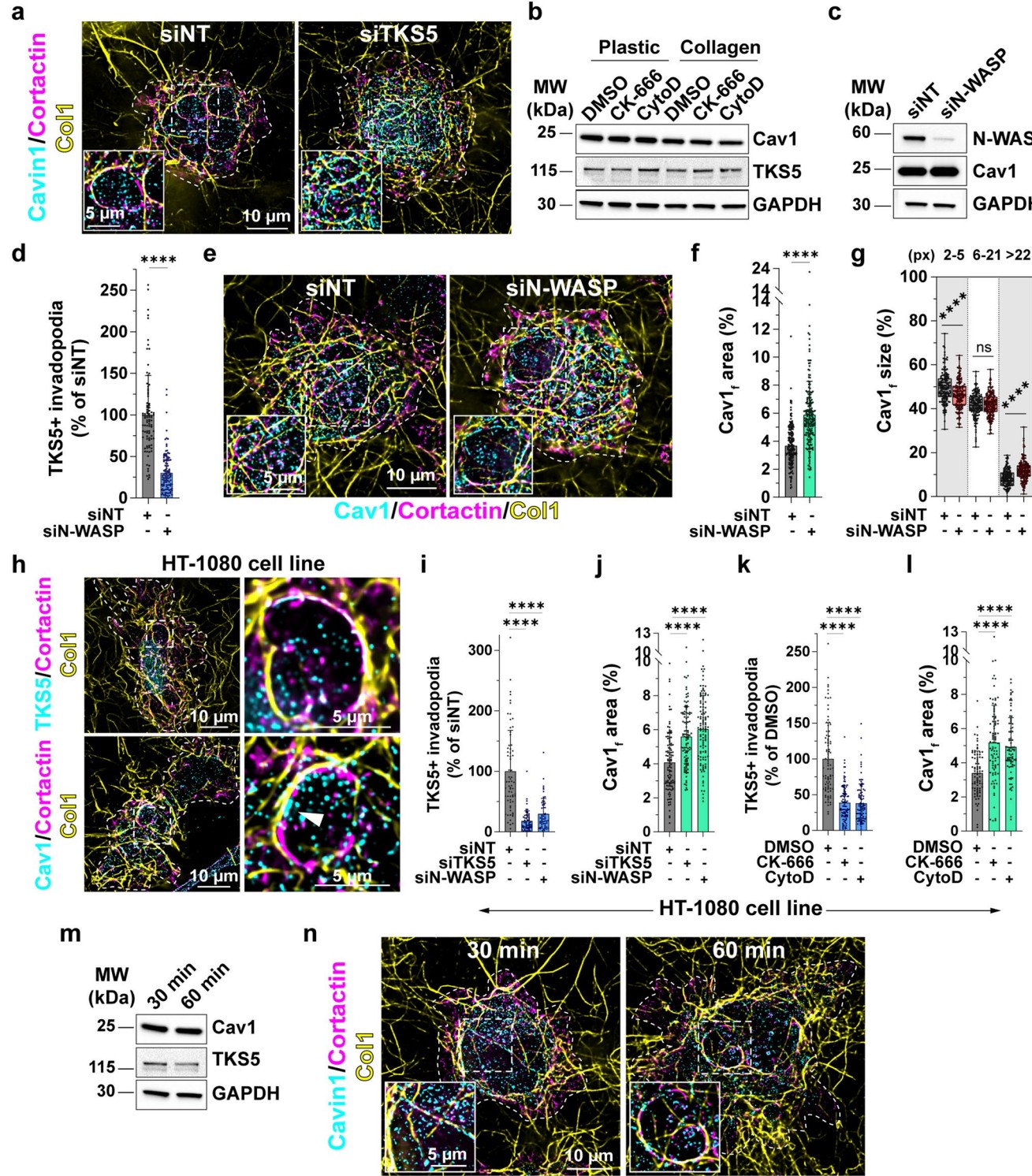

**Extended Data Fig. 5 | See next page for caption.**

**Extended Data Fig. 5 | Invadosome function impacts the dynamic association of caveolae with collagen fibrils in breast cancer and fibrosarcoma cells (supplements Fig. 4). (a)** MDA-MB-231 cells (TKS5-KD) stained for Cavin1/Cortactin. As in all images: insets, zoom-in views of the boxed regions; dotted lines, cell contour. **(b)** CK-666- or CytoD-treated MDA-MB-231 cells. TKS5 and Cav1 expression analysed by immunoblotting. As for all immunoblots: representative immunoblots are shown, loadings were estimated by GAPDH and molecular weights, kDa. **(c)** N-WASP and Cav1 expression in MDA-MB-231 cells (N-WASP-KD) lysates. **(d)** N-WASP-KD effect on invadosomes. Mean and standard deviation (SD) are shown; nsiNT = 93 cells, nsiN-WASP = 115 cells from 3 biologically independent experiments. **(e)** MDA-MB-231 cells (N-WASP-KD) stained for Cav1/Cortactin. **(f)** Cav1f area quantification. Mean and SD are shown; nsiNT = 170 cells, nsiN-WASP = 193 cells from 4 biologically independent experiments; for graphs in (d) and (f), unpaired two-tailed t-test; **** P < 0.0001. **(g)** Fibril-associated Cav1 size distribution. Mean and minimum-to-maximum are shown, box ranges from the first (Q1–25th percentiles) to the third quartile (Q3–75th percentiles) of

the distribution are shown; nsiNT = 154 cells, nsiCavin1 = 160 cells from 4 biologically independent experiments; unpaired two-tailed t-test; **** P < 0.0001. NS, not significant. **(h)** HT-1080 cells stained for Cortactin/TKS5 or Cortactin/Cav1. **(i)** TKS5- or N-WASP-KD effect on invadosomes. Mean and SD are shown; nsiNT = 67 cells, nsiTKS5 = 60 cells, nsiN-WASP = 57 cells from 2 biologically independent experiments. **(j)** TKS5- or N-WASP-KD effect on Cav1f area. Mean and SD are shown; nsiNT = 106 cells, nsiTKS5 = 105 cells, nsiN-WASP = 115 cells from 3 biologically independent experiments. **(k)** CK-666 or CytoD treatment effect on invadosomes. Mean and SD are shown; nDMSO = 88 cells, nCK-666 = 98 cells, nCytoD = 95 cells from 3 biologically independent experiments. **(l)** CK-666 or CytoD effect on Cav1f area. Mean and SD are shown; nDMSO = 79 cells, nCK-666 = 92 cells, nCytoD = 85 cells from 3 biologically independent experiments; for graphs in (i-l), one-way ANOVA using Kruskal-Wallis comparison test, **** P < 0.0001. Source numerical data provided in Source Data file. **(m)** Cav1 and TKS5 expression in lysates of MDA-MB-231 cells incubated on collagen. **(n)** MDA-MB-231 cells stained for Cavin1/Cortactin.

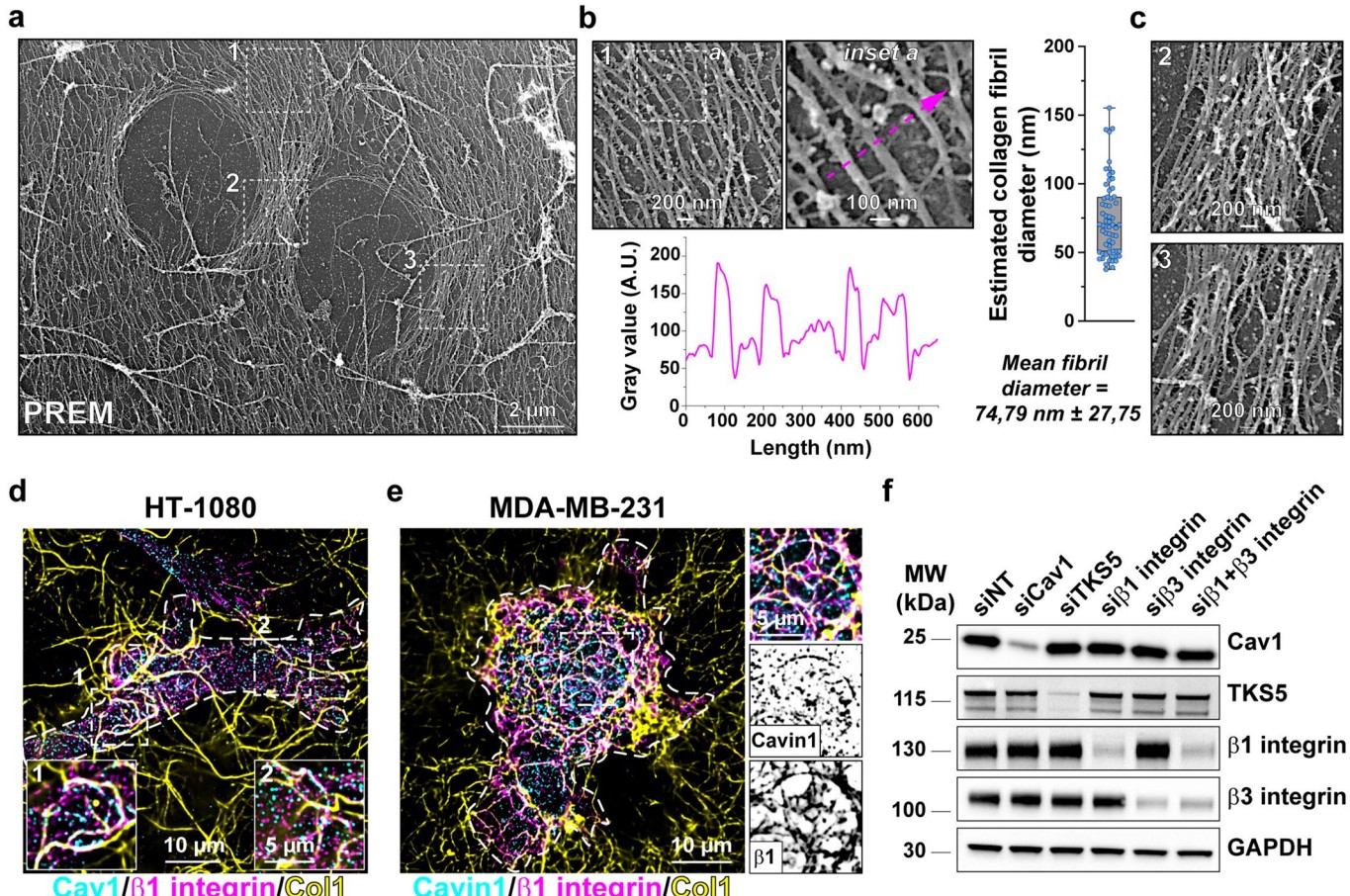

**Extended Data Fig. 6 | Integrin association with the collagen fibrils (supplements Fig. 5). (a)** Low-magnification PREM view of a type I collagen fibril network in inverted grayscale. **(b)** Inset is a zoom-in view of the boxed region #1 showed in panel a. Right row, inset is a zoom-in view of the boxed region showed in panel b. Collagen fibril diameter estimated by linescan analysis through individual collagen fibrils. Distribution of collagen fibril diameter from 62 individual fibrils. Source numerical data are provided in Source Data file. **(c)** Insets are zoom-in views of the indicated boxed regions in panel a showing fibril bundles. **(d)** HT-1080 cells were cultured on a layer of type I collagen fibrils for 60 min and stained for Cav1 and β1 integrin. Dotted line, cell contour. Insets are zoom-in views of the boxed regions. **(e)** MDA-MB-231 cells were cultured on a layer of type I collagen fibrils for 60 min and stained for Cavin1 and β1 integrin. Dotted line, cell contour. Insets are zoom-in views of the boxed regions. **(f)** MDA-MB-231 cells knocked down for the indicated proteins by siRNA treatment for 72 hrs were lysed and proteins were analysed by polyacrylamide gel electrophoresis (SDS-PAGE) and immunoblotting with the indicated antibodies (representative immunoblots are shown). Loading was estimated by detection of GAPDH. Molecular weights are in kDa.

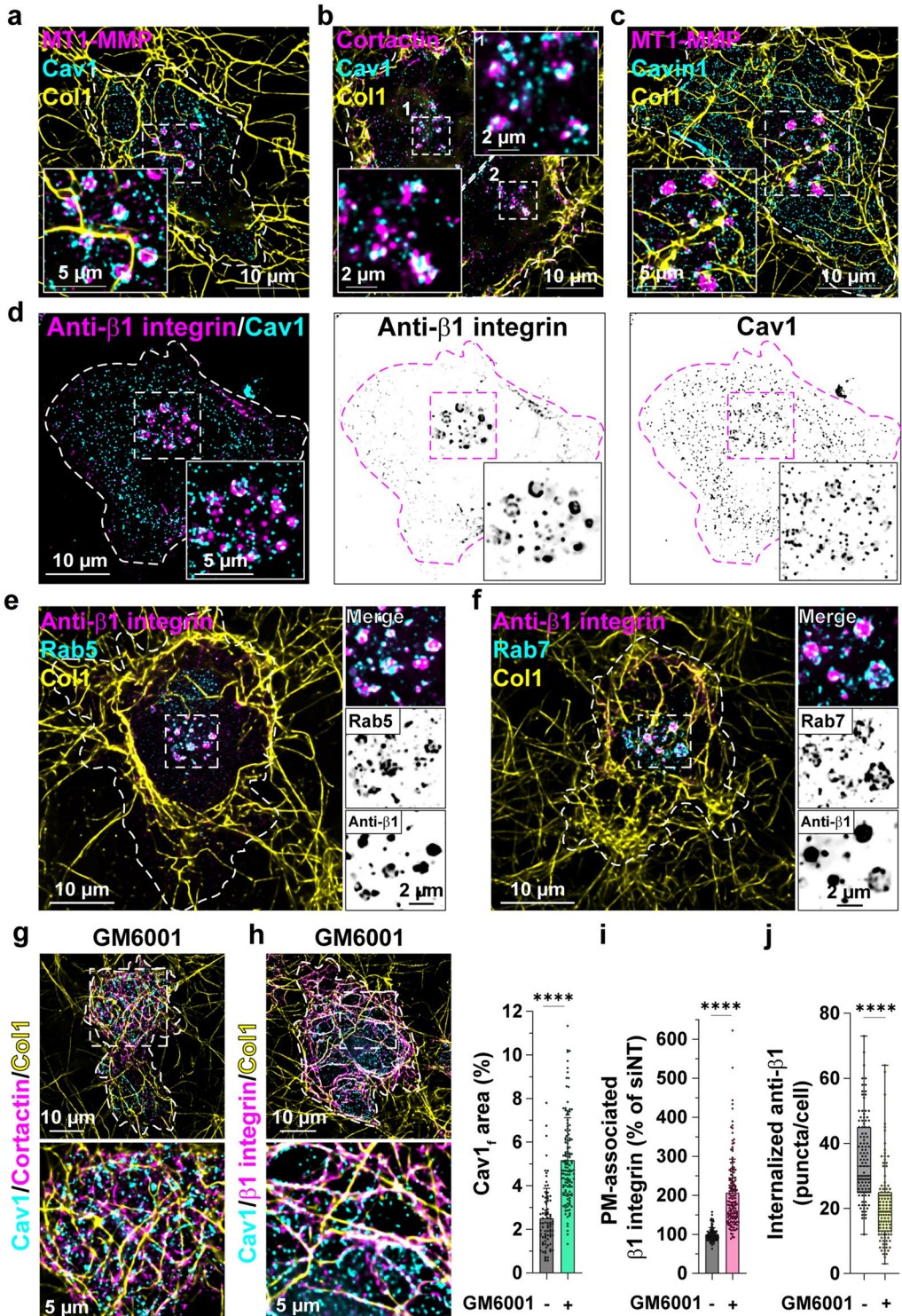

**Extended Data Fig. 7 | See next page for caption.**

**Extended Data Fig. 7 | Caveolae association with endolysosomal compartments (supplements Fig. 6). (a,b,c)** MDA-MB-231 cells cultured on a layer of type I collagen fibrils for 60 min were stained for Cav1 (a,b) or Cavin1 (c) together with the indicated markers. The images are projection of three internal optical sections to show the cytoplasmic vesicles. Dotted line, cell contour. Insets are zoom-in views of the boxed regions. **(d-f)** Surface-exposed β1 integrins were labelled with antibodies added to non-permeabilized MDA-MB-231 cells. After 60 min at 37 °C, cells were fixed and the presence of internalized anti-β1 integrin antibodies, Cav1 and indicated endolysosomal markers was analysed. The views are projection of three internal optical sections to show anti-β1 integrin antibody association with cytoplasmic vesicles positive for Cav1- (d), Rab5- (e) and Rab7- (f). Dotted line, cell contour. Insets are zoom-in views of the boxed regions. **(g)** MDA-MB-231 cells cultured on fibrous type I collagen were treated with pan-MMP inhibitor, GM6001, and stained for Cav1 and cortactin. Inset is a zoom-in view of the boxed region. **(h)** MDA-MB-231 cells treated with GM6001 stained for Cav1 and β1 integrin. The image is a projection of three optical planes at the level of the ventral PM. Inset is a zoom-in view of the boxed region. The effect of GM6001 on Cav1 association with cell-associated fibrils is quantified (Cav1$_f$ area). Mean and standard deviation are shown; $n_{EtOH}$ = 85 cells, $n_{GM6001}$ = 132 cells from 3 biologically independent experiments. **(i)** Ventral PM β1 integrin signal in GM6001-treated cells is plotted and normalized to the signal in vehicle (ethanol)-treated cells. Mean and standard deviation are shown; $n_{EtOH}$ = 109 cells, $n_{GM6001}$ = 150 cells from 4 biologically independent experiments. **(j)** After surface labelling (see Fig. 6f), the presence of internalized anti-β1 integrin antibodies in intracellular vesicles was quantified in GM6001-treated as compared to vehicle-treated cells. Mean and minimum to maximum are shown, box ranges from the first (Q1-25th percentiles) to the third quartile (Q3-75th percentiles) of the distribution are shown; $n_{EtOH}$ = 112 cells, $n_{GM6001}$ = 115 cells from 3 biologically independent experiments; for graphs in (h-j), unpaired two-tailed $t$-test; **** $P$ < 0.0001. Source numerical data are provided in Source Data file.

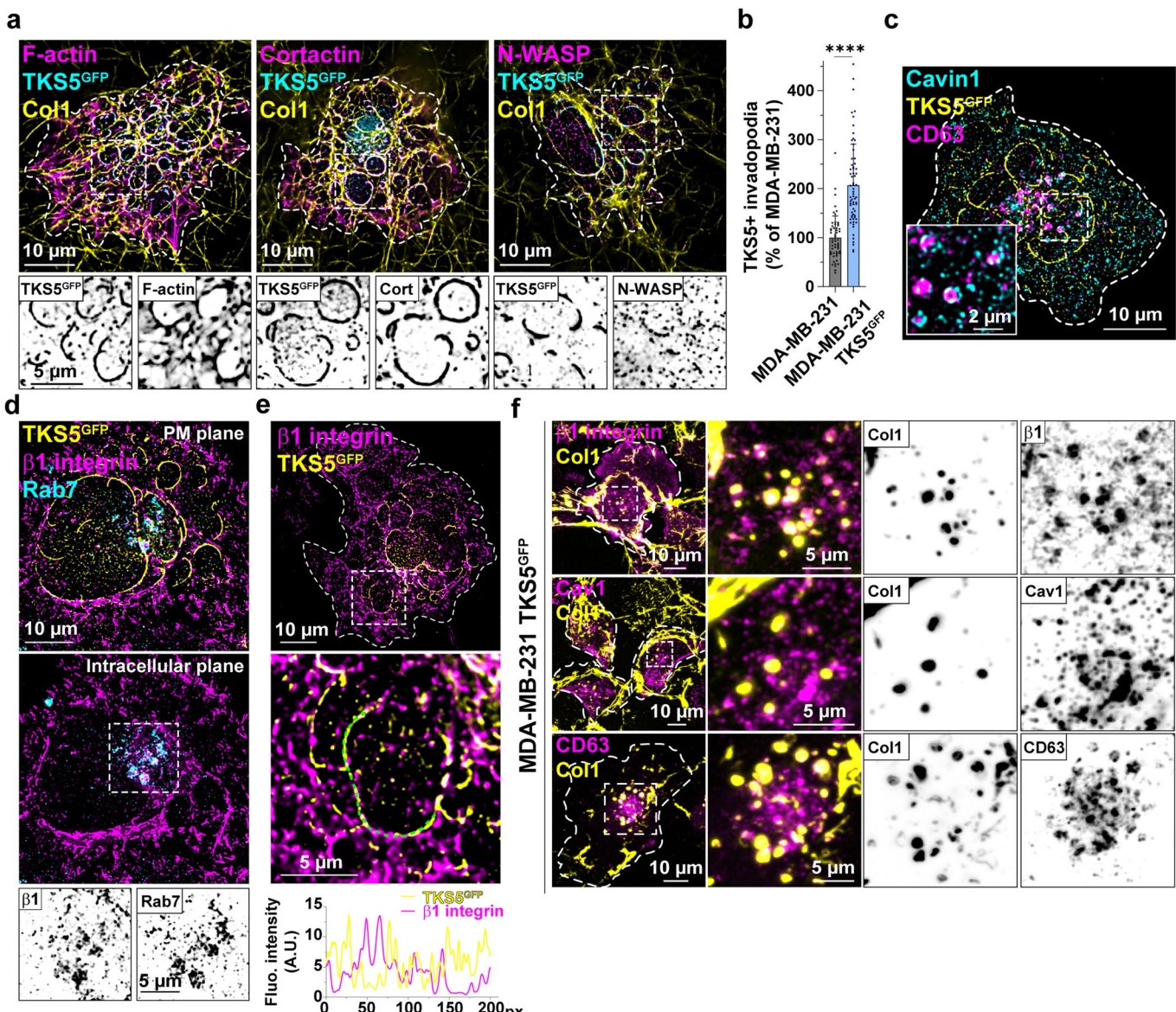

**Extended Data Fig. 8 | Invadosome stimulation upon TKS5 overexpression (supplements Fig. 7). (a)** MDA-MB-231 cells overexpressing TKS5ᴳᶠᴾ were cultured on fibrous type I collagen for 60 min and stained for GFP and counterstained for the indicated markers. Dotted line, cell contour. Insets are zoom-in views of the boxed regions showing TKS5 and F-actin (a), TKS5 and Cortactin (b) or TKS5 and N-WASP (c) in inverted grey LUT. **(b)** TKS5-positive invadosome were detected with anti-TKS5 antibodies. MDA-MB-231 cells were used as control. Mean and standard deviation are shown; $n_{MDA-MB-231}$ = 55 cells, $n_{MDA-MB-231/TKS5GFP}$ = 67 cells 3 biologically independent experiments; unpaired two-tailed $t$-test; **** $P < 0.0001$. Source numerical data are provided in Source Data file. **(c)** MDA-MB-231 cells overexpressing TKS5ᴳᶠᴾ cultured on fibrous type I collagen for 60 min were stained for GFP and Cavin1 and for endolysosomal marker, CD63. Dotted line, cell contour. Inset is a zoom-in view of the boxed

regions. **(d)** Immunofluorescence images of TKS5ᴳᶠᴾ-overexpressing MDA-MB-231 cells on type I collagen stained for β1 integrin and endolysosomal marker, Rab7. The images are projections of three optical planes at the level of the ventral PM (left) and inside the cell (right). Insets are zoom-in views of the boxed region showing β1 integrin and Rab7 using an inverted grey LUT. **(e)** TKS5ᴳᶠᴾ-positive invadosome show reduced β1 integrin association with invadosome. Dotted line, cell contour. Inset is a zoom-in view of the boxed region. The lower panel shows a linescan profiling of β1 integrin and TKS5ᴳᶠᴾ signal intensity. **(f)** Type I collagen was detected in CD63-positive endolysosomes positive for β1 integrin and Cav1. Dotted line, cell contour. Insets are zoom-in views of the boxed regions showing Col1 and β1 integrin (upper row), Col1 and Cav1 (middle row) and Col1 and CD63 (lower row) in inverted grey LUT.

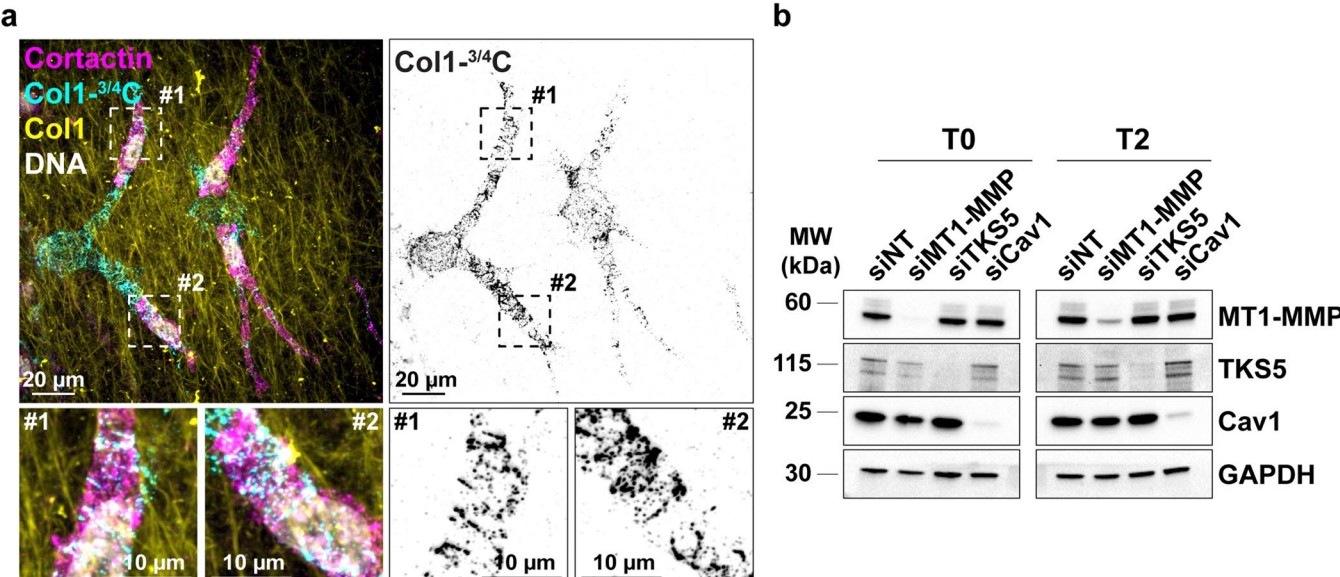

**Extended Data Fig. 9 | 3D cell invasion relies on invadosome-mediated collagenolysis (supplements Fig. 8). (a)** MDA-MB-231 cells embedded in a 3D type I collagen gel for 16 hrs and stained for cleaved collagen (Col1-³/⁴C) and cortactin. Cleaved collagen is shown separately in an inverted grey LUT (right row). Insets are zoom-in views of the boxed regions. **(b)** MDA-MB-231 cells knocked down for the indicated proteins by siRNA treatment for 72 (T0) or 120 hrs (T2) were lysed and proteins were analysed by polyacrylamide gel electrophoresis (SDS-PAGE) and immunoblotting with the indicated antibodies (representative immunoblots are shown). Loading was estimated by detection of GAPDH. Molecular weights are in kDa.

| | |
|---|---|

# Reporting Summary

## Statistics

For all statistical analyses, confirm that the following items are present in the figure legend, table legend, main text, or Methods section.

| n/a | Confirmed | |
|---|---|---|
| ☐ | ☒ | The exact sample size (*n*) for each experimental group/condition, given as a discrete number and unit of measurement |
| ☐ | ☒ | A statement on whether measurements were taken from distinct samples or whether the same sample was measured repeatedly |
| ☐ | ☒ | The statistical test(s) used AND whether they are one- or two-sided<br>*Only common tests should be described solely by name; describe more complex techniques in the Methods section.* |
| ☒ | ☐ | A description of all covariates tested |
| ☐ | ☒ | A description of any assumptions or corrections, such as tests of normality and adjustment for multiple comparisons |
| ☐ | ☒ | A full description of the statistical parameters including central tendency (e.g. means) or other basic estimates (e.g. regression coefficient) AND variation (e.g. standard deviation) or associated estimates of uncertainty (e.g. confidence intervals) |
| ☐ | ☒ | For null hypothesis testing, the test statistic (e.g. *F*, *t*, *r*) with confidence intervals, effect sizes, degrees of freedom and *P* value noted<br>*Give P values as exact values whenever suitable.* |
| ☒ | ☐ | For Bayesian analysis, information on the choice of priors and Markov chain Monte Carlo settings |
| ☒ | ☐ | For hierarchical and complex designs, identification of the appropriate level for tests and full reporting of outcomes |
| ☐ | ☒ | Estimates of effect sizes (e.g. Cohen's *d*, Pearson's *r*), indicating how they were calculated |

*Our web collection on statistics for biologists contains articles on many of the points above.*

## Software and code

Policy information about availability of computer code

| | |
|---|---|
| Data collection | NIS Elements (version 5.42.01) and Metamorph (version 7.8.0.0) softwares were used for data collection.<br>BioRad ChemiDoc MP Imaging System (version 6.1.0.07) was used for western blot analysis. |
| Data analysis | Fiji software (version 1.54) was used for data analysis.<br>GraphPad Prism (versions 8.0, 9.0 and 10.0) software was used for statistic calculations and data representation. |

For manuscripts utilizing custom algorithms or software that are central to the research but not yet described in published literature, software must be made available to editors and reviewers. We strongly encourage code deposition in a community repository (e.g. GitHub). See the Nature Portfolio guidelines for submitting code & software for further information.

## Data

Policy information about availability of data

All manuscripts must include a data availability statement. This statement should provide the following information, where applicable:
- Accession codes, unique identifiers, or web links for publicly available datasets
- A description of any restrictions on data availability
- For clinical datasets or third party data, please ensure that the statement adheres to our policy

The authors declare that the data supporting the findings of this study are available within the main and supplementary figures of this paper. All other data

supporting the findings are available on request.

## Human research participants

Policy information about studies involving human research participants and Sex and Gender in Research.

| | |
|---|---|
| Reporting on sex and gender | N/A |
| Population characteristics | N/A |
| Recruitment | N/A |
| Ethics oversight | N/A |

Note that full information on the approval of the study protocol must also be provided in the manuscript.

# Field-specific reporting

Please select the one below that is the best fit for your research. If you are not sure, read the appropriate sections before making your selection.

☒ Life sciences        ☐ Behavioural & social sciences        ☐ Ecological, evolutionary & environmental sciences

For a reference copy of the document with all sections, see nature.com/documents/nr-reporting-summary-flat.pdf

# Life sciences study design

All studies must disclose on these points even when the disclosure is negative.

| | |
|---|---|
| Sample size | No pre-determined sample size was established. The number of experiments and sample size are indicated in Source data files. Sample size was determined based on previous studies in the field (Infante et al. Nat Commun 2018; Ferrari et al. Nat Commun 2019; Zagryazhskaya-Masson et al. JCB, 2020) |
| Data exclusions | No data were excluded. |
| Replication | All the experimental findings were reliably reproduced as validated by at least three independent experiments. |
| Randomization | This does not apply to the set of experiments presented in this paper where two distinct groups were generally compared and the order of analysis does not influence the experimental outcomes. |
| Blinding | The investigators were not blinded during data collection or analysis as the method of quantification over multiple replicates (for all experiments) and individual cells ensures unbiased processing. Additionally, blinding was not possible as the same researcher produced and analysed the data. |

# Reporting for specific materials, systems and methods

We require information from authors about some types of materials, experimental systems and methods used in many studies. Here, indicate whether each material, system or method listed is relevant to your study. If you are not sure if a list item applies to your research, read the appropriate section before selecting a response.

## Materials & experimental systems

| n/a | Involved in the study |
|---|---|
| ☐ | ☒ Antibodies |
| ☐ | ☒ Eukaryotic cell lines |
| ☒ | ☐ Palaeontology and archaeology |
| ☒ | ☐ Animals and other organisms |
| ☒ | ☐ Clinical data |
| ☒ | ☐ Dual use research of concern |

## Methods

| n/a | Involved in the study |
|---|---|
| ☒ | ☐ ChIP-seq |
| ☒ | ☐ Flow cytometry |
| ☒ | ☐ MRI-based neuroimaging |

## Antibodies

| | |
|---|---|
| Antibodies used | Exhaustive description of antibodies used in this study are listed in Methods section and Extended Data Table 2. |

Validation

All commercially available primary antibodies were validated for the use in westernblotting or immunofluorescence using siRNA approach.

Anti-TKS5, Cell Signaling Technology (#16619), validated for Western Blot and immunofluorescence analyses, relevant citations can be found on the manufacturer's website (https://www.cellsignal.com/products/primary-antibodies/tks5-antibody/16619).

Anti-MT1-MMP, Millipore (#3328), validated for Western Blot and immunofluorescence analyses, relevant citations can be found on the manufacturer's website (https://www.merckmillipore.com/FR/fr/product/Ms-X-MMP-14-Antibody-clone-LEM-2-15.8-MT1-MMP,MM_NF-MAB3328-25UG?ReferrerURL=https%3A%2F%2Fwww.google.com%2F).

Anti-Caveolin-1/Cav1, Cell Signaling Technology (#3238), validated for Western Blot and immunofluorescence analysis, relevant citations can be found on the manufacturer's website (https://www.cellsignal.com/products/primary-antibodies/caveolin-1-antibody/3238).

Anti-Cavin1, Abcam (#ab48824), validated for Western Blot and immunofluorescence analysis, relevant citations can be found on the manbetafacturer's website (https://www.abcam.com/products/primary-antibodies/ptrf-antibody-ab48824.html).

Anti-N-WASP, Cell Signaling Technology (#4848), validated for Western Blot analysis, relevant citations can be found on the manufacturer's website (https://www.cellsignal.com/products/primary-antibodies/n-wasp-30d10-rabbit-mab/4848).

Anti-beta1 integrin, BioLegend (#303036), validated for Western Blot and immunofluorescence analyses, relevant citations can be found on the manufacturer's website (https://www.biolegend.com/nl-nl/products/ultra-leaf-purified-anti-human-cd29-antibody-19166?GroupID=BLG10310).

Anti-beta3 integrin, Cell Signaling Technology (#13166), validated for Western Blot analysis, relevant citations can be found on the manufacturer's website (https://www.cellsignal.com/products/primary-antibodies/integrin-b3-d7x3p-xp-rabbit-mab/13166).

Anti-EHD2, Santa Cruz Biotechnology (#sc-100724), validated for Western Blot analysis, relevant citations can be found on the manufacturer's website (https://www.scbt.com/fr/p/ehd2-antibody-l-05).

Anti-Cortactin, Merck Millipore (#05-180), validated for Western Blot and immufluorescence analyses, relevant citations can be found on the manufacturer's website (https://www.merckmillipore.com/FR/fr/product/Anti-Cortactin-p80-85-Antibody-clone-4F11,MM_NF-05-180-I-100UL).

Anti-Col1-3/4C, ImmunoGlobe GmbH (#0217-050), validated for immufluorescence analysis, relevant citations can be found on the manufacturer's website (https://www.immunoglobe.com/antibodies/items/collagen_cleavage_site.html).

GAPDH, Santa Cruz Biotechnology (#sc-25778), validated for Western Blot analysis, relevant citations can be found on the manufacturer's website (https://www.scbt.com/fr/p/gapdh-antibody-fl-335).

Anti-CD63, BD Biosciences (#556019), validated for Western Blot and immunofluorescence analyses, relevant citations can be found on the manufacturer's website (https://www.bdbiosciences.com/en-at/products/reagents/flow-cytometry-reagents/research-reagents/single-color-antibodies-ruo/purified-mouse-anti-human-cd63.556019).

Anti-Rab5, Cell Signaling Technology (#3547), validated for Western Blot and immunofluorescence analyses, relevant citations can be found on the manufacturer's website (https://www.cellsignal.com/products/primary-antibodies/rab5-c8b1-rabbit-mab/3547).

Anti-Rab7, Cell Signaling Technology (#9367), validated for Western Blot and immunofluorescence analyses, relevant citations can be found on the manufacturer's website (https://www.cellsignal.com/products/primary-antibodies/rab7-d95f2-xp-rabbit-mab/9367).

Anti-GFP, Abcam (#ab13970), validated for Western Blot and immunofluorescence analyses, relevant citations can be found on the manufacturer's website. (https://www.abcam.com/products/primary-antibodies/gfp-antibody-ab13970.html).

Anti IgG-rabbit-A488, Molecular Probes (#A11034), validated for immunofluorescence analysis, relevant citations can be found on the manufacturer's website (https://www.thermofisher.com/antibody/product/Goat-anti-Rabbit-IgG-H-L-Highly-Cross-Adsorbed-Secondary-Antibody-Polyclonal/A-11034).

Anti-IgG-rabbit-Cy3, Jackson ImmunoResearch (#711-165-152), validated for immunofluorescence analysis, relevant citations can be found on the manufacturer's website (https://www.jacksonimmuno.com/catalog/products/711-165-152).

Anti-IgG-mouse-A488, Molecular Probes (#A21202), validated for immunofluorescence analysis, relevant citations can be found on the manufacturer's (https://www.thermofisher.com/antibody/product/Donkey-anti-Mouse-IgG-H-L-Highly-Cross-Adsorbed-Secondary-Antibody-Polyclonal/A-21202).

Anti-IgG-mouse-Cy3, Jackson ImmunoResearch (#711-165-151), validated for immunofluorescence analysis, relevant citations can be found on the manufacturer's website (https://www.jacksonimmuno.com/catalog/products/715-165-151).

Anti-IgG-mouse-647, Molecular Probes (#A31571), validated for immunofluorescence analysis, relevant citations can be found on the manufacturer's website (https://www.thermofisher.com/antibody/product/Donkey-anti-Mouse-IgG-H-L-Highly-Cross-Adsorbed-Secondary-Antibody-Polyclonal/A-31571).

Anti-AlexaFluor488 phalloidin, Molecular Probes (#A12379), validated for immunofluorescence analysis, relevant citations can be found on the manufacturer's website (https://www.thermofisher.com/order/catalog/product/fr/fr/A12379).

Anti-AlexaFluor546 phalloidin, Molecular Probes (#A22283), validated for immunofluorescence analysis, relevant citations can be

found on the manufacturer's website (https://www.thermofisher.com/order/catalog/product/fr/fr/A22283).

Anti-IgG-mouse-HRP, Jackson ImmunoResearch (#115-035-062), validated for Western Blot analysis, relevant citations can be found on the manufacturer's website (https://www.jacksonimmuno.com/catalog/products/115-035-062).

Anti-IgG-rabbit-HRP, Jackson ImmunoResearch (#111-035-045), validated for Western Blot analysis, relevant citations can be found on the manufacturer's website (https://www.jacksonimmuno.com/catalog/products/111-035-045).

# Eukaryotic cell lines

Policy information about cell lines and Sex and Gender in Research

| | |
|---|---|
| Cell line source(s) | Human MDA-MB-231 breast adenocarcinoma cells obtained from ATCC (ATCC HTB-26). Human HT-1080 fibrosarcoma cells were obtained from ATCC (ATCC CCL-121). |
| Authentication | The two cell lines were not further authenticated. |
| Mycoplasma contamination | All cell lines used in this study are routinely tested for Mycoplasma contamination using a PCR-based approach and were negative for mycoplasma contamination. |
| Commonly misidentified lines (See ICLAC register) | No commonly misidentified cell lines were used in this study. |

