## [Peer Review File · Nature Cell Biology]

Peer Review Information

Journal: Nature Cell Biology

Manuscript Title: A mechanosensitive caveolae-invadosome interplay drives matrix remodelling for cancer cell invasion

Corresponding author name(s): Dr Philippe Chavrier

Editorial Notes:

Reviewer Comments & Decisions:

Decision Letter, initial version:
--

Dear Dr Chavrier,

I apologize for the delay. Your manuscript "A mechanosensitive caveolae-invadopodia interplay drives matrix remodeling for cancer cell invasion", has now been seen by 3 referees, who are experts in caveolae (referee 1); integrin trafficking (referee 2); and cancer cell invasion (referee 3), and whose comments are pasted below. In light of their advice, we regret that we cannot offer to publish the study in Nature Cell Biology.

As you will see, although the reviewers find this work interesting, they raise serious concerns that question the conceptual advance that these findings represent over previous work, and the strength of the data and of the novel conclusions that can be drawn at this stage. In particular, among the limitations of the dataset the referees note that the potential novel mechanistic insights are not supported and are under-developed, including unclear support for proposed mechanisms between forces, collagen fibres, membrane dynamics, and integrin trafficking.

We note that you have opted out of consultations for potential transfer. However, we would be happy to consult with our colleagues at Nature Communications and Communications Biology to see whether they would be interested in taking this manuscript further with the existing peer review history. Please do let me know if you would like us to pursue this option.

We are very sorry that we could not be more positive on this occasion, but we thank you for the opportunity to consider this work.

With kind regards,
Daryl Jason David

Daryl J.V. David, PhD

Senior Editor, Nature Cell Biology
Consulting Editor, Nature Communications
Nature Portfolio

Heidelberger Platz 3, 14197 Berlin, Germany
Email: daryl.david@nature.com
ORCID: <https://orcid.org/0000-0002-9253-4805>

Reviewers' comments:

Reviewer #1 (Remarks to the Author):

Comments for Monteiro et al

This paper describes the possible role of caveolae and the cellular invagination having matrix metalloproteinase for extracellular matrix remodeling for cancer cell metastasis, investigating the role of caveolae in invadopodia formation and managing to fill the gap where caveolae were reported to be the essential regulator of invadopodia formation. The hypothesis and the model that the authors propose are novel and interesting. However, several points are better to be addressed.

1. The constriction of the ECM fibers is assumed from the arc-like configuration of the ECM fibers. I am not sure the shape alone can suggest the force applied to the ECM for constriction. The ECM would be caught by integrin, and then the ECM would be bent by force generated by the cells. Then, the ECM fibers would be linear in the absence of these. The linearity of ECM fibers should be quantified in all the figures.
2. Several marker proteins, which are N-WASP, cortactin, and Tks5, of invadopodia were used but not consistent. The protease activity appears to be detected by the collagen cleavage antibody (Col1-3/4C), and the co-localization of this collagen cleavage and the invadopodia markers will be useful information.
3. Related to 2, whether this marker protein could generally represent the invadopodia at all the transition of invadopodia from precursors or to caveolin-1-containing structures is not clear and should be investigated.
4. Figure 1, the line scan needs the statistics of reproducibility. Auto-correlation co-efficient average among various observations would be one of the statistics.
5. Figure 2, whether the thick fibers are collagen fibers or not, would need confirmation. Caveolae are sometimes on thick stress fibers, and it would be challenging to be distinguished them from Fibrils beneath the membrane.
6. Figure 5. The presence of ECM seemed to promote invadopodia formation. Can ECM also promote the clearance of surface integrin-mediated by caveolae due to the upregulated invadopodia-mediated collagen cleavages? Can the nanobar with gelatin can be stained with the antibody for collagen cleavage?
7. They also revealed rather mutually exclusive localization of caveolin-1 and integrin with invadopodia, which is in contrast to the studies (Yamaguchi et al, 2009, which is cited), where the endogenous caveolin-1 was colocalized with MT1-MMP. Is there any transient form of the mixture of MT1-MMP and caveolin-1 in the ultrastructure level? Because model cartoon in Fig 8 suggests the transit form, but it would need to be clarified in the data.

Minor points

In all figures, the blue representation of collagen fibers is sometimes not easy to see and should be modified.

Figure 3 c and d. The amount of collagen cleavage by siTKS5 and siCav1 in the bar chart and the images looks to be the opposite.

Figure 5 c. The groups seemed to have no difference.

Figure 5b, it is not clear where the cells locate.

Reviewer #2 (Remarks to the Author):

Cancer cell interactions with the extracellular matrix is a key determinant in cancer cell migration, invasion and dissemination. Type I collagen is the major stromal component and hence it is important to understand how cells interact with complex collagen fibrillary matrixes and hoe these fibrils are remodeled to facilitate cancer cell invasion. In this study, Monteiro and co-workers have identified interesting adhesive/degradative structures which are assembled on collagen fibers and involved in their remodeling. The authors use a combination of high- and super-resolution microscopy methods in conjunction with functional studies. They show that caveolae and MT1-MMP1 containing invadopodia are recruited in a periodic pattern to the fibrils with these two different structures residing adjacent to each other on the fibril. These structures are sensitive to b1 integrin depletion. Furthermore, depletion of invadopodia components result in accumulation of the caveolae structures along the fibrils indicating that proteolytic turnover of collagen is linked to their dynamics. The authors suggest that these structures and MT1-MMP activity is involved in bending and softening of the fibrils and propose that this mechanism may be relevant in cancer cell migration in collagen rich matrixes.

This is an interesting study. The manuscript is well-written, the data are of high quality and the data support the conclusions to a large extent.

One major concern is that the current data provide very little (if any) demonstration that the cells actively generate these bent CI fibers and that their generation is dependent on the interplay of the two structures on the filament. The data convincingly show that these structures are generated on these peculiar looking curved (sometimes even circular) collagen fibers. However, it is unclear if these are specifically formed by the cells under the cells (and areas elsewhere would have liner fibers). There is also no data currently that would directly support the interesting suggestion that invadopodia soften the fibers resulting in their bending and matrix remodeling.

The other major concern is that all the siRNA experiments are performed with single oligos and lack off-target controls. This must be addressed as the off-target effects of siRNAs are prevalent may result in faulty conclusions.

Additional specific points:

- 1) Figure 1d,e please show that the bending specifically links to these structures
- 2) Figure 3a-d and Figure 4d,e. Do the siRNAs/drugs that interfere with the invadopodia also abolish fiber bending? It would be important to show images of the collagen channel alone in cells covered and cell- free areas and ideally provide some live imaging either here or linked to the point below (Figure 4 f,h)
- 3) Figure 4 f, h. I struggle to find direct evidence for the authors claim " along with conversion of straight fibers into curved ones" page 10.
- 4) Figure 5 b,c why was gelatin used instead of collagen?
- 5) Figure 5g. How do the cells adhere to collagen in the absence of all collagen binding integrins? This experiment is a little hard to understand
- 5) Please show images of the endocytosed b1-integrin co-stained with markers for the endolysosomes

(CD63, Rab7) alongside markers for early endosomes (as numerous studies have indicated integrins traffic via this compartment regardless of the initial entry routes). Is the endolysosomal localization due to the long 60 min incubation? Are the integrins degraded rather than recycled? This should at least be discussed.

6) Figure 7f. Is the ability of TKS5 OE to drive PM accumulation of $\beta 1$ -integrins specific to collagen fiber ECM and the formation of these structures described here or does this occur also on other matrixes?

Minor: Abstract is there a word missing from this sentence "caveolae and collagen $\beta 1$ -integrins co-localize"

The authors should mention the collagen cleavage in the text in the same order as the figures. Now it is mentioned already on page 6.

Reviewer #3 (Remarks to the Author):

Beta1 integrin in association with caveolar structures at adhesion sites to fibrillar collagen was reported earlier by the Montagnac group (PMID: 28619886; Ref. 66). In this work, Monteiro and coworkers report that caveolin structures are juxtaposed with TKS5 and cortactin-rich focalized and linear zones, which alternate in sequence along collagen fibrils. Using high-quality confocal and superresolution microscopy and RNA interference, they show that formation of these structures and associated perinuclear collagen degradation depend both on Cav1 and TKS5 or cortactin. Similarly, they show that caveolin-positive foci and collagenolysis depend on actin polymerization and branching, using pharmacological inhibitors and further demonstrate that caveolae-mediated adhesion precedes collagenolysis. Lastly, they show that caveoli-formation supports the recruitment and internalization of beta1 integrins. The authors propose a sequential model by which caveoli first form from an adhesive bridge to collagen fibrils, which enables MT1-MMP recruitment, collagenolysis and turnover of adhesion and ECM by internalization.

The quality of the cell biological experiments showing juxtaposition of caveolae and proteolytic zones is very high. The sequential development of first caveoli followed by TKS5-positive and, in part, collagenolytic structures is convincingly shown, as the requirement of caveoli for collagenolysis.

It is well established that in 2D and 2.5D models perturbation of cell adhesion can negatively impact ECM degradation, but the existence and mechanistic relevance of the proposed step-wise process in 3D invasion remains to be demonstrated. In addition, the mechanism of phase separation into curved and (non-curved?) lytic domains remains unexplored. Because caveoli provide integrin-mediated interaction to collagen (see Ref 66), it may be considered as a potentially trivial finding that also MT1-MMP mediated collagenolysis is impaired, because a 2.5D system has been used here in which cell-collagen interaction depends on active adhesion.

In conclusion, whereas the relevance of caveoli in mediating focal and linear tumor cell interactions to collagen fibers is confirmed here, the mechanistic advance in how these structures control collagenolysis, other than by mediating adhesion, remains unclear.

1. Figures 1, 2 and 5 rehash Montagnac et al. (Ref 66) with additional detail on topology. The mechanisms and relevance of the juxtaposition of caveoli and lytic zones have not been mechanistically explored. Therefore, the authors may want to move the confirmatory parts of these

Figs. to the supplement, and shorten the text accordingly.

2. The terminology is confusing and poorly motivated by experimental evidence. Cortactin and TKS5 are classical focal adhesion markers, however the authors refer to “invadopodia” throughout this manuscript, without specific justification on how focal adhesion sites, invadopodia and putative other collagenolytic structures have been discriminated. The data in Suppl. Fig. 1 lack clear demonstration where collagen fibers are actually degraded, and where not. Recruitment of MT1-MMP to substrate may occur with or without collagenolysis (e.g., PMID 17618273), but this distinction is lacking here. In Fig. 3, it appears that at least a small subset of these contacts is collagenolytic, however in peripheral regions of the cell, cortactin-rich and MT1-MMP-positive foci occur in the absence of Cav1 and collagen degradation. Should these events be considered as focal adhesions/contacts or invadopodia? Are the morphological criteria (e.g., protrusion formation) fulfilled to justify classification as invadopodia at any of these sites? In Fig. 2, SEM images are inconclusive on morphology. Can sites, with or without evidence for MT1-MMP localization, active proteolysis and/or protrusion (“pod”) be considered “invadopodia”? To avoid unwanted bias, the authors should include stringent reporting on the proteolytic sites (e.g., the fraction of Cav1 and TKS5-foci with or without collagenolysis), and report lytic and nonlytic subdomains throughout their interventions. Alternatively, the terminology should be altered to neutral terms, such cortactin-rich adhesions with or without collagenolytic activity. In case the majority or all of regions described here are indeed focal contacts/adhesions, that are required to bring MT1-MMP in sufficient proximity to collagen fibrils, the conceptual advance compared to the work by Montagnac is not clear.

3. The interventions shown in Fig. 4 with the actin system should also include the effect on collagenolysis, to link adhesion topologies to actual ECM degradation.

4. Fig. 5 seems disconnected, since here a biophysical mechanism of membrane bending is established, without further exploration of the relevance for MT1-MMP activity and collagenolysis. Can the membrane bending explain exclusion of MT1-MMP positioning and/or lytic activity?

5. The link between integrin internalization and collagenolysis remains unexplored. Whereas caveoli appear to form an effective adhesive structure, with integrin clustering and turn-over by internalization, the link of these adhesion dynamics to MT1-MMP positioning and function remains unexplored. The authors speculate that this may aid ECM internalization, without further support by data. Can internalization of ECM be shown, and does this preferentially occur at the interface between both domains, as would be predicted by their topologic model?

6. The relevance of the here described link between caveoli and collagenolysis for tumor cell invasion in 3D models (Fig. 7h, i) is rudimentary. Can the same substructures and domain segregation be demonstrated? Are the effects of Cav1 or TKS5 downregulation dependent on reduced adhesion or compromised collagenolysis, or both?

Minor points

The sequence of events in Fig. 1e is difficult to appreciate from still images. Likewise, the sequence of integrin and caveolin internalization (Fig. 6) is only represented by static images. The primary videos should be included, to showcase dynamic events.

“These results suggest a potential plasticity and conformational adaptation of caveolae to the

underlying fibril topology to adopt a clustered organization, possibly involved in pinching of and/or grabbing the fibrils.” – Ref 66 should be cited here. Likewise, the role of caveolae in cell adhesion to fibrillar collagen should be cited in the introduction.

**I suggest that you consider Nature Communications as a suitable venue for your work. To transfer your manuscript there, please use our manuscript transfer portal. You will not have to re-supply manuscript metadata and files, unless you wish to make modifications, but please note that this link can only be used once and remains active until used. For more information, please see our manuscript transfer FAQ page.

Note that any decision to opt in to In Review at the original journal is not sent to the receiving journal on transfer. You can opt in to In Review at receiving journals that support this service by choosing to modify your manuscript on transfer. In Review is available for primary research manuscript types only.

**For Nature Portfolio general information and news for authors, see <http://npg.nature.com/authors>

Author Rebuttal to Initial comments

NCB-A49883 – Monteiro *et al.* - A mechanosensitive caveolae-invadopodia interplay drives matrix remodeling for cancer cell invasion

Detailed plans and point-by-point response

Reviewer #1 (Remarks to the Author):

Comments for Monteiro et al

*This paper describes the possible role of caveolae and the cellular invagination having matrix metalloproteinase for extracellular matrix remodeling for cancer cell metastasis, investigating the role of caveolae in invadopodia formation and managing to fill the gap where caveolae were reported to be the essential regulator of invadopodia formation. **The hypothesis and the model that the authors propose are novel and interesting.** However, several points are better to be addressed.*

1. The constriction of the ECM fibers is assumed from the arc-like configuration of the ECM fibers. I am not sure the shape alone can suggest the force applied to the ECM for constriction. The ECM would be caught by integrin, and then the ECM would be bent by force generated by the cells. Then, the ECM fibers would be linear in the absence of these. The linearity of ECM fibers should be quantified in all the figures.

Whether cells can bend collagen fibers and the respective contribution of caveolae and invadopodia to matrix bending are critical issues, which were discussed by all three expert referees. We propose to perform a more thorough analysis of the cell effects on ECM bending and remodeling. This analysis will be based on TWOMBli (The Workflow Of Matrix BioLogY Informatics), an ImageJ plugin recently designed by E. Sahai (The Francis Crick Institute, London, UK) and colleagues, which provide descriptors and metrics of the matrix network, including matrix alignment (<https://github.com/wershofe/TWOMBli>). The alignment metric calculated by TWOMBli captures the extent to which fibers within the field of view are oriented in a similar direction. The fiber alignment metric is a value in the range [0, 1], where [0] represents complete isotropy and [1] represents perfect alignment (see Wershof et al. 2021, DOI: 10.26508/lsa.202000880).

The rationale of the proposed analysis is that fiber alignment should change locally under the cells due to invadopodia function and interplay with caveolae (see **Figure 1a**). As a proof-of-concept, we used the TWOMBli plugin to compare fiber alignment in cell-free areas of the collagen network (**Figure 1bc**, cell-free), vs. areas underneath the cells 30 min and 60 min after plating on the type I collagen network. **Figure 1c** shows a time-dependent decrease of fiber alignment underneath the cells, illustrating the cells' capacity to bend the collagen fibers and remodel the network. Moreover, we measured alignment under cells silenced for Cav1 or TKS5 as compared to cells treated with a non-targeting siRNA, and we found that the cells' capacity to bend the fibers was diminished in agreement with our finding that silencing of Cav1 impaired invadopodia formation and function, thereby validating the Alignment metric and TWOMBli-based analysis (**Figure 1d**). TWOMBli-based analysis will be performed under the different conditions tested in our study (siRNAs, drugs...) to understand better the contribution of caveolae and invadopodia to matrix bending.

Figure 1. (a) Example input image showing an MDA-MB-231 cell stained for cortactin (red) plated for 60 min on top the type I collagen matrix (blue). The collagen network only is shown with a RedHot LUT in the right image and the area used to measure fiber alignment under the cell is boxed (cyan). (b) Segmentation of the collagen network by TWOMBLI corresponding to the boxed region in panel (a) to generate a collagen mask used to calculate the Alignment metric. (c) The plot shows the Alignment metric for cell-free regions of the collagen network and regions underneath MDA-MB-231 cells plated for 30 or 60 min on top of the collagen network. (d) The plot compares the Alignment metric of the cell-associated fibers for cells treated with non-targeting siRNA or treated with Cav1 or TKS5 siRNAs.

2. Several marker proteins, which are N-WASP, cortactin, and Tks5, of invadopodia were used but not consistent. The protease activity appears to be detected by the collagen cleavage antibody (Col1-3/4C), and the co-localization of this collagen cleavage and the invadopodia markers will be useful information.

Association of N-WASP, cortactin and Tks5 with invadopodia has been extensively described by us and others (Seals *et al.* 2005 PMID: 15710328). In addition, we have already reported the co-localization of these invadopodial markers with cleaved collagen labeled with Col1-3/4C antibody:

- TKS5^{GFP} – Col1-^{3/4}C: Ferrari *et al.* (Ref.13 in the manuscript); Zagryazhskaya-Masson *et al.* (Ref.14); Colombero *et al.* (Ref.65)

- F-actin - Col1-^{3/4}C: Monteiro *et al.* (Ref.46)

- Cortactin - Col1-^{3/4}C: Zagryazhskaya-Masson *et al.* (Ref.14); Colombero *et al.* (Ref.65)– see example in **Figure II** below.

Figure II. MDA-MB-231 cells plated for 60 min on a layer of fluorescently-labeled type I collagen (blue scale) and fixed and stained for cortactin (red scale) and cleaved collagen with Col1-3/4C antibody (grey scale).

Furthermore, in the present manuscript, we provide additional evidence for co-localization of N-WASP with TKS5^{GFP} along collagen fibers in MDA-MB-231 cells (Ext. Data Fig.7a). Co-staining for endogenous N-WASP and collagen cleavage (Col1-^{3/4}C) could not be performed because antibodies are both rabbit pAb.

3. Related to 2, whether this marker protein could generally represent the invadopodia at all the transition of invadopodia from precursors or to caveolin-1-containing structures is not clear and should be investigated.

The conclusion that invadopodia mature from non-degradative structures to degradative ones is mostly based on published data using gelatin as a matrix construct. There is very little data confirming this transition in the context of a more physiological fibrillar collagen matrix. We agree that it would be interesting to look at this transition further in the context of the caveolae-invadopodia mechanism we are proposing. However, collagen cleavage and caveolae are both best detected with polyclonal antibodies raised in rabbit making it difficult to assess collagenolysis at the same time as caveolar structures.

4. Figure 1, the line scan needs the statistics of reproducibility. Auto-correlation co-efficient average among various observations would be one of the statistics.

Line scans in Figure 1 were used to show alternation of Caveolin-1 (Cav1)-positive and cortactin-positive structures along collagen fibers (Fig.1b,) and to reveal a preferential association of Cav1 (and cavin1)-positive structures with straight segments of the collagen fibers while cortactin (and TKS5)-positive structures are associated with bent regions (Fig.1d).

Related to statistical robustness of co-localization data (comparison of Cav1 and cortactin accumulation along fibers in Fig.1b), we are providing below a plot showing Cav1 and cortactin intensity for each pixel from 16 independent line scans taken along collagen fibers (**Figure IIIa**). In addition, we also plotted the corresponding Pearson correlation coefficient (**Figure IIIb**). These analyses clearly demonstrate a negative correlation between the caveolae and invadopodia markers. In addition to the line scan provided in Fig.1b of the manuscript, these new data will strengthen the conclusion that caveolae (Cav1) and invadopodia (cortactin) markers alternate along fibers.

Figure III. (a) Correlation of cortactin (x axis) and Cav1 (y axis) pixel fluorescence intensity along 16 line scans taken along collagen fibers with associated alternating caveolae and invadopodia structures. **(b)** Pearson correlation coefficient for Cav1 and cortactin markers corresponding to the 16 line scans analyzed in panel (a) and table of corresponding *p*-values.

Related to Fig.1d (association of caveolae and invadopodium markers with straight vs. bent fiber segments, respectively), we analyzed the intensity of Cav1 and cortactin markers for each pixel along several line scans (N=7) drawn along fibers and calculated a cortactin intensity/Cav1 intensity ratio for each pixel. Pixels were then stratified in two classes based on Cortactin/Cav1 intensity ratio >1 (i.e. cortactin-enriched invadopodia, red in **Figure IV**) and Cortactin/Cav1 intensity ratio <1 (i.e. cav1-enriched caveolae, blue in **Figure IV**). The frequency distribution of curvature (μm^{-1}) for each pixel class was plotted showing that cav1-enriched structures (caveolae) are associated with low curvature, in contrast to cortactin-enriched structures (invadopodia) that are associated with higher curvature. In addition to the line scan provided in Fig.1d of the manuscript, these new data will strengthen the conclusion that caveolae associate with straight fiber segments, while invadopodia are associated with bent regions of the fibers.

Figure IV. Frequency distribution of curvature (μm^{-1}) of pixels with Cortactin/Cav1 intensity ratio >1 (i.e. cortactin-enriched invadopodia, red) and Cortactin/Cav1 intensity ratio <1 (i.e. cav1-enriched caveolae, blue).

5. Figure 2, whether the thick fibers are collagen fibers or not, would need confirmation. Caveolae are sometimes on thick stress fibers, and it would be challenging to be distinguished them from Fibrils beneath the membrane.

The identification of the electron-dense elongated material (appearing with light shades in the inverted greyscale image in Fig.2ab and in Ext. Data Fig.2) as bundles of collagen fibers underneath the ventral plasma membrane of unroofed cells is based on our previous published work (Ferrari et al. 2019 - Ref.13 of the manuscript),

performed in collaboration with Dr S. Vassilopoulos, a world expert in platinum replica electron microscopy (PREM) and co-author of the present study. In addition, during this study, we performed correlative light microscopy and PREM analysis, which was not included in the present manuscript, and that is provided below (**Figure V**). The image shows the overlap of fluorescence signal of Cy5-labeled collagen fibers with elongated electron-dense structures, which can thus be identified undoubtedly as collagen fibers. In the classical PREM images shown in Fig.2ab of the manuscript, Cav1^{GFP} and Cavin1^{GFP}-positive structures are associated with the same elongated electron-dense material identified as collagen fibers. Of note, for technical reason it was not possible to perform correlative light microscopy and immunogold PREM analysis that was required to identify Cav1 and Cavin1-positive structures.

Figure V. The left image is a PREM view of the ventral plasma membrane of an unroofed MDA-MB-231 cell plated for 60 min on a layer of type I collagen. The central fluorescence image shows the underlying layer of Cy5-labeled collagen I fibrils (blue). The right image is the correlative fluorescence microscopy/PREM overlay image showing that the elongated electron-dense structures with associated caveolae clusters correspond to the collagen fibers underneath the plasma membrane.

6. *Figure 5. The presence of ECM seemed to promote invadopodia formation. Can ECM also promote the clearance of surface integrin-mediated by caveolae due to the upregulated invadopodia-mediated collagen cleavages? Can the nanobar with gelatin can be stained with the antibody for collagen cleavage?*

We could not assess gelatin degradation on the nanobar substratum using the Col1-³C reagent as this antibody does recognize only fibrillar cleaved collagen (gelatin is non fibrillar denatured collagen). The main rationale of the nanobar chip assay was to generate plasma membrane deformations with size like the dimension of the collagen fiber bundles but with a simpler and fixed geometry. In this regard, coating the nanobar array with fibrillar collagen would be technically challenging and would result in the superimposition of a network of randomly oriented collagen fibers and the simple geometry of parallel nanobar array generating unwanted, unnecessary complexity.

7. *They also revealed rather mutually exclusive localization of cavelin-1 and integrin with invadopodia, which is in contrast to the studies (Yamaguchi et al, 2009, which is cited), where the endogenous cav-1 was colocalized with MT1-MMP. Is there any transient form of the mixture of MT1-MMP and caveolin-1 in the ultrastructure level? Because model cartoon in Fig 8 suggests the transit form, but it would need to be clarified in the data.*

Yamaguchi and colleagues did not actually assess the localization of endogenous Cav1 and MT1-MMP but looked at the distribution of overexpressed mCherry-tagged Cav1 and GFP-tagged MT1-MMP. Moreover, Yamaguchi et al., 2009 provided only a single image of a single cell showing limited overlap between the two overexpressed proteins on some unidentified cellular structures with no information of the overlap of the markers with matrix (gelatin) degradation. In addition, this study did not provide any quantitative analysis of the overlap between these markers (Fig.5b,c). In contrast, our study provides a detailed quantitative analysis of the co-localization of several caveolae and invadopodia markers, including data regarding the distribution of endogenous Cav1 and MT1-MMP (Ext. Data Fig.6a). In addition, Yamaguchi and colleagues showed endogenous Cav1 alone (no counter labeling) associated with some areas of gelatin degradation that could correspond to invadopodia although there was no additional marker to ascertain their conclusion (Yamaguchi et al., Fig.4a).

Although, we mostly used a network of collagen fibers as a more physiological matrix (as compared to a uniform thin coating of gelatin, i.e. denatured collagen), we also looked at – but did not show – Cav1 distribution in cells

plated on gelatin. These experiments revealed that Cav1-positive puncta are associated to focal areas of gelatin degradation (appearing as dark areas). Similar to our observations based on the fibrillar collagen network documented in our paper, Cav1 did not co-localize with invadopodial cortactin-positive puncta forming in association with degraded gelatin spots, but Cav1 and cortactin-positive puncta were rather juxtaposed to each other, in line with our reported observations made using the collagen fiber network (see examples provided in **Figure VI** below).

Figure VI. MDA-MB-231 cells plated for 60 min on Oregon488-gelatin (blue). Cells were fixed and stained for Cav1 (cyan) and cortactin (red). Insets rows correspond to the boxed regions in the lower magnification views.

Finally, the referee raised a point regarding the “mixture of MT1-MMP and caveolin-1 in the ultrastructure level” based on our model (Fig.8). In that respect, we acknowledge that some confusion may have been generated by the limitation of our sketched model (Fig.8 of the manuscript) to represent the complex 3D organization of the structures and their dynamic relationship using a flat scheme. We have now collaborated with an expert in scientific drawing who corrected the model to represent the complexity of the structures and illustrate their sequentiality better (see **Figure VII**). The new scheme will be included in the revised manuscript.

Figure VII. Model of mechanosensitive caveolae-invadopodia interplay in matrix remodeling. Caveolae are present as invaginated pits at the plasma membrane. When cultured on top of a meshwork of type I collagen fibrils, cancer cells harbor elongated clusters of Cav1/Cavin1-positive caveolar structures along PM-collagen

fibrils contact sites, in a process dependent on membrane curvature, collagen-binding integrins (i.e. beta1 integrin) and matrix stiffness (1). Invadopodia form in the vicinity of and alternate with these caveolar clusters (2). Upon remodeling and weakening of the collagen fibrils through invadopodia action, caveolae mediate beta1 integrin clearance from the cell surface (3), along with ECM internalization and transport to endolysosomes (4), from where some of these components could be recycled and delivered to newly formed matrix-PM contact sites as cell moves through the matrix.

Reviewer #2 (Remarks to the Author):

Cancer cell interactions with the extracellular matrix is a key determinant in cancer cell migration, invasion and dissemination. Type I collagen is the major stromal component and hence it is important to understand how cells interact with complex collagen fibrillary matrixes and hoe these fibrils are remodeled to facilitate cancer cell invasion. In this study, Monteiro and co-workers have identified interesting adhesive/degradative structures which are assembled on collagen fibers and involved in their remodeling. The authors use a combination of high- and super-resolution microscopy methods in conjunction with functional studies. They show that caveolae and MT1-MMP1 containing invadopodia are recruited in a periodic pattern to the fibrils with these two different structures residing adjacent to each other on the fibril. These structures are sensitive to b1 integrin depletion. Furthermore, depletion of invadopodia components result in accumulation of the caveolae structures along the fibrils indicating that proteolytic turnover of collagen is linked to their dynamics. The authors suggest that these structures and MT1-MMP activity is involved in bending and softening of the fibrils and propose that this mechanism may be relevant in cancer cell migration in collagen rich matrixes.

This is an interesting study. The manuscript is well-written, the data are of high quality and the data support the conclusions to a large extent.

- One major concern is that the current data provide very little (if any) demonstration that the cells actively generate these bent CI fibers

We would refer this referee to our previously published work providing a seminal analysis describing the fiber remodeling and bending activity of invadopodia based on live cell imaging, laser ablation experiments... (Ferrari et al. 2019 - Ref.13 of the manuscript). In brief, our previous study uncovered the combined action of invadopodial actin polymerization, which is involved in the generation of force that is applied against the fiber, and MT1-MMP-mediated proteolytic activity at invadopodia, which concomitantly weakens the fiber leading to fiber bending and formation of an invasion track (see also Monteiro et al. 2013 - Ref.46, Zagryazhskaya-Masson et al. 2020 - Ref.14)

Additionally, as discussed above regarding Point #1 raised by Referee#1, we propose to use the Alignment metric calculated by the TWOMBLI plugin to provide an unbiased measurement of the cell's capacity to bend the collagen fibers (see **Figure I**).

and that their generation is dependent on the interplay of the two structures on the filament.

This point is similar to Point #1 raised by Referee#1 and is well taken. As discussed above, we will perform a TWOMBLI-based analysis under the different conditions tested in our study (siRNAs, drugs...) to understand better the contribution of caveolae and invadopodia to matrix bending. Based on the proof-of-concept analysis shown in **Figure I** above we are confident the Alignment metric calculated by the TWOMBLI plugin will allow to decipher the specific contribution of caveolae vs. invadopodia activity or both to fiber bending.

The data convincingly show that these structures are generated on these peculiar looking curved (sometimes even circular) collagen fibers. However, it is unclear if these are specifically formed by the cells under the cells (and areas elsewhere would have liner fibers).

Here we refer to the proof-of concept analysis we performed using the TWOMBLI plugin to compare fiber alignment in cell-free areas of the collagen network (**Figure Ibc**, cell-free), vs. areas underneath the cells 30 min and 60 min after plating on the type I collagen network. **Figure Ic** shows a time-dependent decrease of fiber alignment underneath the cells as compared to the cell-free areas of the network showing higher value of fiber alignment. This analysis illustrates the cells capacity to bend the collagen fibers and remodel the network.

There is also no data currently that would directly support the interesting suggestion that invadopodia soften the fibers resulting in their bending and matrix remodeling.

As discussed above, the conclusion that invadopodia soften the fibers through MT1-MMP activity is based on our previous published work (see Ferrari et al. 2019 - Ref.13 of the manuscript; Monteiro et al., 2013 - Ref.46; and see Fig.7e of the manuscript), in which we used live cell imaging to look at the dynamics of TKS5^{GFP} invadopodia and the contacted fluorescently labeled collagen fibers overtime and laser ablation. It allowed us to report that invadopodia exert a pushing force towards the contacted fibers, which requires actin polymerization. This force pushes the fibers aside while maintaining a tight contact between the invadopodial cell surface and the pushed fibers. Upon inhibition of MT1-MMP activity, there is no such movement of the invadopodia/fiber ensemble as the counterforce exerted by non-cleaved, stiff fibers opposes the pushing force generated by actin polymerization at invadopodia (Ferrari et al. 2019 - Ref.13).

The other major concern is that all the siRNA experiments are performed with single oligos and lack off-target controls. This must be addressed as the off-target effects of siRNAs are prevalent may result in faulty conclusions.

As described in the Material and Methods section, most of the experiments in the article were performed with smartPools (i.e. mix of four siRNAs for MMT1-MMP/MMP14, TKS5, Cav1, beta1 integrin, beta3 integrin, N-WASP), which minimize off-target effects. It is only for a limited number of experiments describing the effect of Cavin1 KD that we used a single siRNA. We will add additional siRNAs or a SmartPool for Cavin1.

Additional specific points:

1) Figure 1d,e please show that the bending specifically links to these structures

Related to Fig.1d (association of caveolae and invadopodia markers with straight vs. bent fiber segments), we analyzed the intensity of Cav1 and cortactin markers for each pixel along line scans drawn along several fibers (N=7) and calculated the cortactin intensity/Cav1 intensity ratio for each pixel. Pixels were then stratified in two classes: Cortactin/Cav1 intensity ratio >1 (i.e. cortactin-enriched invadopodia, red in **Figure IV**, above) and Cortactin/Cav1 intensity ratio <1 (i.e. cav1-enriched caveolae, blue in **Figure IV**, above). The distribution of curvature (μm^{-1}) for each pixel class was plotted and clearly show that cav1-enriched structures (caveolae) are associated with low curvature, in contrast to cortactin-enriched structures (invadopodia) that are associated with high curvature.

2) Figure 3a-d and Figure 4d,e. Do the siRNAs/drugs that interfere with the invadopodia also abolish fiber bending? It would be important to show images of the collagen channel alone in ells covered and cell-free areas and ideally provide some live imaging either here or linked to the point below (Figure 4 f,h)

We refer to the proof-of concept analysis we performed using the TWOMBLI plugin to compare fiber alignment in cell-free areas of the collagen network (**Figure lbc**, cell-free), vs. areas underneath the cells 30 min and 60 min after plating on the type I collagen network (see above our reply to Point #1 raised by Referee#1). **Figure lc** shows a time-dependent decrease of fiber alignment underneath the cells as compared to cell-free areas of the network showing higher value of fiber alignment. This analysis illustrates the cells capacity to bend the collagen fibers and remodel the network.

3) Figure 4 f, h. I struggle to fins direct evidence for the authors claim “ along with conversion of straight fibers into curved ones” page 10.

We agree. This was an overstatement as no direct evidence is shown in our data. As mentioned above, quantification of collagen linearity will clarify this point.

4) Figure 5 b,c why was gelatin used instead of collagen?

Same as Referee #1/Point #6. The main rational of the nanobar chip assay was to generate plasma membrane deformations with size similar to collagen fiber bundles but with a simpler and fixed geometry. In this regard, coating the nanobar array with fibrillar collagen would be technically challenging and would most probably lead to the superimposition of a network of randomly oriented collagen fibers to the regularly spaced and parallel nanobar array generating unwanted complexity.

5) Figure 5g. How do the cells adhere to collagen in the absence of all collagen binding integrins? This experiment is a little hard to understand

First, even though integrin knockdown was efficient (see Ext. Data Fig.5), depletion was not complete and residual amount of beta1/beta3 integrins could provide sufficient cell adhesion capacity to the type I collagen network. Yet, we observed – but did not include in the manuscript - that cells depleted for integrins were smaller compared to control cells (see Figure VIII). Thus, silencing beta1/beta3 integrins had a direct effect on cell attachment to and cell spreading on the network of collagen fibers. In addition, we cannot exclude that, in the absence of beta1 or beta3 integrins, some compensatory mechanism may be involved based on another collagen receptor(s) (i.e. DDR1/2, MT1-MMP) known to bind type I collagen fibers. We will explain better this experiment.

Figure VIII. Quantification of cell area of MDA-MB-231 cells treated with non-targeting siRNA (NT) or with beta1 integrins siRNAs (β 1integrin) and plated for 60 min on type I fibrillar collagen.

5) Please show images of the endocytosed b1-integrin co-stained with markers for the endolysosomes (CD63, Rab7) alongside markers for early endosomes (as numerous studies have indicated integrins traffic via this compartment regardless of the initial entry routes). Is the endolysosomal localization due to the long 60 min incubation? Are the integrins degraded rather than recycled? This should at least be discussed.

We will provide data for endocytosed b1-integrin co-stained with markers for the endolysosomes (LBPA/Rab7/CD63) and early endosomes (EEA1). Some level of integrin degradation in the endolysosomal system is possible and will be discussed along with the possibility that integrins – and MT1-MMP – maybe recycled back to the surface from endolysosomes (Lobert et. 2010, PMID: 20643357; Dozynkiewicz et al. 2012, PMID: 22197222).

6) Figure 7f. Is the ability of TKS5 OE to drive PM accumulation of b1-integrins specific to collagen fiber ECM and the formation of these structures described here or does this occur also on other matrixes?

There is a misunderstanding from the referee as our data on the effect of TKS5 overexpression shows the opposite, i.e. less b1 integrins at the PM in cells overexpressing TKS5 (TKS5OE) (Fig.7f).

Reviewer #3 (Remarks to the Author):

Beta1 integrin in association with caveolar structures at adhesion sites to fibrillar collagen was reported earlier by the Montagnac group (PMID: 28619886; Ref. 66).

This is a main, recursive criticism from Referee #3 who is concerned that a previous article published by Montagnac and colleagues (PMID: 28619886; cited as Ref.66 in our manuscript) already reported the association of caveolae with fibrillar collagen. We respectfully disagree with this referee's statement. The article published by Montagnac and colleagues in 2019 did not report the association of caveolae with fibrillar collagen (there is no mention of caveolae in this article). This work established the association of clathrin-coated pits (CCPs) with collagen fibers in the environment. In addition, this study did not investigate or propose any kind of interplay between CCP-collagen association and matrix remodeling and invadopodia. There is no observational nor conceptual overlap between the two studies.

In this work, Monteiro and coworkers report that caveolin structures are juxtaposed with TKS5 and cortactin-rich focalized and linear zones, which alternate in sequence along collagen fibrils. Using high-quality confocal and superresolution microscopy and RNA interference, they show that formation of these structures and associated perinuclear collagen degradation depend both on Cav1 and TKS5 or cortactin. Similarly, they show that caveolin-positive foci and collagenolysis depend on actin polymerization and branching, using pharmacological inhibitors and further demonstrate that caveolae-mediated adhesion precedes collagenolysis. Lastly, they show that caveoli-formation supports the recruitment and internalization of beta1 integrins. The authors propose a sequential model by which caveoli first form from an adhesive bridge to collagen fibrils, which enables MT1-MMP recruitment, collagenolysis and turnover of adhesion and ECM by internalization.

The quality of the cell biological experiments showing juxtaposition of caveolae and proteolytic zones is very high. The sequential development of first caveoli followed by TKS5-positive and, in part, collagenolytic structures is convincingly shown, as the requirement of caveoli for collagenolysis.

It is well established that in 2D and 2.5D models perturbation of cell adhesion can negatively impact ECM degradation, but the existence and mechanistic relevance of the proposed step-wise process in 3D invasion remains to be demonstrated

The link between adhesion (integrins) and matrix degradation has been established using the 2D gelatin substratum model (i.e. denatured collagen), and data are missing in the context of fibrillar collagen, which is known to engage different receptors. In addition, how adhesion can control ECM degradation and invadopodia formation/activity is largely unknown even in the simple 2D gelatin model. Therefore, we are confident that our study that uncovers a new mechanism based on the interplay of caveolae and invadopodia during 2.5D and 3D invasion in fibrillar collagen is particularly relevant and timely.

In addition, the mechanism of phase separation into curved and (non-curved?) lytic domains remains unexplored

This point is well-taken and related to Point #1 from Referee #1 and several points raised by Referee #2, which we addressed above. In brief, based on a pilot analysis performed with the TWOMBLI ImageJ plugin (see **Figure I** provided in response to Point#1/Ref#1), we are confident that we will be able to analyze the effects of the different intervention (siRNAs and drugs) on the frequency of curved vs. non-curved fibers in order to refine the underlying mechanism of the transition from non-curved to curved cleaved segments.

Because caveoli provide integrin-mediated interaction to collagen (see Ref 66) Montagnac et al did not address caveolae (see above).

it may be considered as a potentially trivial finding that also MT1-MMP mediated collagenolysis is impaired, because a 2.5D system has been used here in which cell-collagen interaction depends on active adhesion.

We agree that there is a strong need to understand the interrelations between tumor cells with their matrix environment and how adhesion to the matrix can elicit a degradation response based on invadopodia formation and activity. These are precisely the questions that we are addressing in our study. The originality of our work is to provide the first, comprehensive mechanism, which integrates an adhesive function of caveolae to straight – possibly stiffer – portions of the matrix fibers, the induction of nearby invadopodia formation and activity, the subsequent degradation and bending of the fibers, and the caveolae-mediated clearance of adhesive integrin receptors and bound fibers for degradation or recycling. This process is unprecedented and reveals a higher level of complexity than proposed by the current literature.

In conclusion, whereas the relevance of caveoli in mediating focal and linear tumor cell interactions to collagen fibers is confirmed here, the mechanistic advance in how these structures control collagenolysis, other than by mediating adhesion, remains unclear.

Our study is not a confirmatory one, but represents a significant conceptual advance by providing several original observations and a novel mechanism for matrix remodeling by tumor cells based on an interplay of caveolae and invadopodia. This referee's statements are based on his/her wrong assertion that Montagnac and colleagues already reported an association of caveolar structures with adhesion sites to fibrillar collagen.

1. Figures 1, 2 and 5 rehash Montagnac et al. (Ref 66) with additional detail on topology. Inexact; Montagnac did not address caveolae.

The mechanisms and relevance of the juxtaposition of caveoli and lytic zones have not been mechanistically explored. Therefore, the authors may want to move the confirmatory parts of these Figs. to the supplement, and shorten the text accordingly.

We agree that we did not provide a detailed analysis of how caveolae can influence nearby invadopodia formation except postulating that strong integrin-mediated adhesion at caveolae-collagen fiber contact sites may facilitate nearby invadopodia formation – a point that we will discuss as suggested by this referee.

2. The terminology is confusing and poorly motivated by experimental evidence. Cortactin and TKS5 are classical focal adhesion markers, however the authors refer to "invadopodia" throughout this manuscript, without specific justification on how focal adhesion sites, invadopodia and putative other collagenolytic structures have been discriminated

We respectfully disagree and will provide data showing that there is no overlap between peripheral focal adhesion structures and invadopodia. First, it is worth mentioning that according to Sara Courtneidge who discovered the Tks5 protein, 'Tks proteins are found localized to invadopodia in many invasive cancer cells (Seals et al., 2005; Iizuka et al., 2016). Moreover, Tks5α and Src are considered to be defining elements of invadopodia. These structures were first defined in Src-transformed fibroblasts (Tarone et al., 1985), and the exogenous expression of Tks5α along with active Src induces formation of invadopodia in non-invasive cancer cell lines, which usually lack them (Seals et al., 2005).' (PMID: 29311151).

In addition, a PubMed search for the term 'tks5 and focal adhesion' retrieved 7 results (5 correspond to original papers that mention tks5 and focal adhesion but actually do not equal them: PMID: 18606851; PMID: 19109420; PMID: 19262173; PMID: 25118939; PMID: 33566639), while the terms 'tks5 and podosomes' and 'tks5 and invadopodia' retrieved 67 and 92 results, respectively. In addition, Tks5 was not found as a component of the adhesome, based on proteomic analysis reported independently by the Fässler and Geiger's lab (PMID: 21311561; PMID: 17671451). Accordingly with the fact that Tks5 is not considered as a focal adhesion component, the commonly used definition of Tks5 is an 'Adapter protein involved in invadopodia and podosome formation, extracellular matrix degradation and invasiveness of some cancer cells'.

Regarding cortactin that is also mentioned by the referee, it is true that this protein is not exclusive of invadopodia and can be found in different locations in the cell in relation with Arp2/3 complex-mediated branched actin filament networks. However, when we refer to cortactin-positive invadopodia, we do refer specifically to elongated curvilinear structures present on the cytoplasmic face of the plasma membrane, in

contact with – mostly bent – collagen fibers and that we have shown to be degradative in contrast to more peripheral, non-degradative cortactin-positive structures that could represent a distinct cortactin pool associated with focal adhesions (see **Figure IXa** below; cf. Monteiro et al. 2013/Ref.46, Ferrari et al. 2019/Ref.13, Zagryzhskaya-Masson et al. 2020/Ref.14; Colombero et al. 2021/Ref.65). We will clarify this point better in the manuscript.

Furthermore, we will also provide images showing MDA-MB-231 cells plated on fibrillar collagen and revealing the peripheral localization of focal adhesions that are stained with paxillin, and weakly positive for cortactin, while more internal elongated invadopodia are strongly labeled for cortactin and negative for the focal adhesion protein, paxillin (see **Figure IXb**). These images reveal that there is little, if any, overlap between focal adhesion structures and invadopodia.

Figure IX. (a) MDA-MB-231 cells plated for 60 min on a layer of fluorescently-labeled type I collagen (blue LUT) and fixed and stained for cortactin (red LUT) and cleaved collagen with Col1-3/4C antibody (grey LUT). (b) MDA-MB-231 cells plated for 60 min on a type I collagen (unstained), labeled for the focal adhesion protein, paxillin (red LUT) and cortactin (cyan LUT).

The data in Suppl. Fig. 1 lack clear demonstration where collagen fibers are actually degraded, and where not

Recruitment of MT1-MMP to substrate may occur with or without collagenolysis (e.g., PMID 17618273), but this distinction is lacking here. In Fig. 3, it appears that at least a small subset of these contacts is collagenolytic, however in peripheral regions of the cell, cortactin-rich and MT1-MMP-positive foci occur in the absence of Cav1 and collagen degradation. Should these events be considered as focal adhesions/contacts or invadopodia?

For technical reason we could not label for cleaved collagen the triple stained cells shown in Suppl. Fig.1 due to species incompatibility between some antibodies used to detect the cellular structures and the Col1-3/4 antibody, which is raised in rabbit. Data provided above clearly exclude an overlap between focal adhesions and invadopodia (see our response to Point #2 of this referee and **Figure IX** above)

Are the morphological criteria (e.g., protrusion formation) fulfilled to justify classification as invadopodia at any of these sites? In Fig. 2, SEM images are inconclusive on morphology. Can sites, with or without evidence for MT1-MMP localization, active proteolysis and/or protrusion (“pod”) be considered “invadopodia”?

To avoid unwanted bias, the authors should include stringent reporting on the proteolytic sites (e.g., the fraction of Cav1 and TKS5-foci with or without collagenolysis). and report lytic and nonlytic subdomains throughout their interventions. Alternatively, the terminology should be altered to neutral terms, such cortactin-rich adhesions with or without collagenolytic activity. In case the majority or all of regions described here are indeed focal contacts/adhesions, that are required to bring MT1-MMP in sufficient proximity to collagen fibrils, the conceptual advance compared to the work by Montagnac is not clear.

We are confident that elongated electron-dense material enlightened in the SEM images in Fig.2ab correspond to collagen fibers underneath the ventral plasma membrane of unroofed cells based on our previous experience using PREM (Ferrari et al. 2019 - Ref.13 of the manuscript), and on the new correlative fluorescence and PREM image we included in this letter (see our response to Point#5/referee#1 and **Figure V** above). In the immunogold PREM images shown in Fig.2ab of the manuscript, it was not possible to detect an additional invadopodia marker. Accordingly, in the result section describing Fig.2ab, we referred accurately to ‘*In addition, Cav1GFP and Cavin1GFP labelling was detected in elongated μ m-size clusters of caveolar structures associated with the ridge formed by the PM wrapped around the electron-dense collagen fibrils lying underneath the ventral cell surface*’. In contrast, in the high-resolution STORM image provided in Fig.2c, cells were co-stained for Cav1 and cortactin allowing to unambiguously identify both caveolar and invadopodial structures. The absence of overlap between our study and Montagnac et al has already been commented above.

3. The interventions shown in Fig. 4 with the actin system should also include the effect on collagenolysis, to link adhesion topologies to actual ECM degradation.

There are several previous reports that document the strict requirement of actin polymerization for invadopodia formation and activity including ours (Ferrari et al. 2019 - Ref.13 of the manuscript). Yet, we will analyze the effects of CK666 and CytoD treatment on collagenolysis.

4. Fig. 5 seems disconnected, since here a biophysical mechanism of membrane bending is established, without further exploration of the relevance for MT1-MMP activity and collagenolysis. Can the membrane bending explain exclusion of MT1-MMP positioning and/or lytic activity?

We agree that a potential effect of membrane bending/curvature on MT1-MMP activity is an interesting possibility. Although we believe it is beyond the scope of the present article, we will be happy to discuss this possibility in the Discussion section of the revised manuscript.

5. The link between integrin internalization and collagenolysis remains unexplored.

We respectfully disagree. We provide data showing the effects of MT1-MMP and TKS5 silencing along with MMP inhibitor, GM6001, treatment, all pointing to increased level of integrins at the PM and defects in integrin internalization (Fig.6fg and Ext. Data Fig.6e-h). In addition, we reported that enhanced collagen cleavage by cells overexpressing TKS5 (TKS5^{OE}), integrin internalization is significantly enhanced, indicating that integrin internalization and collagen cleavage are linked.

Whereas caveoli appear to form an effective adhesive structure, with integrin clustering and turn-over by internalization, the link of these adhesion dynamics to MT1-MMP positioning and function remains unexplored.

We agree that we did not provide a detailed analysis of how caveolae can influence nearby invadopodia formation except postulating that strong integrin-mediated adhesion at caveolae-collagen fiber contact sites may facilitate nearby invadopodia formation—a point that we will discuss as suggested by this referee in his/her Point #1.

The authors speculate that this may aid ECM internalization, without further support by data. Can internalization of ECM be shown, and does this preferentially occur at the interface between both domains, as would be predicted by their topologic model?

We partially disagree. In Fig.6g of the manuscript, we show internalized type I collagen in perinuclear CD63-positive endolysosomal compartments as well as perinuclear cytoplasmic vesicles positive for both beta1 integrin and collagen I, which are most probably representative of some endolysosomal compartments. Finally, Fig.6g shows that the vesicles positive for internalized type collagen I are also stained for Cav1. Of note, these perinuclear Cav1-positive endolysosomes are also positive for intracellularly stored MT1-MMP (see Ext. Data Fig.6a-c). We also performed some live cell imaging and timelapse of Cav1 and cortactin-positive structures in MDA-MB-231 cells plated on fluorescently labeled collagen. Although these movies revealed multiple events of caveolae-positive punctate structures budding off nearby cortactin-positive invadopodia forming along the collagen fibers, we could not detect internalized collagen in these caveolar endocytic carriers possibly due to low amount of fluorescence in these small size vesicles. Although this is clearly frustrating, this type of analysis represents the current limit of the state-of-art confocal spinning disk live cell imaging available to us.

6. The relevance of the here described link between caveoli and collagenolysis for tumor cell invasion in 3D models (Fig. 7h, i) is rudimentary. Can the same substructures and domain segregation be demonstrated?

In addition to the data reported in the manuscript (Fig.7hi) that demonstrated a requirement for the invadopodial proteins, MT1-MMP and TKS5, and for the caveolae protein, Cav1, in 3D collagen invasion, we will provide images and line scans showing the alternation of Cav1-positive caveolar structures and invadopodial cortactin along matrix fibers during the invasion of MDA-MB-231 cells through the 3D collagen network (**Figure X**).

Figure X. (a) MDA-MB-231 cells were embedded in a fibrillar 3D network of Cy5-labeled type I collagen. After overnight incubation, cells were fixed and stained for cortactin (red) and Cav1 (Cyan). The nucleus is shown in a grey scale. Insets correspond to the boxed region in the low magnification image. (b) Intensity profile of the Cav1 and cortactin markers along a line scan drawn along the collagen fibers forming a pore in the 3D network. As we reported earlier, invadopodia often form in front of the nucleus to facilitate the passage of this bulky organelle through the dense collagen matrix (Infante et al. Ref.19 of the manuscript).

Are the effects of Cav1 or TKS5 downregulation dependent on reduced adhesion or compromised collagenolysis, or both?

We cannot provide a definitive answer to this point. However, it is worth mentioning that a direct role for TKS5 in cell adhesion can be excluded as this protein is absent from classical focal adhesion structures and exclusively associated with the invadopodial matrix-degradative apparatus of invasive tumor cells (also see our reply to Point#2 raised by this referee). Therefore, our data based on TKS5 knockdown are supportive of a major role for collagenolysis.

Decision Letter, first revision:

Dear Dr Chavrier,

Thank you for your email asking us to reconsider our decision on your manuscript, "A mechanosensitive caveolae-invadopodia interplay drives matrix remodeling for cancer cell invasion". We are always willing to hear the authors' perspective, but we must first prioritize decisions on new submissions. We appreciate your patience while we considered this appeal.

In light of the points you raised, and the referees' comments and your rebuttal, I have discussed your manuscript again with my colleagues. However, I regret to inform you that we cannot reverse our original decision.

We do appreciate that you are willing to address referee concerns with new experiments and have provided some preliminary results, and we are not questioning whether this work would be of interest to researchers working in this field. However, given that the referees were negative on the dataset and raised a multitude of concerns on the causality and mechanistic insight of the dataset, at this stage we cannot commit to further editorial action in the absence of a fully revised manuscript that would address all referee concerns in detail, including those pertaining to mechanistic insight between forces, ECM, invadopodia, and integrin trafficking, and thus our view remains that the study is too preliminary to pursue at this stage.

Again, we are sorry that on this occasion we cannot be more positive, but we wish you success in seeking rapid publication elsewhere.

Yours sincerely,
Daryl Jason David

Daryl J.V. David, PhD

Senior Editor, Nature Cell Biology
Nature Portfolio

Heidelberger Platz 3, 14197 Berlin, Germany
Email: daryl.david@nature.com
ORCID: <https://orcid.org/0000-0002-9253-4805>

** As a service to authors, Springer Nature provides authors with the ability to transfer a manuscript that one journal cannot offer to publish to another journal, without the author having to upload the

manuscript data again. To transfer your manuscript to another Nature journal using this service, please click [Redacted]

** For Nature Portfolio general information and news for authors, see <http://npg.nature.com/authors>.

Author Rebuttal, first revision:

NCB-A49883_A-Z – Monteiro *et al.* - A mechanosensitive caveolae-invadopodia interplay drives matrix remodeling for cancer cell invasion

Detailed point-by-point response

Reviewer #1 (Remarks to the Author):

Comments for Monteiro et al

*This paper describes the possible role of caveolae and the cellular invagination having matrix metalloproteinase for extracellular matrix remodeling for cancer cell metastasis, investigating the role of caveolae in invadopodia formation and managing to fill the gap where caveolae were reported to be the essential regulator of invadopodia formation. **The hypothesis and the model that the authors propose are novel and interesting.** However, several points are better to be addressed.*

We would like to thank this Referee for her/his positive comments and constructive comments on our manuscript.

1. The constriction of the ECM fibers is assumed from the arc-like configuration of the ECM fibers. I am not sure the shape alone can suggest the force applied to the ECM for constriction. The ECM would be caught by integrin, and then the ECM would be bent by force generated by the cells. Then, the ECM fibers would be linear in the absence of these. The linearity of ECM fibers should be quantified in all the figures.

Whether cells can bend collagen fibers and the respective contribution of caveolae and invadopodia to matrix bending are critical issues, which were discussed by all three expert referees.

We thus performed an extensive, more thorough analysis of the effects of cells on ECM bending and remodeling that resulted in the inclusion of several new datasets in the revised manuscript (ms). First, we added a representative movie (**Supplementary Video 1**) and gallery of still images taken from this movie (**Ext. Data Fig. 2b**), showing how TKS5^{GFP}-positive invadopodia (magenta) remodel the underlying collagen fibrils (grey), with progressive bending of the fibrils underneath the cell surface in close association with the dynamic invadopodia. These data confirm our previous published observations (Ref.14).

In addition, we now provide a quantitative assessment of the global orientation of the fibrils and thus of the architecture of the collagen network in regions of contact with the cells in comparison to cell-free areas in the network. For this purpose, we used the OrientationJ Dominant Direction Plugin (Ref.73 in the revised manuscript) in Fiji, which computes the orientation and isotropy of a region of interest with an index of '1' indicating perfectly

oriented, mostly straight fibrils, and '0' representing isotropic areas. Using this metrics, we found that collagen fibril orientation or alignment is higher outside the cells while alignment is reduced underneath the cell in relation with visible bending of the fibrils and invadopodia function (**Fig.2a,b and Ext. Data Fig. 2a**). Furthermore, we found that the collagen orientation index decreased between 30 min and 60 min after cell seeding on the collagen network, confirming and rationalizing the increase in collagen bending over time what we already reported (**Fig.2b and Ext. Data Fig.2b** and see Ref.14). Together, these analyses support our conclusion that cells and invadopodia mediate collagen fibril remodeling.

To refine this analysis and investigate the respective contribution of caveolae and invadopodia to collagen fibril remodeling, we lowered the scale of our analysis and re-examined the tangential curvature metrics that was already included in the initial submission. Now, using a pixel(px)-scale analysis, we could correlate tangential curvature with the preferential enrichment for cortactin (invadopodia) or Cav1/cavin1 (caveolae) by calculating the intensity ratio of cortactin over Cav1 (or cavin1) for each px from several independent linescans (see our detailed response to Point #4 of this Referee). This analysis allowed to conclude that caveolae associate preferentially with straight segments of the collagen fibrils, while invadopodia associated preferentially with curved portions of the fibrils, in agreement with their remodeling function (**Fig.2c-g and Ext. Data Fig. 2c**) (Ref.14). Next, we computed the collagen orientation index for cells in which invadopodia or caveolae formation has been disrupted by TKS5 KD or Cav1 KD, respectively. We found increased alignment (i.e., reduced bending) under both situations along with a diminution in invadopodia formation and activity (**Fig.4a-e**). Furthermore, the requirement for invadopodia in inducing fibril bending was confirmed by assessing alignment in cells in which invadopodia were disrupted by actin disrupting drugs (CK-666 and CytoD; **Fig.5d**), resulting in an increase in fiber alignment (**Fig.5f**). Altogether, these new analyses on the linearity of ECM fibers clearly underscore that cell-mediated collagen remodeling and bending involves the activity of invadopodia with a regulatory role for caveolae.

2. Several marker proteins, which are N-WASP, cortactin, and Tks5, of invadopodia were used but not consistent. The protease activity appears to be detected by the collagen cleavage antibody (Col1-3/4C), and the co-localization of this collagen cleavage and the invadopodia markers will be useful information.

Here, we would like to posit that the association of N-WASP, cortactin and Tks5 with invadopodia has been extensively described by us (Monteiro et al. (Ref.48), Zagryazhskaya-Masson et al. (Ref.15)) and by others (Seals et al. 2005 PMID: 15710328). In addition, we already reported the co-localization of these invadopodial markers with cleaved collagen labeled with Col1-^{3/4}C antibody in several previous publications:

- **TKS5^{GFP} – Col1-^{3/4}C**: Ferrari et al. (Ref.14); Zagryazhskaya-Masson et al. (Ref.15); Colombero et al. (Ref.70)
- **F-actin - Col1-^{3/4}C**: Monteiro et al. (Ref.48)
- **Cortactin - Col1-^{3/4}C**: Zagryazhskaya-Masson et al. (Ref.15); Colombero et al. (Ref.70)

In addition, an example of cortactin and Col1-^{3/4}C co-staining in MDA-MB-231 cell is now provided in the revised manuscript for the reader's benefit (**Fig.1b**).

Finally, the revised manuscript provides additional evidence for the co-localization of N-WASP with TKS5^{GFP} along collagen fibers in MDA-MB-231 cells (**Ext. Data Fig.8a**). Co-staining for endogenous N-WASP and collagen cleavage (Col1-^{3/4}C) could not be performed because antibodies are both rabbit polyclonal.

3. Related to 2, whether this marker protein could generally represent the invadopodia at all the transition of invadopodia from precursors or to caveolin-1-containing structures is not clear and should be investigated.

We think that possibly because of the over simplified model that we initially proposed in the original submission, this referee may have misinterpreted the data (also discussed by the referee in his/her Point #7). We believe our data exclude a transition 'of invadopodia from precursors or to caveolin-1-containing structures', but rather

support the conclusion that caveolae initially associate with the fibrils in a process involving membrane curvature and collagen-binding receptors, namely beta1 integrins. Subsequently, invadopodia formation is favored at locations adjacent to caveolae clusters at the PM-collagen contact sites. Later on, through their collagenolytic activity, invadopodia mediate the detachment and endocytic clearance of caveolae together with integrins from the contact sites. A new '3D' model is now provided in **Fig.9** of the revised article that depicts caveolar structures and invadopodia forming next to each other along the fiber length with no transition from one structure to the other.

Furthermore, the conclusion that invadopodia mature from non-degradative structures to degradative ones is mostly based on published data using gelatin as a matrix construct. There is very little data confirming this transition in the context of the more physiological fibrillar collagen matrix that we used here. In this setting, degradation is detected with the Col1-^{3/4}C after sample fixation and at any given time we do observe degradation together with the invadopodia markers (F-actin, TKS5, cortactin...) we looked at, probably because at any given time 'invadopodia' represent a full spectrum of structures captured at different stages and degradation probably occurs very early at the onset of invadopodia formation. In addition, Col1-^{3/4}C antibody and available caveolae antibodies are all generated in rabbit, making it difficult to assess collagenolysis and caveolar structures at the same time.

4. Figure 1, the line scan needs the statistics of reproducibility. Auto-correlation co-efficient average among various observations would be one of the statistics.

Examples of individual line scans initially provided Figure 1 were useful to document the alternation of caveolin-1 (Cav1)-positive and cortactin-positive structures along collagen fibers (**Fig.1c** in the revised ms) and to reveal a preferential association of Cav1 (and cavin1)-positive structures with straight segments of the collagen fibers while cortactin (and TKS5)-positive structures are associated with bent regions (**Fig.2c and 2e and Ext. Data Fig. 2c** in the revised ms).

We agree with the Referee that this type of representation and analysis lacked statistical robustness. Concerning the co-localization of Cav1 and cortactin along the fibers, we are now providing cloud plots showing Cav1 and cortactin intensity for each pixel from 32 independent line scans taken along different collagen fibers from different cells in different experiments (**Ext. Data Fig. 1d** in the revised ms). The same type of analysis was performed for Cavin1 and cortactin (33 independent line scans; **Ext. Data Fig. 1e** in the revised ms) as well as for TKS5 and cortactin (35 independent line scans; **Ext. Data Fig. 1a** in the revised ms). In addition, we plotted the corresponding Pearson correlation coefficient for each individual line scan for these three conditions (**Fig.1e** in the revised ms). These analyses clearly demonstrate a negative correlation between caveolae (Cav1, Cavin1) and invadopodia (cortactin) markers and positive one between the two analyzed invadopodia markers (TKS5 and cortactin). In addition to the linescan provided in **Fig.1c** of the revised manuscript, these new data strengthen the conclusion that caveolae (Cav1) and invadopodia (cortactin) markers alternate and never colocalize along fibers.

Regarding the statistical analysis of caveolae and invadopodium marker association with straight vs. bent fiber segments, respectively, we analyzed the intensity of Cav1 and cortactin markers for each pixel along several line scans drawn along different fibers and calculated a cortactin/Cav1 intensity ratio for each pixel (**Fig.2c**). Pixels from 22 independent line scans were then stratified in two classes based on cortactin/Cav1 intensity ratio >1 (i.e., cortactin-enriched invadopodia, red in **Fig.2d**) and cortactin/Cav1 intensity ratio <1 (i.e., cav1-enriched caveolae, blue in **Fig.2d**) and we plotted the frequency distribution of tangential curvature (μm^{-1}) for each pixel, showing that cav1-enriched structures (caveolae) are associated with low curvature, in contrast to cortactin-enriched structures (invadopodia) that are associated with higher curvature. The same analysis was performed for Cavin1, as an independent caveolae marker based on 21 independent line scans (**Fig.2f**). Complementing the individual line scans provided in **Fig.2c and 2e** of the revised manuscript, these new analyses compiling many line scans from

different cells strengthen the conclusion that caveolae associate with straight segments, while invadopodia are associated with bent regions of the fibers.

5. Figure 2, whether the thick fibers are collagen fibers or not, would need confirmation. Caveolae are sometimes on thick stress fibers, and it would be challenging to be distinguished them from Fibrils beneath the membrane.

The identification of the electron-dense elongated material (appearing with light shades in the inverted greyscale image in **Fig.3ab** and in **Ext. Data Fig.3**) as bundles of collagen fibers underneath the ventral plasma membrane of unroofed cells is based on our previous published work (Ferrari et al. 2019 - Ref.14 of the manuscript), performed in collaboration with Dr S. Vassilopoulos, a world expert in platinum replica electron microscopy (PREM) and co-author of the present study. In addition, during this study, we performed correlative light microscopy and PREM analysis, which is not included in the present manuscript, and that is provided below for the referee only (**FIGURE I**). The image shows the overlap of fluorescence signal of Cy5-labeled collagen fibrils with elongated electron-dense structures, which can thus be identified undoubtedly as collagen fibers. In the classical PREM images shown in **Fig.3ab** of the manuscript, Cav1^{GFP}- and Cavin1^{GFP}-positive structures are associated with the same elongated electron-dense material identified as collagen fibers. Of note, for technical reason it was not possible to perform correlative light microscopy and immunogold PREM analysis that was required to identify Cav1- and Cavin1-positive structures.

FIGURE I. The left image is a PREM view of the ventral plasma membrane of an unroofed MDA-MB-231 cell plated for 60 min on a layer of type I collagen. The central fluorescence image shows the underlying layer of Cy5-labeled collagen I fibrils (blue). The right image is the correlative fluorescence microscopy/PREM overlay image showing that the elongated electron-dense structures with associated caveolae clusters do correspond to the collagen fibrils underneath the plasma membrane.

6. Figure 5. The presence of ECM seemed to promote invadopodia formation. Can ECM also promote the clearance of surface integrin-mediated by caveolae due to the upregulated invadopodia-mediated collagen cleavages? Can the nanobar with gelatin can be stained with the antibody for collagen cleavage?

We could not assess gelatin degradation on the nanobar substratum using the Col1- α C reagent as this antibody does recognize only fibrillar cleaved collagen (gelatin is non fibrillar denatured collagen). The main rationale of the nanobar chip assay was to generate plasma membrane deformations with size like the dimension of the collagen fibril bundles but with a simpler and fixed geometry. In this regard, coating the nanobar array with fibrillar collagen would be technically challenging and would result in the superimposition of a network of randomly oriented collagen fibrils and the simple geometry of parallel nanobar array generating unnecessary complexity.

7. They also revealed rather mutually exclusive localization of cavelin-1 and integrin with invadopodia, which is in contrast to the studies (Yamaguchi *et al*, 2009, which is cited), where the endogenous cav-1 was colocalized with MT1-MMP. Is there any transient form of the mixture of MT1-MMP and caveolin-1 in the ultrastructure level? Because model cartoon in Fig 8 suggests the transit form, but it would need to be clarified in the data.

In their original 2009 paper, Yamaguchi and colleagues did not actually assess the localization of endogenous Cav1 and MT1-MMP. However, these authors looked at the distribution of overexpressed mCherry-tagged Cav1 and GFP-tagged MT1-MMP. Moreover, Yamaguchi *et al.*, 2009 provided only one single image of a single cell showing limited overlap between the two overexpressed proteins on some unidentified cellular structures with no information of the overlap of the markers with matrix (gelatin) degradation. In addition, this study did not provide any quantitative analysis of the overlap between these markers (Yamaguchi *et al.*, 2009, Fig.5b,c). In contrast, our study provides a detailed quantitative analysis of the co-localization of several caveolae and invadopodia markers, including data regarding the distribution of endogenous Cav1 and MT1-MMP (**Ext. Data Fig.1b,c** and **Ext. Data Fig. 7a**). In addition, Yamaguchi and colleagues showed endogenous Cav1 alone (no counter labeling) associated with some areas of gelatin degradation that could correspond to invadopodia although there was no additional marker to ascertain their conclusion (Yamaguchi *et al.*, 2009, Fig.4a).

Although we mostly used a network of collagen fibrils as a more physiological matrix (as compared to a uniform thin coating of gelatin, i.e., denatured collagen), we also looked at – but did not show – Cav1 distribution in cells plated on gelatin. These experiments revealed that Cav1-positive puncta are associated to focal areas of gelatin degradation (appearing as dark areas). Similar to our observations based on the fibrillar collagen network documented in our paper, Cav1 did not co-localize with invadopodial cortactin-positive puncta forming in association with degraded gelatin spots, but Cav1 and cortactin-positive puncta were rather juxtaposed to each other, in line with our reported observations made using the collagen fiber network (see examples provided in **FIGURE II** below for the referee).

FIGURE II. MDA-MB-231 cells plated for 4 hrs on Oregon488-gelatin (blue). Cells were fixed and stained for Cav1 (cyan) and cortactin (red). Insets correspond to boxed regions in the lower magnification views.

Finally, the referee raised a point regarding the “mixture of MT1-MMP and caveolin-1 in the ultrastructure level” based again on over simplistic model proposed in the previous version of the manuscript. In that respect, we acknowledge that some confusion may have been generated by the limitation of the sketched model to represent the complex 3D organization of the structures and their dynamic relationship using a flat scheme. We have now collaborated with an expert in scientific drawing who corrected the model to represent the complexity of the structures and better illustrate their relationship (see new **Fig.9** of the revised manuscript).

Minor points

- In all figures, the blue representation of collagen fibers is sometimes not easy to see and should be modified.

We have used a blue pseudo-color because, according to our experience, this LUT allowed to show the other markers best. It may not be ideal but most of the figure panels include high magnification insets in which the blue collagen fibers are highlighted and visible.

- Figure 3 c and d. The amount of collagen cleavage by siTKS5 and siCav1 in the bar chart and the images looks to be the opposite.

We agree that the small dotted labeling may seem stronger in siTKS5 as compared to siCav1 in Fig.4cd of the revised ms. We would like to mention that a filter is applied to the Col1-3/4C signal to discard structures of a size below < 10 px, which according to our previous experiments represent mostly background. Indeed, this signal remains even in conditions in which MT1-MMP or TKS5 are silenced, or cells are treated with GM6001 and may also be somehow in cell-free areas of the collagen network. The details of the quantification tool we have setup based on several years of analysis of invadopodia-mediated collagen degradation are described in a recent methodological article that provides the codes for these macros (see **Ref.72** of the revised ms).

- Figure 5 c. The groups seemed to have no difference.

We apologize. It was our mistake. ‘****’ changed to ‘ns’ according to statistical analysis (Supplementary Statistics_Table).

- Figure 5b, it is not clear where the cells locate.

The cell contour has been outlined (**Fig.6b** of the revised ms).

Reviewer #2 (Remarks to the Author):

Cancer cell interactions with the extracellular matrix is a key determinant in cancer cell migration, invasion and dissemination. Type I collagen is the major stromal component and hence it is important to understand how cells interact with complex collagen fibrillary matrixes and hoe these fibrils are remodeled to facilitate cancer cell invasion. In this study, Monteiro and co-workers have identified interesting adhesive/degradative structures which are assembled on collagen fibers and involved in their remodeling. The authors use a combination of high- and super-resolution microscopy methods in conjunction with functional studies. They show that caveolae and MT1-MMP1 containing invadopodia are recruited in a periodic pattern to the fibrils with these two different structures residing adjacent to each other on the fibril. These structures are sensitive to b1 integrin depletion. Furthermore, depletion of invadopodia components result in accumulation of the caveolae structures along the fibrils indicating that proteolytic turnover of collagen is linked to their dynamics. The authors suggest that these structures and MT1-MMP activity is involved in bending and softening of the fibrils and propose that this mechanism may be relevant in cancer cell migration in collagen rich matrixes.

This is an interesting study. The manuscript is well-written, the data are of high quality and the data support the conclusions to a large extent.

We thank the Referee for her/his positive and insightful comments on our work.

- One major concern is that the current data provide very little (if any) demonstration that the cells actively generate these bent CI fibers

Whether cells can bend collagen fibers and the respective contribution of caveolae and invadopodia to matrix bending are critical issues. We have therefore performed an extensive, more thorough analysis of the effects of cells on ECM bending and remodeling that resulted in the inclusion of several new datasets in the revised manuscript.

First, we added a representative movie (**Supplementary Video 1**) and gallery of still images taken form this movie (**Ext. Data Fig. 2b**), showing how TKS5^{GFP}-positive invadopodia (magenta) remodel the underlying collagen fibrils (grey), with progressive bending of the fibrils underneath the cell surface in close association with the dynamic invadopodia. These data confirm our previous observations describing fiber remodeling and bending activity of invadopodia based on live cell imaging, laser ablation experiments... (Ferrari *et al.* 2019 - Ref.14 of the manuscript). In brief, our previous study uncovered the combined action of invadopodial actin polymerization, which is involved in the generation of force that is applied against the fiber, and MT1-MMP-mediated proteolytic activity at invadopodia, which concomitantly weakens the fiber leading to fiber bending and formation of an invasion track (see also Monteiro *et al.* 2013 - Ref.48, Zagryazhskaya-Masson *et al.* 2020 - Ref.15)

In addition, we now provide a quantitative assessment of the global orientation of the fibrils and thus of the architecture of the collagen network in regions of contact with the cells in comparison with cell-free areas. For this purpose, we used the OrientationJ Dominant Direction Plugin (Ref.73 in the revised manuscript) in Fiji, which computes the orientation and isotropy of a region of interest with an index of '1' indicating perfectly oriented, mostly straight fibrils, and '0' representing isotropic areas. Using this metrics, we found that collagen fibril orientation or alignment is higher outside the cells while alignment is reduced underneath the cell in relation with visible bending of the fibrils and invadopodia function (**Fig.2a,b and Ext. Data Fig. 2a**). Furthermore, we found that collagen fibril alignment decreased between 30 min and 60 min after cell seeding on the collagen network, indicating increased collagen bending over time (**Fig.2b and Ext. Data Fig.2b**).

Altogether, these new analyses on the linearity of ECM fibers clearly underscore that cell-mediated collagen remodeling and bending involves the activity of invadopodia with a regulatory role for caveolae.

and that their generation is dependent on the interplay of the two structures on the filament.

This point is well taken and similar to Point #1 raised by Referee #1. As discussed above, we now provide analyses under the different conditions tested in our study (siRNAs, drugs...) to better characterize the contribution of caveolae and invadopodia to matrix remodeling.

We have computed the collagen orientation index for cells in which invadopodia or caveolae formation has been disrupted by TKS5 KD or Cav1 KD, respectively. We found increased alignment (i.e., reduced bending) under both situations along with a diminution in invadopodia formation and activity (**Fig.4a-e**). Furthermore, the requirement for invadopodia in inducing fibril bending was confirmed by assessing alignment in cells in which invadopodia were disrupted by actin disrupting drugs (CK-666 and CytoD; **Fig.5d**), resulting in an increase in fiber co-orientation (**Fig.5f**). Altogether, these new analyses clearly underscore that cell-mediated collagen remodeling and bending involves the activity of invadopodia with a regulatory role for caveolae.

To refine our analysis and investigate the respective contribution of caveolae and invadopodia to collagen fibril remodeling, we also lowered the scale of our analysis and re-examined the tangential curvature metrics that was already included in the initial submission. Now, using a pixel(px)-scale analysis, we could correlate tangential curvature with the preferential enrichment for cortactin (invadopodia) or Cav1/cavin1 (caveolae) by calculating the intensity ratio of cortactin over Cav1 (or cavin1) for each px. This analysis allowed to conclude that caveolae associate preferentially with straight segments of the collagen fibrils, while invadopodia associated preferentially with curved portions of the fibrils, in agreement with their remodeling function (**Fig.2c-g and Ext. Data Fig. 2c**) (Ref.14).

The data convincingly show that these structures are generated on these peculiar looking curved (sometimes even circular) collagen fibers. However, it is unclear if these are specifically formed by the cells under the cells (and areas elsewhere would have liner fibers).

As discussed above, using the Collagen Orientation index, we found that collagen fibril orientation or alignment is higher outside the cells while alignment is reduced underneath the cell in relation with visible bending of the fibrils and invadopodia function (**Fig.2a,b and Ext. Data Fig. 2a**). Furthermore, we found that collagen fibril alignment underneath the cells decreased between 30 min and 60 min after cell seeding on the collagen network, indicating increased collagen bending over time (**Fig.2b and Ext. Data Fig.2b**). Together, these analyses support our conclusion that cells and invadopodia mediate collagen fibril remodeling.

There is also no data currently that would directly support the interesting suggestion that invadopodia soften the fibers resulting in their bending and matrix remodeling.

As already discussed above, the conclusion that invadopodia soften the fibers through MT1-MMP activity is based on our previous published work (see Ferrari et al. 2019 - Ref.14 of the manuscript and see **Fig.8e and Ext. Data Fig. 2b** of the revised manuscript), in which we used live cell imaging to look at the dynamics of TKS5^{GFP}-positive invadopodia and the contacted fluorescently labeled collagen fibers overtime and laser ablation. It allowed us to report that invadopodia exert a pushing force towards the contacted fibers, which requires actin polymerization. This force pushes the fibers aside while maintaining a tight contact between the invadopodial cell surface and the

pushed fibers. Upon inhibition of MT1-MMP activity, there is no such movement of the invadopodia/fiber ensemble as the counterforce exerted by non-cleaved, stiff fibers opposes the pushing force generated by actin polymerization at invadopodia (Ferrari et al. 2019 - Ref.14).

In addition, as mentioned in our response to a previous comment of this referee, we now provide a gallery of still images from a representative movie showing how TKS5^{GFP}-positive invadopodia (magenta) remodel collagen fibrils (grey) (see **Ext. Data Fig. 2b**).

The other major concern is that all the siRNA experiments are performed with single oligos and lack off-target controls. This must be addressed as the off-target effects of siRNAs are prevalent may result in faulty conclusions.

Here we would like to emphasize that as described in the Material and Methods section, most of the experiments in the article were performed with smartPools (i.e., mix of four siRNAs for MT1-MMP/MMP14, TKS5, Cav1, beta1 integrin, beta3 integrin, N-WASP), which minimize off-target effects. It is only for a limited number of experiments describing the effect of Cavin1 KD that we used a single siRNA that has been extensively validated in Christophe Lamaze's publications, an expert in the cell biology of caveolae.

Additional specific points:

1) Figure 1d,e please show that the bending specifically links to these structures

Related to the statistical analysis of caveolae and invadopodium marker association with straight vs. bent fiber segments, respectively, we analyzed the intensity of Cav1 and cortactin markers for each pixel along several line scans drawn along different fibers and calculated a cortactin/Cav1 intensity ratio for each pixel (**Fig.2c** of the revised ms). Pixels were then stratified in two classes based on cortactin/Cav1 intensity ratio >1 (i.e., cortactin-enriched invadopodia, red in **Fig.2d** of the revised ms) and cortactin/Cav1 intensity ratio <1 (i.e., cav1-enriched caveolae, blue in **Fig.2d** of the revised ms), and we plotted the frequency distribution of tangential curvature (μm^{-1}) for each pixel (N, 22 independent line scans) showing that cav1-enriched structures (caveolae) are associated with low curvature, in contrast to cortactin-enriched structures (invadopodia) that are associated with higher curvature. The same analysis was performed for Cavin1, as an independent caveolae marker (**Fig.2e,f** of the revised ms, N, 21 line scans). Complementing the individual line scans provided in **Fig.2c and 2e** of the revised manuscript, these new analyses strengthen the conclusion that caveolae associate with straight segments, while invadopodia are associated with bent regions of the fibers.

2) Figure 3a-d and Figure 4d,e. Do the siRNAs/drugs that interfere with the invadopodia also abolish fiber bending? It would be important to show images of the collagen channel alone in cells covered and cell-free areas and ideally provide some live imaging either here or linked to the point below (Figure 4 f,h)

We now provide quantification of collagen fibril orientation in cell-free vs cell-associated collagen area (**Fig.2a,b** of the revised ms), in cells with disrupted invadopodia and caveolae (**Fig.4e** of the revised ms) and following invadopodia inhibition in cells treated with actin-disrupting drugs CK-666 and CytoD (**Fig.5f** of the revised ms). Furthermore, we provide images of "collagen channel alone in cells covered (i.e., cell-associated) and cell-free areas" (**Fig.2a** and **Ext. Data Fig. 2a** of the revised ms).

In addition, in response to this referee (see comment to "major concern" above) and to referee #1 (see comment to Referee #1/Point #1), we are now providing a gallery of still images from a representative movie showing how

TKSS^{GFP}-positive invadopodia (magenta) remodel collagen fibrils (grey) – ending in an increase bending of the fibrils (Ext. Data Fig. 2b of the revised ms).

3) *Figure 4 f, h. I struggle to find direct evidence for the authors claim “along with conversion of straight fibers into curved ones” page 10.*

We agree. This was an overstatement as no direct evidence was shown in the initial submission. We are now providing qualitative (live cell imaging; Ext. Data Fig. 2b and Supplementary Video 1 of the revised ms) and quantitative (collagen alignment quantification; Fig. 2a,b of the revised ms) evidence for the “conversion of straight fibers into curved ones”.

4) *Figure 5 b,c why was gelatin used instead of collagen?*

We would like to stress that the main rationale of the nanobar chip assay was to generate plasma membrane deformations with size similar to collagen fiber bundles but with a simpler and fixed geometry. In this regard, coating the nanobar array with fibrillar collagen would be technically challenging and would most probably lead to the superimposition of a network of randomly oriented collagen fibers to the regularly spaced and parallel nanobar array generating unwanted complexity.

5) *Figure 5g. How do the cells adhere to collagen in the absence of all collagen binding integrins? This experiment is a little hard to understand*

We would like to clarify that even though integrin knockdown was efficient (see Ext. Data Fig. 6c of the revised ms), depletion was not complete and residual amount of beta1/beta3 integrins could provide sufficient cell adhesion capacity to the type I collagen network. Yet, we observed – but did not include in the manuscript – that cells depleted for integrins were smaller compared to control cells (see FIGURE III provided for this referee). Thus, silencing beta1/beta3 integrins had a direct effect on cell attachment to and cell spreading on the network of collagen fibers. In addition, we cannot exclude that, in the absence of beta1 or beta3 integrins, some compensatory mechanism may be involved based on another collagen receptor(s) (i.e., DDR1/2, MT1-MMP) known to bind type I collagen fibers.

FIGURE III. Quantification of cell area of MDA-MB-231 cells treated with non-targeting siRNA (NT) or with beta1 integrins siRNAs (β 1integrin) and plated for 60 min on type I fibrillar collagen.

5) Please show images of the endocytosed β 1-integrin co-stained with markers for the endolysosomes (CD63, Rab7) alongside markers for early endosomes (as numerous studies have indicated integrins traffic via this compartment regardless of the initial entry routes). Is the endolysosomal localization due to the long 60 min incubation? Are the integrins degraded rather than recycled? This should at least be discussed. We are now providing data for endocytosed β 1-integrin co-stained with endosomal markers Rab5 and Rab7 (**Ext. Data Fig.7ef** of the revised ms).

We imaged total β 1 integrin at steady state and observed localization in endolysosomal compartments (**Fig.7d** of the revised).

Some level of integrin degradation in the endolysosomal system is possible and has been described in other systems (Lobert et. 2010, PMID: 20643357; Dozynkiewicz et al. 2012, PMID: 22197222) but was not addressed in the present work.

6) Figure 7f. Is the ability of TKS5 OE to drive PM accumulation of β 1-integrins specific to collagen fiber ECM and the formation of these structures described here or does this occur also on other matrixes? We believe that there might be a misunderstanding from the referee as our data on the effect of TKS5 overexpression shows the opposite, i.e., less β 1 integrins at the PM in cells overexpressing TKS5 (TKS5^{OE}) (**Fig.8f**).

Reviewer #3 (Remarks to the Author):

Beta1 integrin in association with caveolar structures at adhesion sites to fibrillar collagen was reported earlier by the Montagnac group (PMID: 28619886; Ref. 66).

This is a main, recursive criticism from Referee #3 who was concerned that a previous article published by Montagnac and colleagues (PMID: 28619886; cited as Ref.71 in our manuscript) already reported the association of caveolae with fibrillar collagen. We respectfully disagree with this referee's statement. The article published by Montagnac and colleagues in 2017 did not report the association of caveolae with fibrillar collagen (there is no mention of caveolae in this article). This work established the association of clathrin-coated pits (CCPs) with collagen fibers in the environment. In addition, this study did not investigate or propose any kind of interplay between CCP-collagen association and matrix remodeling and invadopodia. There is no observational nor conceptual overlap between their and our study.

In this work, Monteiro and coworkers report that caveolin structures are juxtaposed with TKS5 and cortactin-rich focalized and linear zones, which alternate in sequence along collagen fibrils. Using high-quality confocal and superresolution microscopy and RNA interference, they show that formation of these structures and associated perinuclear collagen degradation depend both on Cav1 and TKS5 or cortactin. Similarly, they show that caveolin-positive foci and collagenolysis depend on actin polymerization and branching, using pharmacological inhibitors and further demonstrate that caveolae-mediated adhesion precedes collagenolysis. Lastly, they show that caveoli-formation supports the recruitment and internalization of beta1 integrins. The authors propose a sequential model by which caveoli first form from an adhesive bridge to collagen fibrils, which enables MT1-MMP recruitment, collagenolysis and turnover of adhesion and ECM by internalization.

The quality of the cell biological experiments showing juxtaposition of caveolae and proteolytic zones is very high. The sequential development of first caveoli followed by TKS5-positive and, in part, collagenolytic structures is convincingly shown, as the requirement of caveoli for collagenolysis.

We would like to thank this Referee for her/his positive appreciation of our work and helpful and challenging comments.

It is well established that in 2D and 2.5D models perturbation of cell adhesion can negatively impact ECM degradation, but the existence and mechanistic relevance of the proposed step-wise process in 3D invasion remains to be demonstrated

The link between adhesion (integrins) and matrix degradation has been established using the 2D gelatin substratum model (i.e., denatured collagen), and data are missing in the context of fibrillar collagen, which is known to engage different receptors. In addition, how adhesion can control ECM degradation and invadopodia formation/activity is largely unknown even in the simple 2D gelatin model. We are therefore confident that our study that uncovers a new mechanism based on the interplay of caveolae and invadopodia during 2.5D and 3D invasion in fibrillar collagen is particularly relevant and timely.

In addition, the mechanism of phase separation into curved and (non-curved?) lytic domains remains unexplored

This point is well-taken and related to Point #1 from Referee #1 and also points raised by Referee #2. We thus performed an extensive, more thorough analysis of the effects of cells on ECM bending and remodeling that resulted in the inclusion of several new datasets in the revised manuscript.

First, we added a representative movie (**Supplementary Video 1**) and gallery of still images taken from this movie (**Ext. Data Fig. 2b**), showing how TKS5^{GFP}-positive invadopodia (magenta) remodel the underlying collagen fibrils (grey), with progressive bending of the fibrils underneath the cell surface in close association with the dynamic invadopodia. These data confirm our previous published observations (Ref.14).

In addition, we now provide a quantitative assessment of the global orientation of the fibrils and thus of the architecture of the collagen network in regions of contact with the cells in comparison to cell-free areas in the network. For this purpose, we used the OrientationJ Dominant Direction Plugin (Ref.73 in the revised manuscript) in Fiji, which computes the orientation and isotropy of a region of interest with an index of '1' indicating perfectly oriented, mostly straight fibrils, and '0' representing isotropic areas. Using this metrics, we found that collagen fibril orientation or alignment is higher outside the cells while alignment is reduced underneath the cell in relation with visible bending of the fibrils and invadopodia function (**Fig.2a,b and Ext. Data Fig. 2a**). Furthermore, we found that the collagen orientation index decreased between 30 min and 60 min after cell seeding on the collagen network, confirming and rationalizing the increase in collagen bending over time what we already reported (**Fig.2b and Ext. Data Fig.2b** and see Ref.14). Together, these analyses support our conclusion that cells and invadopodia mediate collagen fibril remodeling.

To refine this analysis and investigate the respective contribution of caveolae and invadopodia to collagen fibril remodeling, we lowered the scale of our analysis and re-examined the tangential curvature metrics that was already included in the initial submission. Now, using a pixel(px)-scale analysis, we could correlate tangential curvature with the preferential enrichment for cortactin (invadopodia) or Cav1/cavin1 (caveolae) by calculating the intensity ratio of cortactin over Cav1 (or cavin1) for each px from several independent line scans (see our detailed response to Point #4 of this Referee). This analysis allowed to conclude that caveolae associate preferentially with straight segments of the collagen fibrils, while invadopodia associated preferentially with curved portions of the fibrils, in agreement with their remodeling function (**Fig.2c-g and Ext. Data Fig. 2c**) (Ref.14).

Next, we computed the collagen orientation index for cells in which invadopodia or caveolae formation has been disrupted by TKS5 KD or Cav1 KD, respectively. We found increased alignment (i.e., reduced bending) under both situations along with a diminution in invadopodia formation and activity (**Fig.4a-e**). Furthermore, the requirement for invadopodia in inducing fibril bending was confirmed by assessing alignment in cells in which invadopodia were disrupted by actin disrupting drugs (CK-666 and CytoD; **Fig.5d**), resulting in an increase in fiber alignment (**Fig.5f**).

Altogether, these new analyses on the linearity of ECM fibers clearly underscore that cell-mediated collagen remodeling and bending involves the activity of invadopodia with a regulatory role for caveolae.

Because caveoli provide integrin-mediated interaction to collagen (see Ref 66)

Montagnac and colleagues did not address caveolae (see above).

it may be considered as a potentially trivial finding that also MT1-MMP mediated collagenolysis is impaired, because a 2.5D system has been use here in which cell-collagen interaction depends on active adhesion.

We agree that there is a strong need to understand the interrelations between tumor cells with their matrix environment and how adhesion to the matrix can elicit a degradation response based on invadopodia formation and activity. These are precisely the questions that we have addressed in our study. The originality of our work is to provide the first, comprehensive mechanism, which integrates an adhesive function of caveolae to straight – possibly stiffer – portions of the matrix fibers, the induction of nearby invadopodia formation and activity, the subsequent degradation and bending of the fibers, and the caveolae-mediated clearance of adhesive integrin receptors and bound fibers for degradation or recycling. This process is unprecedented and reveals a higher level of complexity than proposed by the current literature.

In conclusion, whereas the relevance of caveoli in mediating focal and linear tumor cell interactions to collagen fibers is confirmed here, the mechanistic advance in how these structures control collagenolysis, other than by mediating adhesion, remains unclear.

Our study is not a confirmatory one, but it represents a significant conceptual advance by providing several original observations and a novel mechanism for matrix remodeling by tumor cells based on an interplay of caveolae and invadopodia. This referee's statements are possibly based on his/her wrong assertion that Montagnac and colleagues already reported an association of caveolar structures with adhesion sites to fibrillar collagen.

1. *Figures 1, 2 and 5 rehash Montagnac et al. (Ref 66) with additional detail on topology. Inexact; Montagnac did not address caveolae.*

The mechanisms and relevance of the juxtaposition of caveoli and lytic zones have not been mechanistically explored. Therefore, the authors may want to move the confirmatory parts of these Figs. to the supplement, and shorten the text accordingly.

We agree that we did not provide a detailed analysis of how caveolae can influence nearby invadopodia formation except postulating that strong integrin-mediated adhesion at caveolae-collagen fiber contact sites may facilitate nearby invadopodia formation. This issue is now better discussed in the Discussion section of the revised manuscript.

2. *The terminology is confusing and poorly motivated by experimental evidence. Cortactin and TKS5 are classical focal adhesion markers, however the authors refer to "invadopodia" throughout this manuscript, without specific justification on how focal adhesion sites, invadopodia and putative other collagenolytic structures have been discriminated*

We respectfully disagree and will provide data showing that there is no overlap between peripheral focal adhesion structures and invadopodia. First, it is worth mentioning that according to Sara Courtneidge who discovered the Tks5 protein, 'Tks proteins are found localized to invadopodia in many invasive cancer cells (Seals et al., 2005; Iizuka et al., 2016). Moreover, Tks5 α and Src are considered to be defining elements of invadopodia. These structures were first defined in Src-transformed fibroblasts (Tarone et al., 1985), and the exogenous expression of Tks5 α along with active Src induces formation of invadopodia in non-invasive cancer cell lines, which usually lack them (Seals et al., 2005) (PMID: 29311151).

In addition, a PubMed search for the term 'tks5 and focal adhesion' retrieved 7 results (5 correspond to original papers that mention tks5 and focal adhesion but actually do not equal them: PMID: 18606851; PMID: 19109420; PMID: 19262173; PMID: 25118939; PMID: 33566639), while the terms 'tks5 and podosomes' and 'tks5 and invadopodia' retrieved 67 and 92 results, respectively. In addition, Tks5 was not found as a component of the

adhesome, based on proteomic analysis reported independently by the Fässler and Geiger's lab (PMID: 21311561; PMID: 17671451). Accordingly with the fact that Tks5 is not considered as a focal adhesion component, the commonly used definition of Tks5 is an '*Adapter protein involved in invadopodia and podosome formation, extracellular matrix degradation and invasiveness of some cancer cells*' (GeneCards).

Regarding cortactin, it is true that this protein is not exclusive of invadopodia and can be found in several cellular locations in relation with the presence of Arp2/3 complex-mediated branched actin filament networks. However, when we refer to cortactin-positive invadopodia, we do refer specifically to elongated curvilinear structures present on the cytoplasmic face of the plasma membrane, in contact with – mostly bent – collagen fibers and that we have shown to be degradative in contrast to more peripheral, non-degradative cortactin-positive structures that could represent a distinct cortactin pool associated with focal adhesions. To clarify this important point, we have added a new panel in Figure1 (**Fig.1b** of the revised ms), which shows the accumulation of cortactin in association with bent collagen fibers that are labelled for the Col1-^{3/4}C neo-epitope indicative of collagen cleavage; the referee can also look at **FIGURE IVa** below (and Monteiro et al. 2013/Ref.48, Ferrari et al. 2019/Ref.14, Zagryazhskaya-Masson et al. 2020/Ref.15; Colombero et al. 2021/Ref.70). This point has been better clarified in the revised manuscript in the Results section describing Fig.1.

Furthermore, we provide several images below for the sake of the Referee showing MDA-MB-231 cells plated on fibrillar collagen and revealing the peripheral localization of focal adhesions that are stained with paxillin, and weakly positive for cortactin, while more internal elongated invadopodia are strongly labeled for cortactin and negative for the focal adhesion protein, paxillin (see **FIGURE IVb**). These images reveal that there is little, if any, overlap between focal adhesion structures and invadopodia.

FIGURE IV. (a) MDA-MB-231 cells plated for 60 min on a layer of fluorescently-labeled type I collagen (blue LUT) and fixed and stained for cortactin (red LUT) and cleaved collagen with Col1-^{3/4}C antibody (grey LUT). **(b)** MDA-MB-231 cells plated for 60 min on a type I collagen (unstained), labeled for the focal adhesion protein, paxillin (red LUT) and cortactin (cyan LUT).

The data in Suppl. Fig. 1 lack clear demonstration where collagen fibers are actually degraded, and where not

Recruitment of MT1-MMP to substrate may occur with or without collagenolysis (e.g., PMID 17618273), but this distinction is lacking here. In Fig. 3, it appears that at least a small subset of these contacts is collagenolytic, however in peripheral regions of the cell, cortactin-rich and MT1-MMP-positive foci occur in the absence of Cav1 and collagen degradation. Should these events be considered as focal adhesions/contacts or invadopodia?

For technical reason we could not label for cleaved collagen the triple stained cells shown in Suppl. Fig.1 due to species incompatibility between some antibodies used to detect the cellular structures and the Col1-^{3/4}C antibody, which is raised in rabbit. Data discussed and provided above for the referee clearly exclude an overlap between focal adhesions and invadopodia (see our response to Point #2 of this Referee and **FIGURE IV** above)

Are the morphological criteria (e.g., protrusion formation) fulfilled to justify classification as invadopodia at any of these sites? In Fig. 2, SEM images are inconclusive on morphology. Can sites, with or without evidence for MT1-MMP localization, active proteolysis and/or protrusion (“pod”) be considered “invadopodia”?

To avoid unwanted bias, the authors should include stringent reporting on the proteolytic sites (e.g., the fraction of Cav1 and TKS5-foci with or without collagenolysis). and report lytic and nonlytic subdomains throughout their interventions. Alternatively, the terminology should be altered to neutral terms, such as cortactin-rich adhesions with or without collagenolytic activity. In case the majority or all of regions described here are indeed focal contacts/adhesions, that are required to bring MT1-MMP in sufficient proximity to collagen fibrils, the conceptual advance compared to the work by Montagnac is not clear.

We are confident that elongated electron-dense material enlightened in the SEM images in **Fig.3ab** correspond to collagen fibers underneath the ventral plasma membrane of unroofed cells based on our previous experience using platinum replica electron microscopy (PREM) (Ferrari et al. 2019 - Ref.14 of the manuscript), and on the new correlative fluorescence and PREM image we included in this letter (see **FIGURE I** above accompanying our detailed response to Point #5/Referee #1). In the immunogold PREM images shown in **Fig.3ab** of the revised manuscript, it was not possible to detect an additional invadopodia marker. Accordingly, in the result section describing **Fig.3ab**, we referred accurately to ‘In addition, Cav1GFP and Cavin1GFP labelling was detected in elongated μm -size clusters of caveolar structures associated with the ridge formed by the PM wrapped around the electron-dense collagen fibrils lying underneath the ventral cell surface’. In contrast, in the high-resolution STORM image provided in **Fig.3c**, cells were co-stained for Cav1 and cortactin allowing to unambiguously identify both caveolar and invadopodial structures. The absence of overlap between our study and Montagnac et al has already been commented above.

3. The interventions shown in Fig. 4 with the actin system should also include the effect on collagenolysis, to link adhesion topologies to actual ECM degradation.

There are several previous reports that document the strict requirement of actin polymerization for invadopodia formation and activity including ours (Ferrari et al. 2019 - Ref.14 of the manuscript). To clarify this point, we now provide a new quantification of the effects of CK666 and CytoD treatment on collagenolysis activity of MDA-MB-231 cells (**Fig.5d**, right row).

4. Fig. 5 seems disconnected, since here a biophysical mechanism of membrane bending is established, without further exploration of the relevance for MT1-MMP activity and collagenolysis. Can the membrane bending explain exclusion of MT1-MMP positioning and/or lytic activity?

We agree that a potential effect of membrane bending/curvature on MT1-MMP activity is an interesting possibility. At this stage, we believe it is beyond the scope of the present article.

5. The link between integrin internalization and collagenolysis remains unexplored.

We respectfully disagree. We provide data showing the effects of MT1-MMP and TKS5 silencing along with MMP inhibitor, GM6001, treatment, all pointing to increased level of integrins at the PM and defects in integrin internalization (**Fig.7f,h** and **Ext. Data Fig.7h-j** in the revised ms). In addition, we reported that enhanced collagen cleavage by cells overexpressing TKS5 (TKS5^{OE}), integrin internalization is significantly enhanced, indicating that integrin internalization and collagen cleavage are linked.

Whereas caveoli appear to form an effective adhesive structure, with integrin clustering and turn-over by internalization, the link of these adhesion dynamics to MT1-MMP positioning and function remains unexplored.

We agree that we did not provide a detailed analysis of how caveolae can influence nearby invadopodia formation except postulating that strong integrin-mediated adhesion at caveolae-collagen fiber contact sites may facilitate nearby invadopodia formation. This issue is now better discussed in the Discussion section of the revised manuscript.

The authors speculate that this may aid ECM internalization, without further support by data. Can internalization of ECM be shown, and does this preferentially occur at the interface between both domains, as would be predicted by their topologic model?

We partially disagree. In **Fig.8g** (revised manuscript), we show internalized type I collagen in perinuclear CD63-positive endolysosomal compartments as well as perinuclear cytoplasmic vesicles positive for both beta1 integrin and collagen I, which are most probably representative of some endolysosomal compartments. Finally, **Fig.8g** shows that the vesicles positive for internalized type collagen I are also stained for Cav1 and beta1 integrins. Of note, these perinuclear Cav1-positive endolysosomes are also positive for intracellularly stored MT1-MMP (see **Ext. Data Fig.7a-c**). We also performed some live cell imaging and time-lapse of Cav1 and cortactin-positive structures in MDA-MB-231 cells plated on fluorescently labeled collagen. We have included a Movie (**Supplementary Video 3**) that documents several events of caveolae-positive punctate structures budding off nearby cortactin-positive invadopodia and detaching from the plasma membrane in association with the collagen fibers (**Fig.7e**). However, we could not detect internalized collagen in these caveolar endocytic carriers possibly due to low amount of fluorescence in these small size vesicles. Although this is clearly frustrating, this type of analysis represents the current limitation of the state-of-art confocal spinning disk live cell imaging available to us.

6. The relevance of the here described link between caveoli and collagenolysis for tumor cell invasion in 3D models (Fig. 7h, i) is rudimentary. Can the same substructures and domain segregation be demonstrated?

In addition to the data reported in the manuscript (**Fig.8k,l**) that demonstrated a requirement for the invadopodial proteins, MT1-MMP and TKS5, and for the caveolae protein, Cav1, in 3D collagen invasion, we are now providing images and line scans analysis showing the alternation/exclusion of Cav1-positive caveolar structures and invadopodial cortactin along matrix fibers during the invasion of MDA-MB-231 cells through the 3D collagen network (**Fig.8h-j**).

Are the effects of Cav1 or TKS5 downregulation dependent on reduced adhesion or compromised collagenolysis, or both?

It is worth mentioning that a direct role for TKS5 in cell adhesion can be excluded as this protein is absent from classical focal adhesion structures and exclusively associated with the invadopodial matrix-degradative apparatus of invasive tumor cells (also see our reply to Point #2 raised by this referee). Therefore, our data based on TKS5 knockdown are supportive of a major role for collagenolysis. Regarding Cav1, although we cannot formally exclude a reduction of adhesion, this possibility seems rather unlikely since we found that KD of Cav1 correlated with an increase in surface expression of adhesion beta1 integrin receptors. Thus, our data rather implicate a primary effect of Cav1 KD on collagenolysis.

Minor points

The sequence of events in Fig. 1e is difficult to appreciate from still images. Likewise, the sequence of integrin and caveolin internalization (Fig. 6) is only represented by static images. The primary videos should be included, to showcase dynamic events.

We thank the referee for her/his suggestion of showing representative movies. We added three supplementary movies documenting the dynamics of the TKS5 invadopodia and underlying fibrils (**Supplementary Video 1**), the dynamic association of DsRedCortactin invadopodia and Cav1GFP-positive caveolar structures in contact with the fibrils (**Supplementary Video 2**), and the detachment of caveolar structures from ECM contact sites (**Supplementary Video 3**).

"These results suggest a potential plasticity and conformational adaptation of caveolae to the underlying fibril topology to adopt a clustered organization, possibly involved in pinching of and/or grabbing the fibrils." – Ref 66 should be cited here. Likewise, the role of caveolae in cell adhesion to fibrillar collagen should be cited in the introduction.

As we already discussed above, the article published by Montagnac and colleagues in 2017 did not report the association of caveolae with fibrillar collagen (there is no mention of caveolae in this article). This work established the association of clathrin-coated pits (CCPs) with collagen fibers in the environment. In addition, this study did not investigate or propose any kind of interplay between CCP-collagen association and matrix remodeling and invadopodia. Thus, we believe that the citation of this work would not be appropriate there.

Decision Letter, second revision:

*Please delete the link to your author homepage if you wish to forward this email to co-authors.

Dear Dr Chavrier,

Your manuscript, "A mechanosensitive caveolae-invadopodia interplay drives matrix remodeling for cancer cell invasion", has now been seen by 3 of our original referees, who are experts in caveolae (referee 1); integrin trafficking (referee 2); and cancer cell invasion (referee 3). As you will see from their comments (attached below) they find this work of interest, but have raised some important points. Although we are also interested in this study, we believe that their concerns should be addressed before we can consider publication in Nature Cell Biology.

Nature Cell Biology editors discuss the referee reports in detail within the editorial team, including the

chief editor, to identify key referee points that should be addressed with priority, and requests that are overruled as being beyond the scope of the current study. To guide the scope of the revisions, I have listed these points below. We are committed to providing a fair and constructive peer-review process, so please feel free to contact me if you would like to discuss any of the referee comments further.

In particular, it would be essential to:

A) Experimentally show internalization endosomal collagen in response to caveolae perturbations (Reviewer #3).

B) Consider alternative nomenclature of your observed structures (Reviewer #3), as well as add potential caveats and discussion on Tsk6 and lytic structure localization in discussion (Reviewer #3).

C) All other referee concerns pertaining to strengthening existing data, providing controls, methodological details, clarifications and textual changes, should also be addressed.

D) Finally please pay close attention to our guidelines on statistical and methodological reporting (listed below) as failure to do so may delay the reconsideration of the revised manuscript. In particular please provide:

In contrast, although we agree with referee #1 that use of different nanobars of diameters closer to the collagen fibrils shown in your experiments would provide valuable insights, we consider this point to be beyond the scope of the present study. Thus, addressing it experimentally will not be necessary for reconsideration of the manuscript at this journal, but we would require explicit descriptions on this part to clarify the wording not on the collagen fibrils per se but more so dealing with membrane inward curvature.

We therefore invite you to take these points into account when revising the manuscript. In addition, when preparing the revision please:

- ensure that it conforms to our format instructions and publication policies (see below and www.nature.com/nature/authors/).

- provide a point-by-point rebuttal to the full referee reports verbatim, as provided at the end of this letter.

- provide the completed Editorial Policy Checklist (found

here <https://www.nature.com/authors/policies/Policy.pdf>), and Reporting Summary (found here <https://www.nature.com/authors/policies/ReportingSummary.pdf>). This is essential for reconsideration of the manuscript and these documents will be available to editors and referees in the event of peer review. For more information see <http://www.nature.com/authors/policies/availability.html> or contact me.

Nature Cell Biology is committed to improving transparency in authorship. As part of our efforts in this direction, we are now requesting that all authors identified as 'corresponding author' on published papers create and link their Open Researcher and Contributor Identifier (ORCID) with their account on the Manuscript Tracking System (MTS), prior to acceptance. ORCID helps the scientific community achieve unambiguous attribution of all scholarly contributions. You can create and link your ORCID from the home page of the MTS by clicking on 'Modify my Springer Nature account'. For more information please visit www.springernature.com/orcid.

[Redacted]

We would like to receive the revision within four weeks. If submitted within this time period, reconsideration of the revised manuscript will not be affected by related studies published elsewhere, or accepted for publication in Nature Cell Biology in the meantime. We would be happy to consider a revision even after this timeframe, but in that case we will consider the published literature at the time of resubmission when assessing the file.

We hope that you will find our referees' comments, and editorial guidance helpful. Please do not hesitate to contact me if there is anything you would like to discuss.

Best wishes,

Daryl Jason David

Daryl Jason Verzosa David, PhD

Senior Editor, Nature Cell Biology
Nature Portfolio

Heidelberger Platz 3, 14197 Berlin, Germany
Email: daryl.david@nature.com
ORCID: <https://orcid.org/0000-0002-9253-4805>

Reviewers' Comments:

Reviewer #1:

Remarks to the Author:

Comment for Monteiro et al.

The manuscript was significantly improved. The only concern is on the nanobar experiments. The dimension of the nanobar is 300 nm in width and 900 nm in height. This is a fine experiment, but the diameter of the collagen fibers in the other experiments would be better to be compared. 300 nm is in the optical imaging range, and the authors should have an estimate of the diameter distribution of the collagen fibers in their experiments. Or still the authors can coat the collagens on the nanobar.

Reviewer #2:

Remarks to the Author:

The authors have carried out a substantial number of new experiments addressing all the points I raised in the initial review. These have strengthened the conclusions made in this interesting study.

Reviewer #3:

Remarks to the Author:

The referee acknowledges to have confused clathrin-coated pit-mediated adhesive cell-collagen interactions with the here reported caveolar structures associated with juxtaposed zones of collagenolysis. I appreciate, and agree to, the corrective statement made by the authors in the rebuttal letter. In addition, the authors have refined their claims to now propose two independent interaction systems side-by-side, which cooperate for grabbing and cleaving collagen fibrils, which is better supported by their data than the previous disputed claim of sequential maturation and interconversion of both systems. As a consequence, originality and novelty of the here reported findings is generally high, in contrast to my earlier ill-fated statements.

In the revised manuscript, new time-lapse data convincingly show the deformation of collagen fibrils in association with Tsk5 kinetics and somewhat reduced bending after downregulation of Tsk5, in line with bending as a result of invadopod-associated collagen remodeling. New data further include decrease of collagenolysis after interference with actin dynamics, and 3D images demonstrating alternating Cav1/cortactin domains along collagen fibrils in cancer cells invading 3D collagen matrix. The newly included videos are of high quality and nicely illustrate aspects of this process. These new data and the conceptual reframing strengthen this work significantly.

However, two key claims, namely (i) the conceptual framing as "invadopodia"-mediated collagen degradation and (ii) the relevance of this machinery for collagen internalization, remain unconvincing and potentially misleading.

1. Terminology: The authors maintain the term "invadopodia" or "invadopodal structures", however no evidence for a protrusive (i.e., "pod") is provided throughout the manuscript. This differs greatly from classical reports where cells drill a little hole by means of a protrusion. Potentially, a novel type of lytic

structures was discovered here, which are morphologically distinct from invadopodia. Hence, the use of this term based on molecular markers alone, but without convincing morphological evidence, should call for the use of a more neutral term.

2. The evidence that the identified structures support collagen internalization remains indirect. This is an important aspect of their work, as it differs from the invadopod concept (which would not be expected to result in collagen internalization after degradation – at least I am not aware of published evidence for this). To support this claim, the authors should provide data showing change of endosomal collagen in response to molecular interference with caveolar structures. This could be easily reached by re-probing the molecular interference samples (e.g., Fig. 4, Fig 6 f,g) for endocytosed collagen.

Minor points

The representation of the change of collagen pattern in Fig. 4c is impossible to appreciate, due to the blue color of collagen and small image size. Thresholded single-channel representations of collagen organization and cortactin are needed to link the images to the data. In addition, fibril orientation index for cell-free areas should be included in Fig. 4c as an internal negative control. Also other figures should be re-checked for clarity of key claims related to collagen fibril organization.

The authors repeatedly claim that colocalization analysis with two or multiple polyclonal rabbit antibodies is impossible, which however ignores the possibility of sequential immunostaining, with a removal step of labelling antibody in between imaging (e.g., PMID: 28692376). Because reports previously associated Tsk6 with focal adhesions in cell models under certain conditions, the authors should, at least, consider to detail their considerations associated with Tsk5 distribution and lytic structures versus focal adhesions (provided in the rebuttal letter), in brief form, in the results section (e.g., Line 127).

GUIDELINES FOR SUBMISSION OF NATURE CELL BIOLOGY ARTICLES

ARTICLE FORMAT

ABSTRACT – should not exceed 150 words and should be unreferenced. This paragraph is the most visible part of the paper and should briefly outline the background and rationale for the work, and accurately summarize the main results and conclusions. Key genes, proteins and organisms should be specified to ensure discoverability of the paper in online searches.

TEXT – the main text consists of the Introduction, Results, and Discussion sections and must not exceed 3500 words including the abstract. The Introduction should expand on the background relating to the work. The Results should be divided in subsections with subheadings, and should provide a concise and accurate description of the experimental findings. The Discussion should expand on the findings and their implications. All relevant primary literature should be cited, in particular when discussing the background and specific findings.

REFERENCES – are limited to a total of 70 in the main text and Methods combined,. They must be numbered sequentially as they appear in the main text, tables and figure legends and Methods and must follow the precise style of Nature Cell Biology references. References only cited in the Methods should be numbered consecutively following the last reference cited in the main text. References only associated with Supplementary Information (e.g. in supplementary legends) do not count toward the total reference limit and do not need to be cited in numerical continuity with references in the main text. Only published papers can be cited, and each publication cited should be included in the numbered reference list, which should include the manuscript titles. Footnotes are not permitted.

METHODS – Nature Cell Biology publishes methods online. The methods section should be provided as a separate Word document, which will be copyedited and appended to the manuscript PDF, and

incorporated within the HTML format of the paper.

Methods should be written concisely, but should contain all elements necessary to allow interpretation and replication of the results. As a guideline, Methods sections typically do not exceed 3,000 words. The Methods should be divided into subsections listing reagents and techniques. When citing previous methods, accurate references should be provided and any alterations should be noted. Information must be provided about: antibody dilutions, company names, catalogue numbers and clone numbers for monoclonal antibodies; sequences of RNAi and cDNA probes/primers or company names and catalogue numbers if reagents are commercial; cell line names, sources and information on cell line identity and authentication. Animal studies and experiments involving human subjects must be reported in detail, identifying the committees approving the protocols. For studies involving human subjects/samples, a statement must be included confirming that informed consent was obtained. Statistical analyses and information on the reproducibility of experimental results should be provided in a section titled "Statistics and Reproducibility".

All Nature Cell Biology manuscripts submitted on or after March 21 2016, must include a Data availability statement as a separate section after Methods but before references, under the heading "Data Availability". For Springer Nature policies on data availability see <http://www.nature.com/authors/policies/availability.html>; for more information on this particular policy see <http://www.nature.com/authors/policies/data/data-availability-statements-data-citations.pdf>. The Data availability statement should include:

- Accession codes for primary datasets (generated during the study under consideration and designated as "primary accessions") and secondary datasets (published datasets reanalysed during the study under consideration, designated as "referenced accessions"). For primary accessions data should be made public to coincide with publication of the manuscript. A list of data types for which submission to community-endorsed public repositories is mandated (including sequence, structure, microarray, deep sequencing data) can be found here <http://www.nature.com/authors/policies/availability.html#data>.
- Unique identifiers (accession codes, DOIs or other unique persistent identifier) and hyperlinks for datasets deposited in an approved repository, but for which data deposition is not mandated (see here for details <http://www.nature.com/sdata/data-policies/repositories>).
- At a minimum, please include a statement confirming that all relevant data are available from the authors, and/or are included with the manuscript (e.g. as source data or supplementary information), listing which data are included (e.g. by figure panels and data types) and mentioning any restrictions on availability.
- If a dataset has a Digital Object Identifier (DOI) as its unique identifier, we strongly encourage including this in the Reference list and citing the dataset in the Methods.

We recommend that you upload the step-by-step protocols used in this manuscript to the Protocol Exchange. More details can found at www.nature.com/protocolexchange/about.

DISPLAY ITEMS – main display items are limited to 6-8 main figures and/or main tables. For Supplementary Information see below.

FIGURES – Colour figure publication costs \$395 per colour figure. All panels of a multi-panel figure must be logically connected and arranged as they would appear in the final version. Unnecessary figures and figure panels should be avoided (e.g. data presented in small tables could be stated briefly in the text instead).

All imaging data should be accompanied by scale bars, which should be defined in the legend. Cropped images of gels/blots are acceptable, but need to be accompanied by size markers, and to retain visible background signal within the linear range (i.e. should not be saturated). The boundaries of panels with low background have to be demarked with black lines. Splicing of panels should only be considered if unavoidable, and must be clearly marked on the figure, and noted in the legend with a statement on whether the samples were obtained and processed simultaneously. Quantitative comparisons between samples on different gels/blots are discouraged; if this is unavoidable, it has to be performed for samples derived from the same experiment with gels/blots were processed in parallel, which needs to be stated in the legend.

Regardless of format, all figures must be vector graphic compatible files, not supplied in a flattened

raster/bitmap graphics format, but should be fully editable, allowing us to highlight/copy/paste all text and move individual parts of the figures (i.e. arrows, lines, x and y axes, graphs, tick marks, scale bars etc). The only parts of the figure that should be in pixel raster/bitmap format are photographic images or 3D rendered graphics/complex technical illustrations.

Unprocessed scans of all key data generated through electrophoretic separation techniques need to be presented in a supplementary figure that should be labeled and numbered as the final supplementary figure, and should be mentioned in every relevant figure legend. This figure does not count towards the total number of figures and is the only figure that can be displayed over multiple pages, but should be provided as a single file, in PDF or TIFF format. Data in this figure can be displayed in a relatively informal style, but size markers and the figures panels corresponding to the presented data must be indicated.

The total number of Supplementary Figures (not including the “unprocessed scans” Supplementary Figure) should not exceed the number of main display items (figures and/or tables (see our Guide to Authors and March 2012 editorial <http://www.nature.com/ncb/authors/submit/index.html#suppinfo>; <http://www.nature.com/ncb/journal/v14/n3/index.html#ed>). No restrictions apply to Supplementary Tables or Videos, but we advise authors to be selective in including supplemental data.

Each Supplementary Figure should be provided as a single page and as an individual file in one of our

accepted figure formats and should be presented according to our figure guidelines (see above). Supplementary Tables should be provided as individual Excel files. Supplementary Videos should be provided as .avi or .mov files up to 50 MB in size. Supplementary Figures, Tables and Videos must be accompanied by a separate Word document including titles and legends.

GUIDELINES FOR EXPERIMENTAL AND STATISTICAL REPORTING

REPORTING REQUIREMENTS – To improve the quality of methods and statistics reporting in our papers we have recently revised the reporting checklist we introduced in 2013. We are now asking all life sciences authors to complete two items: an Editorial Policy Checklist (found here <https://www.nature.com/authors/policies/Policy.pdf>) that verifies compliance with all required editorial policies and a Reporting Summary (found here <https://www.nature.com/authors/policies/ReportingSummary.pdf>) that collects information on experimental design and reagents. These documents are available to referees to aid the evaluation of the manuscript. Please note that these forms are dynamic 'smart pdfs' and must therefore be downloaded and completed in Adobe Reader. We will then flatten them for ease of use by the reviewers. If you would like to reference the guidance text as you complete the template, please access these flattened versions at <http://www.nature.com/authors/policies/availability.html>.

Author Rebuttal, second revision:

NCB-A49883B-Z – Monteiro *et al.* - A mechanosensitive caveolae-invadosome interplay drives matrix remodeling for cancer cell invasion

Detailed point-by-point response

Reviewer #1:

The manuscript was significantly improved. The only concern is on the nanobar experiments. The dimension of the nanobar is 300 nm in width and 900 nm in height. This is a fine experiment, but the diameter of the collagen fibers in the other experiments would be better to be compared. 300 nm is in the optical imaging range, and the authors should have an estimate of the diameter distribution of the collagen fibers in their experiments. Or still the authors can coat the collagens on the nanobar.

We would like to thank this Referee for her/his constructive comments on our manuscript.

As suggested by the Referee, we estimated the diameter distribution of the collagen fibrils in our assay based on platinum replica transmission electron microscopy (PREM) images of the collagen network and found a mean fibril diameter of 75 ± 28 nm (N = 61 fibrils). This value corresponds to a minimal dimension as individual fibrils were frequently bundled due to network reorganization by the cells, giving rise to larger dimension of the fibers in the collagen network. These data, that are now reported in Extended Data Fig.6a-c of the revised manuscript legitimate the use of the 300 x 900 nm nanobar array as a mimic of the collagen fibril array.

Reviewer #2:

Remarks to the Author:

The authors have carried out a substantial number of new experiments addressing all the points I raised in the initial review. These have strengthened the conclusions made in this interesting study.

We thank this Referee for her/his positive comment on our manuscript.

Reviewer #3:

The referee acknowledges to have confused clathrin-coated pit-mediated adhesive cell-collagen interactions with the here reported caveolar structures associated with juxtaposed zones of collagenolysis. I appreciate, and agree to, the corrective statement made by the authors in the rebuttal letter. In addition, the authors have refined their claims to now propose two independent interaction systems side-by-side, which cooperate for grabbing and cleaving collagen fibrils, which is better supported by their data than the previous disputed claim of sequential maturation and interconversion of both systems. As a consequence, originality and novelty of the here reported findings is generally high, in contrast to my earlier ill-fated statements.

In the revised manuscript, new time-lapse data convincingly show the deformation of collagen fibrils in association with Tsk5 kinetics and somewhat reduced bending after downregulation of Tsk5, in line with bending as a result of invadopod-associated collagen remodeling. New data further include decrease of collagenolysis after interference with actin dynamics, and 3D images demonstrating alternating Cav1/cortactin domains along collagen fibrils in cancer cells invading 3D collagen matrix. The newly included videos are of high quality and nicely illustrate aspects of this process. These new data and the conceptual reframing strengthen this work significantly.

We are grateful to this referee for his/her constructive criticisms and his/her open mindedness is much appreciated.

However, two key claims, namely (i) the conceptual framing as “invadopodia”-mediated collagen degradation and (ii) the relevance of this machinery for collagen internalization, remain unconvincing and potentially misleading.

1. Terminology: The authors maintain the term “invadopodia” or “invadopodal structures”, however no evidence for a protrusive (i.e., “pod”) is provided throughout the manuscript. This differs greatly from classical reports where cells drill a little hole by means of a protrusion. Potentially, a novel type of lytic structures was discovered here, which are morphologically distinct from invadopodia. Hence, the use of this term based on molecular markers alone, but without convincing morphological evidence, should call for the use of a more neutral term.

We agree that the term ‘invadopodia’ should be used for structures that are protrusive by definition. We would like to briefly comment on the fact that the main experimental construct used to assay invadopodia formation and activity in hundreds of publications, the so-called gelatin assay, relies on a crosslinked gelatin coat that is typically 50–100 nm thick, thus leaving limited space for protrusive activity (plasma membrane is typically ~10 nm thick and the lamellipodial actin meshwork is several hundred nm wide). It is why in our study we differently and exclusively relied on a relevant pathophysiological matrix construct and a mimic of the interstitial tumor tissue consisting of fibrillar acid-extracted type I collagen network. Indeed, under these conditions, we and others observed that tumor cells form non-protruding cell-fibril contact sites with the inherent ability to lyse and weaken the collagen fibrils. We also agree that in this respect the ‘invadopodia’ terminology could be confusing and thus, as suggested by this Referee, we switched to a more neutral and generic term and designed the cell-ECM proteolytic contacts as ‘invadosomes’ throughout the revised manuscript.

2. The evidence that the identified structures support collagen internalization remains indirect. This is an important aspect of their work, as it differs from the invadopod concept (which would not be expected to result in collagen internalization after degradation – at least I am not aware of published evidence for this). To support this claim, the authors should provide data showing change of endosomal collagen in response to molecular interference with caveolar structures. This could be easily reached by re-probing the molecular interference samples (e.g., Fig. 4, Fig 6 f,g) for endocytosed collagen.

This point is well-taken. As suggested by the Referee we designed an assay to address specifically the uptake of collagen and the interference of caveolar structures with this process. Our assay is based on antibody labeling of extracellular type I collagen fibrils. Labeling was performed at the end of the incubation of the different cell populations on top of the CY5-labelled type I collagen network (i.e. control MDA-MB-231 cells or cells knocked down for Caveolin-1, Cavin1 or MT1-MMP). Samples were

unfixed/unpermeabilized and labelling of the collagen fibrils outside the cells with anti-collagen 1 antibodies was performed at 4°C to block further internalization. After detection of anti-collagen 1 mouse IgGs with fluorescently labelled secondary antibodies, internalized collagen can be visualized as it is visible in CY5 channel only. These experiments that are described in Fig.7g-l of the revised manuscript, show a significant reduction of collagen uptake upon Cav1 or Cavin1 knockdown indicating dependency on caveolar structures for ECM uptake.

Minor points

The representation of the change of collagen pattern in Fig. 4c is impossible to appreciate, due to the blue color of collagen and small image size. Thresholded single-channel representations of collagen organization and cortactin are needed to link the images to the data.

Based on this Referee's comment and to improve data visibility, we modified the entire figure and extended figure set and chosen a yellow LUT to depict the organization of the type I collagen network and Cyan and Magenta LUTs to show additional markers. Whenever possible, we presented high magnification insets and single channel representation of the collagen network organization and individual markers (Fig1a,b,c,d; Fig3a,c,f; Fig4a,e,h; Fig5b,e,f; Fig6d,e,f; Fig7a,b,e,g; Fig8a).

In addition, fibril orientation index for cell-free areas should be included in Fig. 4c as an internal negative control. Also other figures should be re-checked for clarity of key claims related to collagen fibril organization.

Orientation/alignment index data corresponding to cell-free areas have been added in revised Fig. 3e (corresponding to Fig.4e -not Fig. 4c - in the previous submission).

The authors repeatedly claim that colocalization analysis with two or multiple polyclonal rabbit antibodies is impossible, which however ignores the possibility of sequential immunostaining, with a removal step of labelling antibody in between imaging (e.g., PMID: 28692376). Because reports previously associated Tsk6 with focal adhesions in cell models under certain conditions, the authors should, at least, consider to detail

their considerations associated with Tsk5 distribution and lytic structures versus focal adhesions (provided in the rebuttal letter), in brief form, in the results section (e.g., Line 127).

Data showing the peripheral distribution of paxillin-positive focal adhesions in contrast to the more internal distribution of cortactin-positive invadosomes are now provided in Extended Data Fig1g of the revised manuscript.

Decision Letter, third revision:

Our ref: NCB-A49883C

10th August 2023

Dear Dr. Chavrier,

Thank you for submitting your revised manuscript "A mechanosensitive caveolae-invadosome interplay drives matrix remodelling for cancer cell invasion" (NCB-A49883C). It has now been seen by the original referees and their comments are below. The reviewers find that the paper has improved in revision, and therefore we'll be happy in principle to publish it in Nature Cell Biology, pending minor revisions to satisfy the referees' final requests and to comply with our editorial and formatting guidelines.

As FYI only: (If the current version of your manuscript is in a PDF format, please email us a copy of the file in an editable format (Microsoft Word or LaTeX)-- we can not proceed with PDFs at this stage.)

Thank you again for your interest in Nature Cell Biology Please do not hesitate to contact me if you have any questions.

Sincerely,
Daryl

Daryl Jason Verzosa David, PhD

Senior Editor, Nature Cell Biology

Nature Portfolio

Heidelberger Platz 3, 14197 Berlin, Germany
Email: daryl.david@nature.com
ORCID: <https://orcid.org/0000-0002-9253-4805>

Reviewer #1 (Remarks to the Author):

I think the manuscript can be accepted. I appreciate your great effort on this work.

Reviewer #3 (Remarks to the Author):

The authors have added convincing results on the dependence of collagen internalization on Cav1 expression. They further improved the color code of many image panels for clarity of collagen 1 structure and replaced the term invadopod adequately by the neutral term "invadosome". In addition, no overlap between ring-like cortactin-positive structures and the focal adhesion marker paxillin was detected.

Minor points

The data basis of Fig. 7g-I needs to be added (independent experiments, number of cells, type of statistical analysis).

The colocalization analysis of cortactin (invadosomes) and paxillin (focal adhesions) indeed shows preferential zonal compartments of each marker in arch-like zones, whereas in both peripheral and more central focal structures colocalization can be appreciated (whitish regions). This may suggest partial overlap of invadosomes and focal adhesions in these compartments. Hence, whereas cortactin-positive linear structures lack significant paxillin staining and the analyses throughout this manuscript are likely robust enough to detect this invadosome compartment, the authors should still add a disclaimer that partial overlap of focal adhesions and more focal invadosomes can be present in other regions.

Decision Letter, final checks:

Our ref: NCB-A49883C

30th August 2023

Dear Dr. Chavrier,

Thank you for your patience as we've prepared the guidelines for final submission of your Nature Cell

Biology manuscript, "A mechanosensitive caveolae-invadosome interplay drives matrix remodelling for cancer cell invasion" (NCB-A49883C). Please carefully follow the step-by-step instructions provided in the attached file, and add a response in each row of the table to indicate the changes that you have made. Please also check and comment on any additional marked-up edits we have proposed within the text. Ensuring that each point is addressed will help to ensure that your revised manuscript can be swiftly handed over to our production team.

In recognition of the time and expertise our reviewers provide to Nature Cell Biology's editorial process, we would like to formally acknowledge their contribution to the external peer review of your manuscript entitled "A mechanosensitive caveolae-invadosome interplay drives matrix remodelling for cancer cell invasion". For those reviewers who give their assent, we will be publishing their names alongside the published article.

Nature Cell Biology offers a Transparent Peer Review option for new original research manuscripts submitted after December 1st, 2019. As part of this initiative, we encourage our authors to support increased transparency into the peer review process by agreeing to have the reviewer comments, author rebuttal letters, and editorial decision letters published as a Supplementary item. When you submit your final files please clearly state in your cover letter whether or not you would like to participate in this initiative. Please note that failure to state your preference will result in delays in accepting your manuscript for publication.

Cover suggestions

COVER ARTWORK: We welcome submissions of artwork for consideration for our cover. For more information, please see our guide for cover artwork.

Nature Cell Biology has now transitioned to a unified Rights Collection system which will allow our Author Services team to quickly and easily collect the rights and permissions required to publish your work. Approximately 10 days after your paper is formally accepted, you will receive an email in providing you with a link to complete the grant of rights. If your paper is eligible for Open Access, our Author Services team will also be in touch regarding any additional information that may be required to arrange payment for your article.

Please note that *Nature Cell Biology* is a Transformative Journal (TJ). Authors may publish their research with us through the traditional subscription access route or make their paper immediately open access through payment of an article-processing charge (APC). Authors will not be required to make a final decision about access to their article until it has been accepted. Find out more about Transformative Journals

Authors may need to take specific actions to achieve compliance with funder and

institutional open access mandates. If your research is supported by a funder that requires immediate open access (e.g. according to Plan S principles) then you should select the gold OA route, and we will direct you to the compliant route where possible. For authors selecting the subscription publication route, the journal's standard licensing terms will need to be accepted, including self-archiving policies. Those licensing terms will supersede any other terms that the author or any third party may assert apply to any version of the manuscript.

Please use the following link for uploading these materials:
[Redacted]

Best regards,

Kendra Donahue
Staff
Nature Cell Biology

On behalf of

Daryl Jason Verzosa David, PhD

Senior Editor, Nature Cell Biology
Nature Portfolio

Heidelberger Platz 3, 14197 Berlin, Germany
Email: daryl.david@nature.com
ORCID: <https://orcid.org/0000-0002-9253-4805>

Reviewer #1:

Remarks to the Author:

I think the manuscript can be accepted. I appreciate your great effort on this work.

Reviewer #3:

Remarks to the Author:

The authors have added convincing results on the dependence of collagen internalization on Cav1

expression. They further improved the color code of many image panels for clarity of collagen 1 structure and replaced the term invadopod adequately by the neutral term "invadosome". In addition, no overlap between ring-like cortactin-positive structures and the focal adhesion marker paxillin was detected.

Minor points

The data basis of Fig. 7g-I needs to be added (independent experiments, number of cells, type of statistical analysis).

The colocalization analysis of cortactin (invadosomes) and paxillin (focal adhesions) indeed shows preferential zonal compartments of each marker in arch-like zones, whereas in both peripheral and more central focal structures colocalization can be appreciated (whitish regions). This may suggest partial overlap of invadosomes and focal adhesions in these compartments. Hence, whereas cortactin-positive linear structures lack significant paxillin staining and the analyses throughout this manuscript are likely robust enough to detect this invadosome compartment, the authors should still add a disclaimer that partial overlap of focal adhesions and more focal invadosomes can be present in other regions.

Author Rebuttal, third revision:

NCB-A49883C – Monteiro *et al.* - A mechanosensitive caveolae-invadosome interplay drives matrix remodeling for cancer cell invasion

Detailed point-by-point response

Reviewer #3:

Remarks to the Author

The authors have added convincing results on the dependence of collagen internalization on Cav1 expression. They further improved the color code of many image panels for clarity of collagen 1 structure and replaced the term invadopod adequately by the neutral term "invadosome". In addition, no overlap between ring-like cortactin-positive structures and the focal adhesion marker paxillin was detected.

Minor points

The data basis of Fig. 7g-I needs to be added (independent experiments, number of cells, type of statistical analysis).

This information has been added to the legend of figure 7 and all numerical source data are provided in the Source Data files (Monteiro_SourceData_Fig7.xlsx).

The colocalization analysis of cortactin (invadosomes) and paxillin (focal adhesions) indeed shows preferential zonal compartments of each marker in arch-like zones, whereas in both peripheral and more central focal structures colocalization can be appreciated (whitish regions). This may suggest partial overlap of invadosomes and focal adhesions in these compartments. Hence, whereas cortactin-positive linear structures lack significant paxillin staining and the analyses throughout this manuscript are likely robust enough to detect this invadosome compartment, the authors should still add a disclaimer that partial overlap of focal adhesions and more focal invadosomes can be present in other regions.

As requested by the referee, the following disclaimer has been added in the Results section regarding the colocalization analysis of cortactin and paxillin (lines 127-130):

'Of interest, we found no overlap between peripheral focal adhesions stained with paxillin (and weakly positive for cortactin), and more internal, elongated invadosomes strongly labelled for cortactin and negative for paxillin although minimal colocalization cannot be ruled out (Extended Data Fig.1g).'

Final Decision Letter:

Dear Dr Chavrier,

I am pleased to inform you that your manuscript, "A mechanosensitive caveolae-invadosome interplay drives matrix remodelling for cancer cell invasion", has now been accepted for publication in Nature Cell Biology.

Please note that *Nature Cell Biology* is a Transformative Journal (TJ). Authors may publish their research with us through the traditional subscription access route or make their paper immediately open access through payment of an article-processing charge (APC). Authors will not be required to make a final decision about access to their article until it has been accepted. Find out more about Transformative Journals

If you have not already done so, we strongly recommend that you upload the step-by-step protocols used in this manuscript to the Protocol Exchange (www.nature.com/protocolexchange), an open online resource established by Nature Protocols that allows researchers to share their detailed experimental know-how. All uploaded protocols are made freely available, assigned DOIs for ease of citation and are fully searchable through nature.com. Protocols and Nature Portfolio journal papers in which they are used can be linked to one another, and this link is clearly and prominently visible in the online versions of both papers. Authors who performed the specific experiments can act as primary authors for the Protocol as they will be best placed to share the methodology details, but the Corresponding Author of the present research paper should be included as one of the authors. By uploading your Protocols to Protocol Exchange, you are enabling researchers to more readily reproduce or adapt the methodology you use, as well as increasing the visibility of your protocols and papers. You can also establish a dedicated page to collect your lab Protocols. Further information can be found at www.nature.com/protocolexchange/about

You can use a single sign-on for all your accounts, view the status of all your manuscript submissions

and reviews, access usage statistics for your published articles and download a record of your refereeing activity for the Nature Portfolio.

With kind regards,

Daryl

Daryl Jason Verzosa David, PhD

Senior Editor, Nature Cell Biology
Nature Portfolio

Heidelberger Platz 3, 14197 Berlin, Germany
Email: daryl.david@nature.com
ORCID: <https://orcid.org/0000-0002-9253-4805>

** Visit the Springer Nature Editorial and Publishing website at www.springernature.com/editorial-and-publishing-jobs for more information about our career opportunities. If you have any questions please click here.**